# MIMOMamba:
# From Scalar Duality to Matrix-Valued Attention

Yanbo Li [1]  Richard Cornelius Suwandi [1]  Feng Yin [1]  Yiyong Sun [1]  Wei Huang [2]  Wenqiang Pu [3]

## Abstract

The state space duality (SSD) framework, central to modern state-space models (SSMs) such as Mamba, has established an efficient attention-like mechanism by leveraging the commutative property of linear recurrences. However, existing formulations are limited to single-input single-output (SISO) systems that enforce commutativity with a restrictive scalar-identity constraint, which prevents cross-dimensional interactions within the state dynamics. In this work, we generalize SSD to the multi-input multi-output (MIMO) setting by introducing a matrix polynomial parameterization. This approach not only provides a principled way to ensure commutativity for generalized duality but also induces a shared algebraic structure across state transitions, thereby significantly reducing parameter redundancy. Building on this foundation, we present **MIMOMamba**, a multi-head SSM architecture that captures rich cross-dimensional dynamics while retaining linear-time training. Empirical evaluations on a sequence modeling benchmark demonstrate that MIMOMamba achieves performance comparable to or surpassing that of standard Transformers while utilizing substantially fewer parameters.

## 1. Introduction

The Transformer architecture, underpinned by its self-attention mechanism, has become the cornerstone of modern large-scale models (Vaswani et al., 2017). However, its quadratic computational complexity in terms of sequence length poses a substantial challenge for long-context learning applications. The demand for linear-time solutions has led researchers to exploit fundamental algebraic properties of matrix operations to avoid explicitly computing the prohibitive $L \times L$ intermediate attention matrix.

A prominent strategy leverages the **associative property** of matrix multiplication (Katharopoulos et al., 2020), which enables a family of Linear Attention variants (Sun et al., 2023; Yang et al., 2024; Peng et al., 2023). A distinct and complementary line of research is rooted in the *state space duality* (SSD) framework (Dao & Gu, 2024), which exploits the **commutative property** to establish an equivalence between sequential state-space models (SSMs) and parallel attention.

However, existing SSD formulations are inherently restricted to single-input single-output (SISO) systems, as commutativity is enforced through scalar-identity matrices. This restriction introduces a critical limitation: cross-dimensional feature interactions cannot be modeled within the recurrence itself and must instead be delegated to a separate channel mixer. In practice, this limitation is typically addressed by adopting a two-stage design. Several recent efficient SSMs (Dao & Gu, 2024; Gu & Dao, 2023) keep the selective recurrence in a SISO form for SSD/scan efficiency and rely on a separate pointwise mixer for channel interaction. When cross-channel coupling is time-varying and entangled with temporal dynamics, this separation can be insufficient, which motivates multi-input multi-output (MIMO) recurrences that model temporal and cross-channel interactions jointly.

In this work, we propose **MIMOMamba**, a novel architecture that generalizes the SSD framework to the MIMO setting. Our approach is based on a fundamental design dilemma: how to enable rich cross-dimensional dynamics without sacrificing the efficiency of duality. We solve this by introducing a *matrix polynomial parameterization*, where state matrices are constructed as polynomials of a single, shared base matrix.

Crucially, this work moves beyond merely establishing algebraic isomorphism,we frame our contribution through the lens of *parameter efficiency*. Viewed as a time-varying system, a standard Transformer instantiates fully independent transition matrices for every time step. Unlike SSMs, whose underlying dynamics are shared, this independence forces

[1]The Chinese University of Hong Kong [2] Ocean University of China [3]Shenzhen Research Institute of Big Data. Correspondence to: Feng Yin <yinfeng@cuhk.edu.cn>.

*Proceedings of the 43rd International Conference on Machine Learning*, Seoul, South Korea. PMLR 306, 2026. Copyright 2026 by the author(s).

the model to relearn transition rules at each step, leading to massive parameter redundancy. Conversely, prior efficient SSMs often enforce restrictive parameter constraints that have evolved from diagonal matrices (Gu & Dao, 2023) to scalar-identity matrices (Dao & Gu, 2024), although computationally efficient, these constraints severely limit expressivity. MIMOMamba strikes an optimal balance: the shared algebraic basis drastically reduces the parameter count, which is approximately one-third of that of a comparable Transformer, while the time-varying coefficients retain the capacity to model complex, coupled dynamics.

Our main contributions can be summarized as follows:

- We generalize state space duality to the MIMO setting, establishing a formal equivalence between a multi-dimensional SSM recurrence and a matrix-valued structured attention mechanism.

- We introduce a matrix polynomial parameterization that ensures commutativity. We prove that this parameterization is theoretically complete within the commutative subspace when the base matrix is non-derogatory.

- We propose MIMOMamba, a multi-head architecture that leverages our MIMO SSM formulation to enhance model capacity while preserving linear-time training.

- We empirically validate MIMOMamba across multiple benchmarks, including physics-informed SSP forecasting, UEA time-series classification (Bagnall et al., 2018), and comparisons against recent SSM baselines. And wall-clock profiling on GPU confirms favorable memory and throughput scaling.

## 2. Notation

We use a notation system designed to distinguish between local (time-step) elements and global (sequence) structures. Scalars (e.g., $a$), vectors (e.g., $x$), and **basic matrices** (e.g., $W, A$) are denoted by **unbolded** letters. We reserve **bold uppercase letters** (e.g., $\mathbf{M}, \mathbf{X}$) strictly for **block matrices** or operators representing full sequences. The transpose is written as $A^\top$ or $\mathbf{M}^\top$. The vectorization operator $\mathrm{vec}(\cdot)$ stacks the rows of a matrix into a single column vector: $\mathrm{vec}(\mathbf{X}) = [x_0^\top, \ldots, x_{L-1}^\top]^\top \in \mathbb{R}^{DL}$, with inverse $\mathrm{vec}^{-1}$.

We distinguish between two types of products. The symbol $\circ$ denotes the standard **element-wise Hadamard product** for scalars or tensors. In contrast, we introduce $\circledast$ to denote the **block-wise matrix product**.

We consider sequences of length $L$, where the $t$-th element is a vector $x_t \in \mathbb{R}^D$. The entire input sequence is stacked into a matrix $\mathbf{X} \in \mathbb{R}^{L \times D}$. Linear transformations acting on full sequences are represented by block matrices $\mathbf{M} \in$

$\mathbb{R}^{DL \times DL}$, viewed as $L \times L$ blocks where each block $M_{ij} \in \mathbb{R}^{D \times D}$ is a basic matrix. We define the operator action via $\mathbf{M} \triangleright \mathbf{X} \triangleq \mathrm{vec}^{-1}(\mathbf{M}\, \mathrm{vec}(\mathbf{X}))$, which allows the input–output relationship $\mathbf{Y} = \mathbf{M} \triangleright \mathbf{X}$. Time indices appear as subscripts (e.g., $x_t$). For products of time-varying basic matrices, we use the shorthand $A_{i:j} \triangleq A_i A_{i-1} \cdots A_j$ $(i \geq j)$, with $A_{i:i-1} = I$.

## 3. Related Work

This section contextualizes our work within the broader landscape of efficient sequence modeling. We first review the technical foundations of selective SSMs and the duality framework, and then critically analyze the parameter sharing mechanisms that distinguish our approach from prior SISO-based methods and standard Transformers.

### 3.1. Selective State-Space Models and Duality

A selective state-space model (SSM) transforms an input sequence $x$ into an output sequence $y$ through a latent state $h$. For a sequence of length $L$, the evolution is governed by the time-varying recurrence:

$$
\begin{aligned}
h_t &= A_t h_{t-1} + B_t x_t, \\
y_t &= C_t h_t.
\end{aligned}
\tag{1}
$$

The term *selective* emphasizes that the state evolution is *time-varying* (i.e., $A_t, B_t, C_t$ depend on $t$), which is crucial for modeling rich *temporal* dependencies in long sequences. Here, $h_t \in \mathbb{R}^D$ is the hidden state. In the general MIMO setting, the input $x_t \in \mathbb{R}^{D_{in}}$ and output $y_t \in \mathbb{R}^{D_{out}}$ are vectors, implying that $B_t \in \mathbb{R}^{D \times D_{in}}$ and $C_t \in \mathbb{R}^{D_{out} \times D}$ are matrices.

The State Space Duality (SSD) framework (Dao & Gu, 2024) connects the linear recurrence to an attention-like form by unrolling the system over time. Since both the input and output are vector-valued sequences (equivalently, matrices), we use the operator $\triangleright$ (Section 2) to denote the dimensionally consistent action of a (block) matrix on such sequences. With this notation, the input–output mapping can be written compactly as the linear transformation induced by a block semiseparable matrix $\mathbf{M}$: $\mathbf{Y} = \mathbf{M} \triangleright \mathbf{X}$,

where $\mathbf{M}$ is block semiseparable (Vandebril et al., 2005) and is block lower-triangular under the natural time ordering, since causality implies $\mathbf{M}_{ij} = 0$ for $i < j$.

Crucially, the nature of the "attention score" $M_{ij}$ depends on the dimensionality of the system:

**General MIMO Case (Matrix-Valued Attention):** Since $B_j$ and $C_i$ are matrices, the interaction term $M_{ij}$ represents a linear transformation $\mathbb{R}^{D_{in}} \to \mathbb{R}^{D_{out}}$. Consequently, $\mathbf{M}$ acts as a *block matrix*, where each "element" of the attention map is itself a matrix describing the coupling

between timestep $j$ and $i$.

**Mamba-2 Constraint (Scalar-Valued Attention):** To align with standard attention mechanisms (where scores are scalars) and leverage efficient GPU kernels, Mamba-2 restricts the system to the **SISO case** ($D_{in} = D_{out} = 1$).

In the specific case of Mamba-2 (Dao & Gu, 2024), projections $B_t$ and $C_t$ are constrained to be vectors rather than matrices. This restriction forces the interaction term $M_{ij}$ to collapse from a linear transformation into a single *scalar value*. Consequently, the complex state-space recurrence becomes isomorphic to a structured form of linear attention.

By defining the sequence of $C_t$ as Queries (**Q**), $B_t$ as Keys (**K**), and the input **X** as Values (**V**), the system output **Y** can be computed via a masked attention operation: $\mathbf{Y} = \left(\mathbf{L} \circ (\mathbf{Q}\mathbf{K}^\top)\right)\mathbf{V}$.

Here **L** is the SSM-induced causal mask (Appendix C.2).

While this scalar reduction enables the use of highly optimized GPU kernels, it necessarily limits the model to a SISO regime. As a result, the model treats the feature dimension as a collection of independent 1D channels, thereby preventing the rich cross-dimensional vector-vector interactions inherent to the general matrix-valued formulation (as visualized in Figure 1).

In practice, many efficient SSM architectures (e.g., the Mamba family (Dao & Gu, 2024; Gu & Dao, 2023)) adopt a layered design that **decouples** temporal modeling from cross-channel interaction: to preserve fast SSD/scan-style kernels enabled by scalar (or scalar-identity) transitions, the selective recurrence is kept **SISO**, while feature mixing is delegated to a separate pointwise *channel mixer*. This decomposition works well when cross-channel dependencies are simple or approximately time-invariant. However, when cross-channel coupling is *time-varying* and intertwined with temporal dynamics, pushing channel interaction outside the recurrence becomes a structural limitation, motivating **MIMO** state-space formulations that model temporal and cross-channel interactions *jointly*.

However, for more complex systems where cross-channel dependencies are themselves *time-varying* and entangled with temporal dynamics, enforcing a SISO selective SSM and outsourcing channel interaction to a separate mixer becomes a modeling limitation. This motivates moving beyond SISO toward **MIMO** state-space formulations. With MIMO recurrences, an SSM layer can **jointly** model temporal dependence and time-varying cross-channel coupling, which leads to MIMOMamba.

### 3.2. The Parameter Sharing Trade-off: SISO vs. Transformer

To realize the SSD framework, previous work (Dao & Gu, 2024) imposes a **scalar-identity constraint** on the state transition matrix, specifically setting $A_t = a_t I$. This design choice is driven by two key motivations. First, from a **computational** perspective, a scalar matrix commutes with any projection matrix. This property is mathematically necessary to factorize the attention matrix into the efficient element-wise form, enabling fast linear-time training.

Second, from a **modeling** perspective, this constraint aligns with a SISO formulation: although the latent state $h_t$ lies in $\mathbb{R}^D$, the input $x_t$ is projected to a scalar. As a result, $A_t = a_t I$ effectively yields **one-dimensional** autoregressive dynamics, with the $D$-dimensional state acting primarily as a memory bank governed by a shared scalar transition.

While this constraint enables the duality described above, it represents a specific point on the efficiency-expressivity trade-off curve. Existing SSMs, including the Mamba series, predominantly rely on these SISO factorizations. This paradigm effectively treats the feature dimension $D$ as $D$ independent heads, which in turn requires a drastic reduction in the complexity of each head to maintain computational tractability. Consequently, state transition matrices are typically constrained to be extremely simple—such as diagonal (Gu & Dao, 2023) or scalar-identity (Dao & Gu, 2024) matrices.

Despite its efficiency, the scalar-identity (or diagonal) constraint effectively yields channel-wise dynamics, where cross-channel interactions are largely handled by separate mixing layers. This creates an efficiency–expressivity trade-off: can we incorporate **time-varying cross-channel coupling within** the recurrence while retaining an SSD/scan-compatible parameter-sharing mechanism?

In this work, we address this through a novel **matrix polynomial parameterization**, which serves as a bridge between theoretical duality and practical efficiency. On the theoretical side, constructing state matrices as polynomials of a shared base matrix guarantees the mutual commutativity required to generalize the SSD framework to the MIMO setting.

More importantly, from a **parameter-sharing perspective**, our parameterization replaces fully independent time-varying state matrices with a shared base matrix modulated by low-dimensional coefficients. If one parameterizes $(A_t, B_t, C_t)$ independently at each step, the model incurs substantial parameter redundancy. In contrast, MIMOMamba enforces a rigorous parameterization constraint: dynamics are governed by time-varying scalar coefficients modulating a **single, shared base matrix**. This algebraic sharing mechanism can substantially reduce the number of

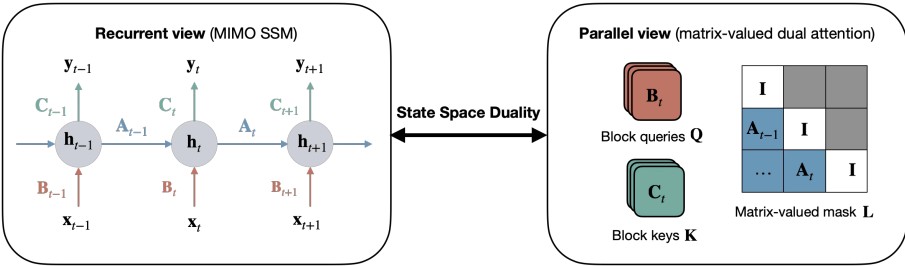

*Figure 1.* Recurrent and parallel views of State Space Duality (SSD). **Recurrent view** (left): The SSM evolves as $h_t = A_t h_{t-1} + B_t x_t$, $y_t = C_t h_t$. **Parallel view** (right): The same computation is expressed as structured attention $\mathbf{Y} = (\mathbf{L} \circledast (\mathbf{Q}\mathbf{K}^\top)) \triangleright \mathbf{X}$, with $\mathbf{Q} = \{C_t\}$, $\mathbf{K} = \{B_t\}$, and $L_{ij} = A_{i-1:j}$. Prior work enforces $A_t = a_t I$ to ensure duality, but this limits dynamics to independent scalar channels. MIMO-Mamba generalizes $A_t$ beyond scalar identity, enabling cross-channel interactions while retaining both inference modes.

free parameters relative to an unshared parameterization. In our configurations, the reduction is often a constant factor (e.g., $\sim 3\times$), while retaining the ability to represent rich, coupled interactions.

**Concurrent Work.** Mamba-3 (Lahoti et al., 2026) also explores a MIMO formulation for SSMs. However, the two works differ fundamentally in motivation and methodology. Mamba-3 (Lahoti et al., 2026) is primarily **engineering-driven**, which adopts MIMO as a means to increase the arithmetic intensity of the decoding step, thereby optimizing hardware utilization on GPUs. To enhance expressivity, it introduces complex-valued state dynamics, which are proven to be equivalent to data-dependent Rotational Position Embeddings (RoPE) (Su et al., 2024). In contrast, our work is grounded in a **theoretical first-principles**. We address the fundamental algebraic challenge of generalizing State Space Duality to the MIMO setting. Instead of using standard dense matrix expansions, we introduce a principled **matrix polynomial parameterization** that strictly enforces the commutativity required for the duality to hold. While Mamba-3 (Lahoti et al., 2026) leverages MIMO for inference throughput, our framework focuses on mathematically restoring the rich, coupled dynamics within the SSD theoretical framework.

# 4. Method: MIMOMamba via Matrix Polynomials

We present **MIMOMamba**, a generalized SSM architecture that extends State Space Duality (SSD) from scalar-identity constraints to rich, MIMO interactions. To achieve this while preserving the dual inference modes, we introduce a rigorous algebraic parameterization.

## 4.1. Matrix Polynomial Parameterization

The core challenge in MIMO-SSD is the *Commutativity Bottleneck*: for the efficient dual form to exist, the time-

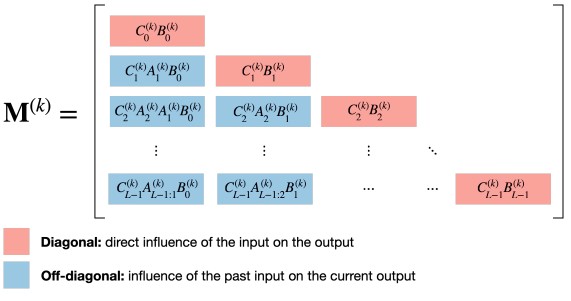

*Figure 2.* Visualization of the Matrix-Valued Dual Attention mechanism. Unlike standard SISO SSMs that use scalar masks, MIMO-Mamba employs block matrices for Queries ($\mathbf{Q}^{(k)}$), Keys ($\mathbf{K}^{(k)}$), and the causal Mask ($\mathbf{L}^{(k)}$), as defined in Eqs. (7) and (8). The entries of the mask $\mathbf{L}^{(k)}$ are themselves $D \times D$ state transition matrices evolving in a commutative polynomial ring, enabling rich cross-dimensional dynamics.

varying state transition matrices must be able to commute. We resolve this by constraining the state matrices $A_t, B_t, C_t$ to be within the polynomial ring of a shared base matrix $\mathbf{A} \in \mathbb{R}^{D \times D}$. This design is grounded in a three-step algebraic logic chain:

**1. Sufficiency via Polynomial Structure.** It is a fundamental property of linear algebra that any two polynomials of the same matrix commute. By parameterizing $A_t = \sum_{k=0}^{p} c_k A^k$, we guarantee that $A_t A_{t'} = A_{t'} A_t$ for all $t, t'$, which enforces the commutativity required for SSD (proof in Appendix E.2).

**2. Optimality via the Centralizer Theorem.** Does restricting matrices to polynomials limit expressivity? The Centralizer Theorem (Horn & Johnson, 2012) (see detailed discussion in Appendix F.2.3) provides a condition for optimality: if the base matrix $A$ is *non-derogatory* (i.e., its minimal polynomial equals its characteristic polynomial),

then the set of all matrices commuting with $A$ is *exactly* the set of polynomials in $A$. Under this condition, our parameterization captures the *entire* valid parameter space for commuting MIMO dynamics.

**3. Realization via Random Initialization.** Practically, ensuring $A$ is non-derogatory is straightforward. A sufficient condition for a matrix to be non-derogatory is having **distinct eigenvalues** (algebraic multiplicity of 1). Since the set of matrices with repeated eigenvalues has measure zero in $\mathbb{R}^{D \times D}$, standard random initializations (e.g., Kaiming or Xavier) yield a non-derogatory base matrix with probability 1. Formally, this follows since repeated eigenvalues correspond to the zero set of the discriminant of the characteristic polynomial; see Appendix F (Prop. F.7). Thus, our theoretically optimal condition is naturally satisfied at initialization.

### 4.2. Deriving Matrix-Valued Duality

With the commutativity established, the unrolling of the MIMO recurrence follows the same structural derivation as Eq. 1 in Section 3. However, unlike prior work where the interaction terms degenerate to scalars, our formulation preserves the full matrix structure.

Substituting the polynomial parameterization into the recurrence allows the transition matrices to pass through each other, enabling the grouping of terms into a parallel form. The resulting **Matrix-Valued Attention** is computed as $\mathbf{Y} = (\mathbf{L} \circledast (\mathbf{Q}\mathbf{K}^\top)) \triangleright \mathbf{X}$.

The mask $\mathbf{L}$ is no longer a scalar decay but a matrix-valued operator derived from state transition matrices(Appendix C.2). This upgrades the fundamental atomic operation of the model: unlike the scalar-identity constraints in prior SSD frameworks that treat feature channels independently, MIMOMamba performs **dense matrix mixing** directly within the attention mechanism. This allows the model to capture rich cross-feature correlations and coupled dynamics at every timestep, fully realizing the potential of the matrix-valued duality.

### 4.3. Architecture Design

We implement the full model using a Direct Sum of Algebras. We instantiate $H$ parallel heads, where the $h$-th head is governed by its own unique base matrix $A^{(h)}$. This allows the model to learn distinct "dynamical modes" (e.g., different oscillation frequencies or decay rates) in parallel, further enhancing expressivity while maintaining the parameter sharing benefits described in Section 3.

**Endogenous Order vs. Explicit Position Embedding.** A distinct advantage of our formulation is its principled treatment of sequence order. In the standard Transformers, the attention mechanism is permutation-invariant, necessi-

tating explicit Position Embeddings (PE). In contrast, the state evolution in MIMOMamba is governed by the powers of the base matrix $A^k$. This structure imposes a strong, yet principled inductive bias for temporal order.

We note that concurrent work (Lahoti et al., 2026) approaches this by defining dynamics over the complex field, explicitly drawing an equivalence between complex-valued SSMs and data-dependent RoPE (Su et al., 2024). Our analysis suggests that such complex parameterizations and explicit positional biases can be avoided in our framework. The matrix polynomial recurrence naturally encodes the relative positions through the algebraic properties of the real-valued operator $A$. This intrinsic property allows our framework to validate a "first principles" approach to dynamics modeling: the sequence order is structurally enforced by the recurrence itself rather than injected via auxiliary mechanisms (see Appendix D for formal derivation).

## 5. Generalized Dual Attention

In this section, we utilize the algebraic structure established in Section 4 to derive the primary theoretical result of this work: the *matrix-valued state space duality*.

While the polynomial ring $\mathbb{R}[A]$ provides a rigorous commutative foundation, relying on a single base matrix restricts the latent state capacity. To address this, we employ a **Parallel Multi-Head Formulation** via a direct sum of algebras, $\mathcal{A}_{total} = \bigoplus_{k=1}^{H} \mathbb{R}[A^{(k)}]$ (detailed construction and analysis in Appendix B.5). Crucially, this framework overcomes the fundamental obstacle of non-commutativity in generic MIMO systems. The direct sum structure ensures that commutativity is strictly preserved *within* each independent subspace $\mathbb{R}[A^{(k)}]$, allowing the MIMO recurrence to be factorized into a structured attention mechanism where queries, keys, and masks are matrix-valued. Since the heads operate disjointly, we proceed to derive the duality for a single generic head $k$ without loss of generality.

### 5.1. Commutative Factorization of the MIMO Recurrence

Consider the output computation for a single algebraic subspace (head) $k$. Unrolling the linear recurrence $h_t = A_t h_{t-1} + B_t x_t$ yields the output $y_i$ as a superposition of past inputs, mediated by a chain of state transitions: $y_i^{(k)} = \sum_{j=0}^{i} C_i^{(k)} A_{i:j+1}^{(k)} B_j^{(k)} x_j^{(k)}$.

This operation can be viewed as a global matrix transformation $\mathbf{Y}^{(\mathbf{k})} = \mathbf{M}^{(k)} \triangleright \mathbf{X}^{(\mathbf{k})}$, where $\mathbf{M}^{(k)}$ is a block-lower-triangular matrix with blocks $\mathbf{M}_{ij}^{(k)} \in \mathbb{R}^{D \times D}$. For $i > j$, the $(i, j)$-th block is given by: $\mathbf{M}_{ij}^{(k)} = C_{i-1}^{(k)} A_{i-1:j}^{(k)} B_{j-1}^{(k)}$.

In a MIMO system, the factors need not lie in a common commutative subalgebra; consequently, the product is order-

dependent in general and cannot be freely rearranged. Under our construction in Section 4, $A_t^{(k)}, B_t^{(k)}, C_t^{(k)} \in \mathbb{R}[A^{(k)}]$, and hence commute. This allows us to reorder the product and factor each attention block into a "state-dynamics" term and an "input–output" term: $M_{ij}^{(k)} = A_{i-1:j}^{(k)}(C_{i-1}^{(k)} B_{j-1}^{(k)})$, with a detailed derivation and the annotated factorization deferred to Appendix C.3.

**Algebraic Simplification.** It is worth highlighting a powerful property of the *dynamics term* derived above. Since each state matrix is parameterized as a polynomial $A_t^{(k)} = f_t(A^{(k)})$, the product of these matrices corresponds directly to the product of their underlying scalar polynomials. Formally, the functional calculus implies:

$$\prod_{t=j+1}^{i} A_t^{(k)} = \prod_{t=j+1}^{i} f_t(A^{(k)}) = \left( \prod_{t=j+1}^{i} f_t \right)(A^{(k)}). \quad (2)$$

Eq. (2) reveals that the complex sequence of matrix multiplications compresses into a *single* polynomial evaluation. The accumulated dynamics are governed by the scalar product polynomial $P(x) = \prod_{t=j+1}^{i} f_t(x)$, evaluated at the base matrix argument $A^{(k)}$. This algebraic reduction not only clarifies the structural behavior of the mask but, as we will discuss in Section 6, lays the foundation for accelerating the computation via Fast Fourier Transforms (FFT) (Cooley & Tukey, 1965).

### 5.2. Matrix-Valued Attention Primitives

Using the factorization derived in Appendix C.3, we formalize the generalized dual attention form. The sequence transformation is equivalent to: $\mathbf{Y}^{(\mathbf{k})} = \left( \mathbf{L}^{(k)} \circledast (\mathbf{Q}^{(k)}(\mathbf{K}^{(k)})^\top) \right) \triangleright \mathbf{X}^{(\mathbf{k})}$, where $\circ$ denotes the block-wise Hadamard product. This generalized attention structure is visualized in Fig. 2, which illustrates how the scalar mask entries are replaced by structured matrix blocks. Unlike standard attention where $Q, K \in \mathbb{R}^{L \times D}$ and the attention matrix is $L \times L$ scalars, our primitives are block matrices defined as follows:

- **Block Queries ($\mathbf{Q}^{(k)}$) and Keys ($\mathbf{K}^{(k)}$):** The queries and keys are formed by stacking the local projection matrices.

  Both are matrices of size $(LD) \times D$. The product $\mathbf{Q}^{(k)}(\mathbf{K}^{(k)})^\top$ yields an $(LD) \times (LD)$ matrix, functionally treating the $D \times D$ interaction blocks as atomic units.

- **Matrix-Valued Mask ($\mathbf{L}^{(k)}$):** The mask $\mathbf{L}^{(k)}$ encodes the state evolution. It is a block-lower-triangular matrix where the $(i, j)$-th block is the accumulated state transition: $L_{ij}^{(k)} = A_{i-1:j}^{(k)}$ if $i > j$, else $I$ or $0$

This formulation generalizes the concept of "attention" from scalar inner products to matrix-matrix products. The mask $\mathbf{L}^{(k)}$ is no longer a fixed scalar decay or a scalar gate applied uniformly across channels (i.e., a scalar-identity matrix, as in Mamba-2 (Dao & Gu, 2024)), but a learnable, matrix-valued operator that can implement feature-wise mixing, including rotations and scaling, within $\mathbb{R}^D$.

To bridge the gap between this theoretical formulation and practical implementation, we describe the forward-pass computation at a high level here, and defer the complete step-by-step pseudocode to Appendix C (Algorithm 1). Crucially, this algorithmic structure elucidates the source of the parameter-efficiency advantage observed in our experiments. In a fully expressive time-varying system (like a standard Transformer), the transition ($A_t$), input ($B_t$), and output ($C_t$) matrices would be independent dense parameters, incurring a cost of $\approx 3D^2$. In contrast, our **matrix polynomial parameterization** generates all three dynamic matrices from a **single shared base matrix A**. Since the scalar coefficients $\alpha, \beta, \gamma$ introduce negligible overhead compared to the quadratic cost of the dense basis, the parameter complexity of the core dynamics is effectively reduced by a factor of three ($3D^2 \to 1D^2$).

Finally, the computation explicitly codifies the SSD through conditional execution branches. Depending on the computational context, the flow selects either the **Parallel Mode**, which leverages the block-matrix masks derived above for efficient parallel training, or the **Recurrent Mode**, which utilizes the underlying state-space dynamics for fast autoregressive inference (see Appendix C.6 for the detailed procedure). This structure confirms that both views are mathematically equivalent realizations of the same underlying system.

We conclude this section by noting that our matrix-valued duality constitutes a strict superset of the SISO duality in Mamba-2 (Dao & Gu, 2024). As formally derived in Appendix C.4, restricting the base matrix to a scalar identity (i.e., $A^{(k)} = \lambda I$) decouples the feature dimensions into independent scalar channels.

## 6. Computational Complexity and Hardware-Aware Algorithms

While the matrix polynomial parameterization established in Section 4 provides a commutativity guarantee, realizing it efficiently requires additional algorithmic structure. A naive implementation of the matrix-valued duality (Section 5) would involve sequential products of dense $D \times D$ matrices, incurring a prohibitive cubic cost $\mathcal{O}(L \cdot D^3)$ for sequence length $L$ (Trefethen & Bau, 2022).

To overcome this bottleneck, our framework leverages two complementary mathematical properties. First, from a struc-

tural perspective, our formulation strictly preserves the **block-semiseparable structure** (Vandebril et al., 2005) characteristic of efficient SSMs (Dao & Gu, 2024) ( see Appendix C(Prop. C.2) ). This structural inheritance ensures that our generalized operator remains compatible with mature hardware-aware parallel scan algorithms (Pernet & Storjohann, 2018).

Second, and more importantly for asymptotic complexity, the algebraic structure of $\mathbb{R}[A]$ facilitates efficient polynomial multiplication via FFT (Cooley & Tukey, 1965).

**A note on the role of FFT in the current implementation.** The FFT-based polynomial multiplication described below is a *theoretically grounded component designed for future scalability*, not a source of speedup in our current experimental configuration. At degree $p = 4$, the polynomial coefficient vectors are short enough that direct evaluation is more efficient than the FFT pipeline (which involves padding, forward/inverse transforms, and modular reduction). The practical advantage of FFT-based multiplication emerges at higher polynomial degrees or larger state dimensions, where the $\mathcal{O}(p \log p)$ scaling becomes favorable over the $\mathcal{O}(p^2)$ direct convolution.

In this section, we demonstrate how the algebraic structure of $\mathbb{R}[A]$ unlocks an algorithmic path to reduce arithmetic and memory traffic *in principle*, bridging expressive MIMO modeling with scan-compatible implementations.

### 6.1. FLOP Reduction via Polynomial Isomorphism

The most computationally expensive step in computing the dual attention mask $\mathbf{L}^{(k)}$ is the accumulation of state transitions $\prod_t A_t^{(k)}$. As hinted in Eq. (2), the polynomial parameterization induces an algebra isomorphism between the matrix ring $\mathbb{R}[A^{(k)}]$ and the scalar polynomial quotient ring $\mathbb{R}[x]/\langle \mu(x) \rangle$ (Dummit & Foote, 2004), where $\mu(x)$ is the minimal polynomial of the base matrix $A^{(k)}$ (Horn & Johnson, 2012; Golub & Van Loan, 2013).

This isomorphism allows us to lift the computation from the expensive matrix domain to the more efficient polynomial coefficient domain:

1. **Coefficient-Space Multiplication:** Instead of multiplying matrices $A_t^{(k)} A_{t-1}^{(k)}$, we compose their coefficient representations by multiplying the associated polynomials $f_t(x) f_{t-1}(x)$ in $\mathbb{R}[x]/\langle \mu(x) \rangle$.

2. **FFT Acceleration:** Polynomial multiplication corresponds to convolution of coefficient sequences. FFT-based convolution can reduce the cost of multiplying degree-$(D-1)$ polynomials compared to naive coefficient-space multiplication (Cooley & Tukey, 1965).

Importantly, the operations above are purely numerical: the time-varying dynamics are represented by floating-point coefficient vectors, and the corresponding algebraic composition reduces to coefficient-space convolutions and modular reductions in $\mathbb{R}[x]/\langle \mu(x) \rangle$. This avoids materializing dense $D \times D$ products at each step and provides a concrete path toward sub-cubic arithmetic *for transition accumulation*, without computing dense $D \times D$ products at each step.

Our discussion in this section concerns algorithmic structure and asymptotic trends under standard arithmetic cost models. End-to-end throughput and memory efficiency depend on kernel-level engineering choices (e.g., batching, layout, and fusion) and are left to future work.

Note that during inference, we can cache the evaluated transition matrix $A_{\text{step}}^{(k)} = (\prod f_t)(A^{(k)})$. This yields a deterministic matrix-vector update cost of $\mathcal{O}(D^2)$ per head.

### 6.2. I/O-Aware Scaling: From Communication to SRAM-Resident Computation

Beyond arithmetic cost, practical performance is often dominated by memory traffic and communication.

In modern large-scale training, system performance is increasingly constrained by the *memory wall*—the bottleneck imposed by limited memory bandwidth and inter-GPU communication latency (Dao et al., 2022b; Gholami et al., 2024). While standard MIMO models often involve materializing or communicating dense, time-varying operators $A_t, B_t, C_t \in \mathbb{R}^{D \times D}$, which can incur an $\mathcal{O}(D^2)$ payload when treated explicitly.

**Communication Efficiency via Coordinate Representation.** By fixing a shared, time-invariant base matrix $A$, the time-varying dynamics are entirely encapsulated in the scalar polynomial coefficients via the mapping $\phi : \mathbb{R}^{p+1} \to \mathbb{R}^{D \times D}$. In distributed settings where time-varying operators would otherwise be explicitly synchronized, it suffices to communicate the coefficient vectors $\{c_{t,j}\}$ that parameterize them. This reduces the communication volume from $\mathcal{O}(H \cdot D^2)$ to $\mathcal{O}(H \cdot D)$, effectively compressing the synchronization payload by a factor of $D$.

**Potential for SRAM-Resident Algebra.** Beyond communication, this algebraic structure offers a theoretical pathway for optimizing GPU memory hierarchy usage. Drawing inspiration from the I/O-aware tiling scaling of FlashAttention (Dao et al., 2022b), our parameterization can potentially keep intermediate computations in SRAM. Since the state matrices form a commutative ring isomorphic to the polynomial quotient ring (Dummit & Foote, 2004), operations can theoretically be performed in the low-dimensional "coefficient space" within fast SRAM, deferring the bandwidth-intensive materialization of dense matrices to HBM until necessary.

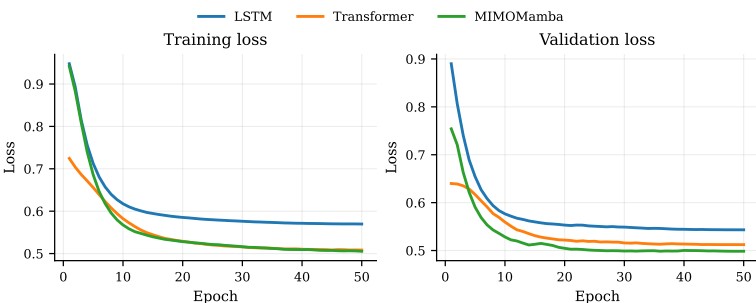

*Figure 3.* **Training and Validation Loss Dynamics.** Comparison of convergence behaviors over 50 epochs. **MIMOMamba** (green) exhibits a steeper initial descent and stabilizes at a lower loss floor compared to the Transformer (orange) and LSTM (blue), validating the optimization efficiency of the shared polynomial parameterization.

### 6.3. Empirical Verification of Efficiency Claims

We corroborate the theoretical analyses above with wall-clock profiling on an NVIDIA RTX 6000 GPU (FP32, batch size 8); full tables are reported in Appendix H.4.

**Sub-quadratic memory growth with model width.** Sweeping $d_{model}$ from 64 to 256 (4×) at fixed sequence length, MIMOMamba's training memory grows only $\sim 1.7\times$ (parallel variant) and $\sim 2.8\times$ (memory-optimized variant), compared with $\sim 3.9\times$ for Mamba-2 and $\sim 3.8\times$ for the Transformer—both tracking the near-quadratic scaling expected from independent dense projections.

**Linear inference memory scaling with sequence length.** At fixed $d_{model} = 64$, sweeping $L$ from 256 to 4096 (16×), MIMOMamba-Memory's peak inference memory grows from 27 MB to 155 MB ($\sim 5.7\times$), confirming the $\mathcal{O}(L)$ recurrent-state footprint. The Transformer grows from 44 MB to 8846 MB ($\sim 200\times$), reflecting attention's quadratic dependence in $L$.

**Matched-implementation throughput.** To enable a controlled comparison at the algorithm level, we evaluate both SSMs under a pure-PyTorch implementation without custom CUDA kernels; in particular, we reimplement Mamba-2 in pure PyTorch since its official fused kernels have limited portability across hardware/software configurations. Under this matched setting, MIMOMamba-Fast is 1.5–1.6× faster than Mamba-2 in training throughput across all measured sequence lengths. The Transformer baseline is built on PyTorch's built-in `nn.TransformerEncoder`, whose internally fused attention kernels provide kernel-level optimizations that neither SSM enjoys in our setup; developing custom CUDA fusion for MIMOMamba is left to future work.

## 7. Empirical Validation

This section empirically evaluates the practical implications of our matrix polynomial parameterization (Section 4) on a physics-informed forecasting benchmark. We focus on two questions: (1) can a strongly shared algebraic structure retain predictive accuracy with substantially fewer parameters than a standard Transformer? and (2) does this structure act as an inductive bias for learning cross-depth coupled dynamics, as reflected by faster/more stable optimization and improved errors in strongly coupled regions?

### 7.1. Setup

We consider the Internal Wave Sound Speed Profile (SSP) prediction task (Huang et al., 2024), which provides an ideal physics-informed benchmark. The underlying dynamics are governed by partial differential equations, where perturbations at adjacent depth layers are strictly coupled through buoyancy and stratification effects. Hence, modeling these vertical correlations exhibits strong cross-depth coupling, making it a natural testbed for evaluating whether architectures with explicit cross-channel (matrix-valued) interactions can provide practical benefits on cross-depth coupled dynamics. We therefore focus our empirical evaluation on predictive accuracy and parameter efficiency on this benchmark.

Following the simulation protocol of the Shallow Water 2006 experiment (Lin et al., 2010), we generate sequences of sound speed profiles across 80 depth layers at 30-second intervals for 300 timesteps, and cast the task as many-to-one forecasting: a 10-step history predicts the full profile at the next timestep. Our MIMOMamba uses a lightweight 2-layer configuration with degree-4 polynomial parameterization; implementation details for all models (e.g., hidden size, number of heads, and random seed) are provided in the Appendix G.4.

### 7.2. Results

As summarized in Table 1, MIMOMamba outperforms both baselines with the lowest RMSE$_{mean}$ (0.688 m/s) and the highest $R^2$ (0.481) despite having only $\approx$35k parameters,

*Table 1.* **SSP Prediction Results.** Comparing parameter efficiency and error rates after 50 epochs of training.

| MODEL | PARAMS | RMSE$_{\text{MEAN}}$ ↓ | RMSE$_{\text{MAX}}$ ↓ | $R^2$ ↑ |
|---|---|---|---|---|
| LSTM | 50,497 | 0.999 | 2.993 | 0.434 |
| TRANSFORMER | 100,801 | 0.748 | 1.926 | 0.467 |
| **MIMOMAMBA** | **35,593** | **0.688** | **1.662** | **0.481** |

significantly outperforming both baselines in accuracy while using 65% fewer parameters than the Transformer. This result corroborates our theoretical analysis that the matrix polynomial parameterization successfully compresses effective degrees of freedom without degrading predictive accuracy on this benchmark. Moreover, Figure 3 shows that MIMOMamba exhibits a steeper initial descent and stabilizes at a lower loss floor compared to the Transformer, whose more gradual learning trajectory suggests a less favorable optimization landscape on this task. This behavior is consistent with the shared algebraic basis, providing an inductive bias aligned with the underlying coupled dynamics.

### 7.3. Additional Results

We also conducted three targeted analyses detailed in Appendix G.4 to probe why MIMOMamba succeeds where the baselines struggle. First, we examined thermocline dynamics, which is the most challenging region at approximately 30 m depth where sound speed gradients are steepest due to temperature stratification, and found that MIMOMamba reduces worst-case error (RMSE$_{\text{max}}$) by 14% over the Transformer. Second, our convergence analysis shows that MIMOMamba consistently maintains higher $R^2$ throughout training, indicating faster capture of multi-scale wave interactions that the Transformer's independent per-timestep projections struggle to learn. Collectively, these additional analyses demonstrate that MIMOMamba's advantage stems not merely from parameter-count reduction but from its algebraic structure aligning with the physical constraints of coupled dynamical systems, which provide both computational efficiency and a physically meaningful inductive bias that enables accurate modeling of cross-dimensional interactions at a fraction of the parameter budget. Additional multi-seed robustness results are provided in Appendix G.5.

### 7.4. Extended Empirical Studies

To complement the SSP analyses above, we conducted four additional empirical studies, summarized briefly here; full protocols, tables, and analyses are deferred to Appendix H.

**Generalization to a UEA classification benchmark.** We evaluate MIMOMamba on SCP1 from the UEA multivariate time-series classification archive, following the protocol of Rusch & Rus (2025). Averaged over five random seeds,

MIMOMamba attains $85.5 \pm 2.9\%$ accuracy, outperforming all SISO baselines reported under this protocol (Mamba, S6, LRU, NRDE, NCDE) and narrowing the gap to continuous-time models such as LinOSS and S5 (Appendix H.1).

**Extended SSM baselines on SSP.** Under the same SSP protocol as in Section 7, we additionally compare against four recent SSM variants: Mamba-2 (Dao & Gu, 2024), Mamba-3 (Lahoti et al., 2026), Gated DeltaNet (Yang et al., 2025), and PD-SSM (Terzic et al., 2025). MIMOMamba achieves the best RMSE$_{\text{mean}}$ of 0.687, ahead of Gated DeltaNet (0.699), Mamba-3 (0.715), Mamba-2 (0.717), and PD-SSM (0.774); see Appendix H.2 for the full results.

**Polynomial-degree ablation.** An ablation over polynomial degrees $p \in \{2, \ldots, 8\}$ on the SSP benchmark indicates that $p = 4$ is the sweet spot for our default rank-4 low-rank base matrix configuration, with diminishing returns at higher degrees (Appendix H.3).

**Memory and throughput profiling.** Wall-clock measurements on an RTX 6000 GPU empirically confirm the efficiency claims of Section 6; see Section 6.3 for a summary and Appendix H.4 for the full tables.

**Expressivity–commutativity tradeoff.** The matrix polynomial parameterization realizes a deliberate tradeoff between expressivity and the commutativity required for the SSD duality. The multi-head design partially mitigates this restriction via a direct sum of commutative subalgebras; we discuss this tradeoff in detail in Appendix H.5.

## 8. Conclusion

We presented **MIMOMamba**, a MIMO selective SSM that extends State Space Duality via a matrix polynomial parameterization enforcing commutativity. This enables matrix-valued (cross-channel) interactions inside the recurrence while preserving the dual recurrent/parallel computation views. Viewed as an algebraic parameter-sharing scheme, it significantly reduces redundancy compared to standard attention, and our SSP experiments indicate that strong accuracy can be retained under a much smaller parameter budget. Optimizing kernel-level throughput and end-to-end benchmarking remain important directions for future work.

## Acknowledgements

This work was supported in part by the National Key R&D Program of China under 2024YFA1014202 and in part by the NSFC under 62271433 and in part by the Shenzhen Science and Technology Program under RCJC20210609104448114. The corresponding author is Feng Yin.

## Impact Statement

This paper presents work whose goal is to advance the field of Machine Learning. There are many potential societal consequences of our work, none which we feel must be specifically highlighted here.

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

# Appendix Overview

The appendices provide detailed derivations, implementation specifics, and extended experimental analysis to support the main text. To assist readers in navigating the supplementary material, we organize the content as follows:

**Part I: Algorithms and Hardware Efficiency**

- **Appendix A (Background):** Reviews the evolution from SISO to MIMO SSMs and contextualizes our hardware-aware design within the HBM/SRAM memory hierarchy constraints.

- **Appendix B (Fast Algorithms):** Details the FFT-based acceleration for time-varying convolution and discusses I/O-aware strategies for distributed training.

- **Appendix C (Implementation):** Provides visual expansions of the block-matrix duality forms and presents the complete pseudocode for MIMOMamba (Algorithm 1).

**Part II: Theoretical Foundations and Proofs**

- **Appendix D (Positional Analysis):** Discusses the endogenous ordering of SSMs versus the permutation equivariance of Attention, providing a theoretical justification for omitting explicit position embeddings.

- **Appendix E (Algebraic Structure):** Establishes the matrix polynomial parameterization, detailing the isomorphism between matrix rings and scalar polynomials that enables efficient computation via Cayley-Hamilton reduction.

- **Appendix F (Formal Proofs):** Contains rigorous proofs for the propositions in the main text, including the block semiseparability of the operator (Prop C.2) and the optimality of non-derogatory base matrices via the Centralizer Theorem.

**Part III: Empirical Details**

- **Appendix G (Physics Simulation):** Describes the Internal Wave Sound Speed Profile (SSP) benchmark, including the governing physical equations, data generation pipeline, and detailed hyperparameter configurations.

- **Appendix H (Extended Empirical Evaluation):** Reports (i) generalization on the SCP1 classification benchmark from the UEA archive, (ii) extended SSP baselines including Mamba-2, Mamba-3, Gated DeltaNet and PD-SSM, (iii) polynomial-degree ablation, (iv) memory/speed scaling on GPU across sequence lengths and model widths, and (v) a discussion of the expressivity–commutativity tradeoff.

## A. Detailed Background and Related Work

This appendix provides comprehensive background context for the architectural decisions behind MIMOMamba. We first review the algebraic principles of linear attention and the hardware-aware computational constraints (e.g., the HBM-SRAM memory hierarchy) that motivate our efficiency optimizations. Subsequently, we analyze the historical evolution of Structured State-Space Models (SSMs) along two critical axes: the structural tension between SISO and MIMO formulations, and the dynamical transition from time-invariant to selective discretization.

### A.1. Linear Attention: Reordering Computation via Associativity

The core efficiency principle behind linear attention mechanisms is the **associative property** of matrix multiplication. As illustrated in Figure 4, standard softmax attention is bottlenecked by the explicit computation of the $L \times L$ matrix $(QK^\top)$. The non-linear softmax function prevents a reordering of this computation.

Linear attention variants (Katharopoulos et al., 2020; Choromanski et al., 2021) address this by replacing softmax with a linear kernel function $\phi(\cdot)$. This allows the computation to be rearranged from $(QK^\top)V$ to $Q(K^\top V)$. Since $K^\top V$ results in a small, fixed-size $D \times D$ matrix, this reordering avoids materializing the large intermediate matrix and reduces the overall complexity to be linear in the sequence length, $L$. This algebraic maneuver is foundational to many efficient Transformer alternatives, but it faces challenges in causal settings where masking operations can disrupt the direct application of associativity.

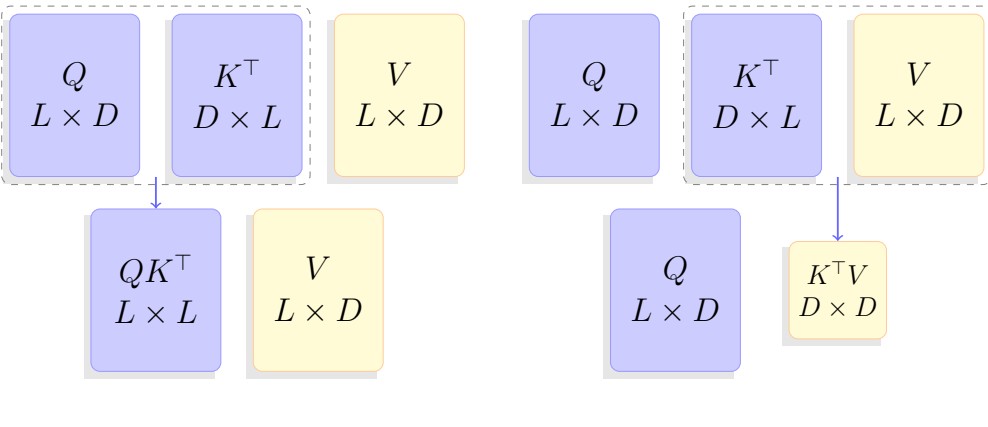

Softmax Attention $\mathcal{O}(L^2 D)$     Linear Attention $\mathcal{O}(L D^2)$

*Figure 4.* Softmax Attention vs. Linear Attention. **Left**: Softmax attention's quadratic complexity arises from computing the large $L \times L$ matrix $QK^\top$. **Right**: Linear attention leverages the associative property to first compute the smaller $D \times D$ matrix $K^\top V$, achieving linear complexity.

## A.2. Hardware-Aware Computation: The Memory Hierarchy Perspective

To fully appreciate the computational efficiency discussion in Section 6.2, it is essential to understand the modern GPU memory hierarchy, particularly the distinction between **High Bandwidth Memory (HBM)** and **Static Random Access Memory (SRAM)**.

**The Memory Bottleneck.**   Modern GPUs (e.g., NVIDIA A100/H100) are characterized by a massive disparity between compute capability (FLOPs) and memory bandwidth. The memory hierarchy roughly consists of:

- **HBM (Global Memory):** Large capacity (e.g., 40–80 GB) but relatively limited bandwidth (approx. 1.5–3 TB/s). This is where standard model inputs, outputs, and weights reside.

- **SRAM (Shared Memory):** Extremely limited capacity (e.g., 192 KB per Streaming Multiprocessor, totaling $\approx$ 20–50 MB per GPU) but significantly higher bandwidth (approx. 19 TB/s on A100).

In large-scale deep learning, operations are often *IO-bound* rather than compute-bound. The total runtime is dominated by the cost of moving data between HBM and SRAM, rather than the arithmetic operations themselves (the "Memory Wall"). Therefore, a key design principle for efficient algorithms is to **minimize HBM accesses** by keeping intermediate results "resident" in SRAM.

**Revisiting FlashAttention and Mamba.**   Recent advancements in efficient sequence modeling heavily leverage this hierarchy optimization:

- **FlashAttention (Dao et al., 2022b):** Standard attention mechanisms require materializing an $L \times L$ attention matrix to HBM to compute the softmax, incurring quadratic memory IO. FlashAttention employs *Tiling* techniques to load blocks of Queries, Keys, and Values into SRAM, compute the attention scores, and perform the softmax normalization locally within SRAM. The massive intermediate attention matrix is never fully materialized in HBM, resulting in linear IO complexity.

- **Mamba (Gu & Dao, 2023):** Similarly, Selective State Space Models involve a recurrent state expansion. A naive implementation would require reading and writing the high-dimensional latent states $h_t$ to HBM at every timestep. Hardware-aware SSM implementations perform the *Parallel Scan* or recurrence strictly within SRAM. The inputs are

loaded from HBM, the state expansion and update occur in the high-bandwidth SRAM, and only the projected outputs are written back to HBM.

**Implications for MIMOMamba.** Our discussion in Section 6.2 regarding "SRAM-Resident Algebra" builds directly upon these principles. By parameterizing the MIMO dynamics via matrix polynomials (represented by low-rank scalar coefficients), we allow the system to perform the heavy matrix algebra (polynomial evaluation and state mixing) within the fast SRAM, communicating only the lightweight coefficients via HBM. This avoids the prohibitive cost of broadcasting dense time-varying transition matrices, effectively mitigating the memory bandwidth bottleneck for multi-dimensional state dynamics.

### A.3. The Second Axis of SSM Evolution: From Static to Dynamic Discretization

Beyond the SISO-MIMO distinction, a second, equally important axis of evolution in SSMs is the transition from **time-invariant** to **time-varying (or selective)** dynamics. This transition is best understood through the lens of how the underlying continuous-time systems are discretized.

**Continuous-Time Foundation and Discretization** The theoretical foundation for modern SSMs is a continuous-time linear system. To distinguish it from the discrete systems used throughout the rest of this paper, we denote its state matrices with a bar:

$$\dot{h}(t) = \bar{A}h(t) + \bar{B}x(t). \tag{3}$$

To be implemented on digital hardware, this system must be discretized into a sequence-to-sequence recurrence using a step size $\Delta$. This yields the discrete-time matrices, which we denote without a bar to align with the notation in the main text:

$$h_k = A_k h_{k-1} + B_k x_k. \tag{4}$$

The following derivation shows how the discrete matrices $A_k$ and $B_k$ are derived from the continuous matrices $\bar{A}$, $\bar{B}$ and the step size $\Delta$.

**Illustrative Derivation: The Zero-Order Hold (ZOH)** The exact solution to the continuous-time ODE(3) over an interval $[t_0, t]$ is given by the variation of constants formula (Hartman, 2002):

$$h(t) = e^{\bar{A}(t-t_0)}h(t_0) + \int_{t_0}^{t} e^{\bar{A}(t-\tau)}\bar{B}x(\tau)d\tau.$$

A canonical method for discretization is the Zero-Order Hold (Franklin et al., 1998), which assumes the input $x(\tau)$ is held constant at the value $x_k$ over the interval $[t_k, t_{k+1}]$, where $\Delta = t_{k+1} - t_k$. Applying this to the integral solution yields:

$$h(t_{k+1}) = e^{\bar{A}\Delta}h(t_k) + \left( \int_{t_k}^{t_{k+1}} e^{\bar{A}(t_{k+1}-\tau)}\bar{B}d\tau \right) x_k,$$

$$h_{k+1} = e^{\bar{A}\Delta}h_k + \left( \int_{0}^{\Delta} e^{\bar{A}s}\bar{B}ds \right) x_k.$$

Solving the integral yields the discrete-time system matrices:

$$A_k = e^{\Delta\bar{A}},$$

$$B_k = \left( \int_{0}^{\Delta} e^{\bar{A}s}ds \right) \bar{B}.$$

**Assuming $\bar{A}$ is invertible** (non-singular)—a condition typically satisfied in structured SSM initializations—the integral for $B_k$ admits the closed-form solution referenced in the literature:

$$B_k = \bar{A}^{-1}(e^{\Delta\bar{A}} - I)\bar{B}.$$

While ZOH is a common and illustrative method, other discretization techniques, such as the bilinear transform (Franklin et al., 1998), are used in practice. The crucial insight—that a dynamic $\Delta_t$ makes the system selective—is independent of the specific discretization rule chosen.

*Table 2.* Comparison of Time-Invariant and Selective SSM Discretization.

| Component | Time-Invariant SSM (S4/S5) | Selective SSM (Mamba) |
|---|---|---|
| Step Size | Fixed parameter: $\Delta$ | Input-dependent: $\Delta_t = s_\Delta(x_t)$ |
| Discrete Matrix $A_k$ | Static: $A = e^{\Delta\bar{A}}$ | Dynamic: $A_t = e^{\Delta_t\bar{A}}$ |
| Discrete Matrix $B_k$ | Static: $B = \bar{A}^{-1}(e^{\Delta\bar{A}} - I)\bar{B}$ | Dynamic: $B_t = \bar{A}^{-1}(e^{\Delta_t\bar{A}} - I)\bar{B}$ |

**Unifying the Two Axes**  This brings us to a crucial distinction: while S5 (Smith et al., 2023) was a MIMO model, it was fundamentally a time-invariant one with a fixed $\Delta$. Conversely, the original Mamba was a selective model with a dynamic $\Delta_t$, but was developed in a SISO context. Our work, MIMOMamba, operates at the confluence of these two advancements and defines a selective MIMO SSM that unifies the **cross-dimensional modeling** of S5 (Smith et al., 2023) with the content-aware, dynamic discretization of Mamba(Gu & Dao, 2023).

# B. Efficient Computation of Time-Varying Kernel Coefficients via FFT

This appendix details the algorithm for efficiently computing the coefficients of the time-varying convolution kernel in MIMOMamba. A crucial distinction must be made from time-invariant (LTI) models like S4 (Gu et al., 2022), which can compute a global, fixed convolution kernel in a single pass using FFT(Fu et al., 2023). As our model is selective and thus time-varying (LTV), we cannot use this approach directly. Instead, we leverage our polynomial parameterization to find a different path for computational acceleration.

## B.1. The Challenge: From Matrix Products to Polynomial Chains

The coefficients of the LTV kernel depend on prefix products of the selective state matrices, e.g., $\prod_{t=0}^{k} A_t$. A naive sequential multiplication of these $D \times D$ matrices would be prohibitively slow.

Our polynomial parameterization, introduced primarily to ensure commutativity, offers an elegant solution. Each matrix $A_t$ is a polynomial of a shared base matrix, $A_t = f_t(A_{\text{base}})$. This allows us to transform the problem of multiplying matrices into multiplying their corresponding scalar polynomials first:

$$\prod_{t=0}^{k} A_t = \prod_{t=0}^{k} f_t(A_{\text{base}}) = \left(\prod_{t=0}^{k} f_t\right)(A_{\text{base}}).$$

The core computational task thus becomes efficiently multiplying chains of scalar polynomials, such as $F_k(x) = f_0(x) \cdot f_1(x) \cdots f_k(x)$. This is accomplished by repeatedly using a highly efficient sub-routine for multiplying two polynomials.

## B.2. The Sub-routine: Two-Polynomial Multiplication via FFT

**Principle: Switching Representations for Efficient Multiplication**  A polynomial can be represented in two primary ways (Cormen et al., 2022). The first is the standard **coefficient representation**, $f(x) = \sum_{i=0}^{p} a_i x^i$. Multiplying two polynomials in this form requires computing the discrete convolution of their coefficients, an operation with $\mathcal{O}(p^2)$ complexity.

The second is the **point-value representation**, which uniquely defines a degree-$p$ polynomial by a set of $p + 1$ point-value pairs, $\{(x_0, y_0), (x_1, y_1), \ldots, (x_p, y_p)\}$, where $y_i = f(x_i)$. The key advantage is that multiplying two polynomials, $h(x) = f(x) \cdot g(x)$, in this representation is computationally trivial: if $f$ is represented by $\{(x_i, y_i)\}$ and $g$ by $\{(x_i, z_i)\}$, then $h$ is represented by $\{(x_i, y_i \cdot z_i)\}$. This is a simple element-wise product with $\mathcal{O}(p)$ complexity.

The FFT-based algorithm leverages this insight. It provides an extremely efficient method to convert a polynomial from its coefficient representation to a point-value representation (a process called **evaluation**), and back (a process called **interpolation**).

**The FFT as an Efficient Conversion Algorithm**  The Discrete Fourier Transform (DFT) is a mathematical operator that evaluates a polynomial at a specific set of points: the $N$-th roots of unity. The Fast Fourier Transform (FFT) (Cooley & Tukey, 1965) is a highly optimized algorithm that computes the DFT in $\mathcal{O}(N \log N)$ time. The Inverse FFT (IFFT)

efficiently performs the reverse interpolation process.

**Algorithmic Steps**   The algorithm for computing $h(x) = f(x) \cdot g(x)$ therefore consists of three stages, framed as a "change-multiply-change back" process:

1. **Evaluation via Forward FFT**: Convert the polynomials $f(x)$ and $g(x)$ from their coefficient representation to their point-value representation. This is done by applying the FFT to their (zero-padded) coefficient vectors $a$ and $b$, which produces their values at the roots of unity: $\hat{f} = \text{FFT}(\text{pad}(a))$, $\hat{g} = \text{FFT}(\text{pad}(b))$.

2. **Point-wise Product in the Value Domain**: Perform the cheap multiplication in the point-value domain: $\hat{h}_k = \hat{f}_k \cdot \hat{g}_k$.

3. **Interpolation via Inverse FFT**: Convert the resulting point-value representation of $h(x)$ back to its coefficient representation. This is done by applying the IFFT to the vector $\hat{h}$: $c = \text{IFFT}(\hat{h})$.

**A Note on Zero-Padding for Correctness**   The DFT and FFT mathematically compute a *circular* convolution, where indices "wrap around". However, standard polynomial multiplication corresponds to a *linear* convolution. To ensure a correct result, we must use zero-padding. The product of two degree-$p$ polynomials has a degree of $2p$, requiring $2p + 1$ coefficients. By zero-padding the input coefficient vectors to a length $N_{\text{fft}} \geq 2p + 1$, we guarantee that the wrap-around part of the circular convolution only involves zeros, which ensures that the result is identical to that of a linear convolution.

**Sub-routine Complexity**   The complexity of this two-polynomial multiplication sub-routine is dominated by the FFT/IFFT operations, as shown in Table 3. Given that $N_{\text{fft}}$ is on the order of $\mathcal{O}(p)$, this method achieves a near-linear time complexity.

*Table 3.* Complexity for multiplying two degree-$p$ polynomials.

| Step | Direct Convolution | FFT-based Method |
|------|:---:|:---:|
| Transformation | N/A | $\mathcal{O}(N_{\text{fft}} \log N_{\text{fft}})$ |
| Multiplication | $\mathcal{O}(p^2)$ | $\mathcal{O}(N_{\text{fft}})$ |
| Inverse Transform | N/A | $\mathcal{O}(N_{\text{fft}} \log N_{\text{fft}})$ |
| **Total Complexity** | $\mathcal{O}(p^2)$ | $\mathcal{O}(p \log p)$ |

### B.3. Application to Computing the Full Polynomial Chain

To compute the coefficients for the entire time-varying kernel, we must compute the chain of polynomial products $F_k(x) = \prod_{t=0}^{k} f_t(x)$ for $k = 0, \ldots, L-1$. This is done by repeatedly applying the FFT-based sub-routine. A straightforward sequential approach is:

- Initialize $F_0(x) = f_0(x)$.

- For $k = 1, \ldots, L-1$, compute $F_k(x) = F_{k-1}(x) \cdot f_k(x)$ using the sub-routine.

This sequential method requires $L - 1$ calls to the multiplication sub-routine. A more parallelizable approach is to use a "divide and conquer" or tree-based reduction structure. This reduces the number of **sequential multiplication steps** to $\mathcal{O}(\log L)$, which makes the computation highly efficient on parallel hardware, although the total amount of work remains proportional to $L$.

*Remark* B.1. The critical insight remains: the FFT is used here not as a global signal processing tool, but as a purely algebraic shortcut to accelerate the multiplication of polynomial coefficient vectors, a task made necessary and possible by our model's selective, LTV nature.

### B.4. Multi-Head MIMO Architecture

A key design consideration in sequence models is the *state expansion ratio*, namely the ratio between the latent state dimension $N$ in the SSM and the embedding dimension $D$ of the input sequence. As discussed in prior work (Dao & Gu, 2024),traditional RNNs typically operate with $N = 1$ (no expansion) or adopt a MIMO formulation with dense $B_t$ and $C_t$

matrices, both of which constrain $N$ to be comparable to or smaller than $D$. This restriction limits the model's capacity to store and process rich temporal information, particularly in high-entropy domains such as language (Dao & Gu, 2024; Gu et al., 2022). Empirical evidence indicates that larger state expansion factors ($N \gg D$) are essential for modeling complex dependencies in information-dense sequences, which allows effective compression of long histories into expressive latent representations (Gu et al., 2020; Smith et al., 2023).

In our proposed MIMOMamba architecture, the latent state $h_t^{(k)} \in \mathbb{R}^D$ of each head matches the input dimension $D$ for parameter efficiency. However, this design choice precludes a single MIMO SSM from leveraging a larger expansion ratio $N \gg D$, potentially limiting its ability to capture diverse temporal patterns. To overcome this limitation and enhance representational capacity, we employ a **multi-head** design inspired by multi-head attention in Transformers (Vaswani et al., 2017). By instantiating $H$ parallel MIMO SSM heads, we achieve an effective joint latent state dimension of $H \cdot D$, which enables state expansion without inflating per-head computational costs. This architecture facilitates a division of labor where distinct heads isolate specific sequence properties, including varying frequency bands, timescales, or channel dependencies. Such a design effectively replicates the specialization found in multi-head attention (Vaswani et al., 2017; Michel et al., 2019) and mixture-of-experts models (Shazeer et al., 2017).

Formally, we assume $H$ independent MIMO SSM heads, where each head $k$ learns its own set of selective state matrices: $(A_t^{(k)}, B_t^{(k)}, C_t^{(k)}), \quad k = 1, \ldots, H$.

Each head independently processes the input sequence, producing an output sequence $Y^{(k)}$ according to the MIMO recurrence and dual attention form described earlier. The outputs are concatenated along the channel dimension and processed through an integration stage: $Y_{\text{out}} = \text{Merge}\left([Y^{(1)}, Y^{(2)}, \ldots, Y^{(H)}]\right)$. where $\text{Merge}(\cdot)$ is a fusion module, which is typically a linear projection(LeCun et al., 2015)

While the fusion module can effectively integrate cross-head information through linear projections and non-linear activations, a natural extension for enhanced expressiveness would be to incorporate a *mixture-of-experts* (MoE) layer within the merge stage (Shazeer et al., 2017; Anthony et al., 2024). **We emphasize that MoE is not part of the current MIMOMamba architecture; all experiments in this paper use a standard linear projection as the merge function, with no MoE routing or gating involved.** We discuss MoE here as a promising future direction. In the MoE framework, multiple expert sub-networks are adopted, with a lightweight gating function dynamically routing each token to a sparse subset of experts (typically top-$k$) for computational efficiency (Shazeer et al., 2017; Fedus et al., 2022). This conditional computation paradigm significantly increases model capacity while maintaining modest computational overhead, as only a fraction of parameters activate per input. Within our architecture, the MoE merge stage could learn to allocate specialized experts to process representations from specific SSM heads, promoting diversity and enabling task-specific adaptations (Anthony et al., 2024).

In summary, this multi-head MIMO architecture expands the effective latent state dimension from $D$ to $H \cdot D$, in order to achieve **state expansion** without compromising per-head efficiency. While this paper employs a standard linear projection for the merge function, a MoE-based integration stage represents a promising extension that could further scale capacity and exploit head diversity. This design bridges expressive MIMO dynamics with the practical benefits of state expansion, and it supports the development of more powerful sequence models.

### B.5. Parallel State Expansion via Direct Sum of Algebras

While the polynomial ring $\mathbb{R}[A]$ provides a complete commutative structure, relying on a single base matrix limits the latent state capacity to the embedding dimension $D$. In sequence modeling, maximizing the effective state dimension $N$ is crucial for capturing complex temporal dependencies (Gu, 2023).

To address this, we adopt a **Direct Sum of Algebras** framework. It is worth noting that standard SISO SSMs can be viewed as a degenerate case of this framework where the system is decomposed into $D$ independent scalar subspaces (i.e., $H = D$ with $1 \times 1$ blocks). While computationally efficient, this scalar-diagonal structure restricts the interaction within the state dynamics.

We propose a **Parallel Multi-Head Formulation** that extends this generalization from scalars to full matrices. Instead of partitioning the feature dimension, we instantiate $H$ parallel, independent algebraic systems operating on the full feature dimension. We define the global base operator $\mathbf{A}$ as a block-diagonal matrix: $\mathbf{A} = \text{diag}(A^{(1)}, A^{(2)}, \ldots, A^{(H)})$, where each block $A^{(k)} \in \mathbb{R}^{D \times D}$ is defined strictly over the full embedding dimension $D$. This dimension alignment is essential to ensure

that all time-varying operators (e.g., $A_t^{(k)}, B_t^{(k)}$) remain square matrices, thereby strictly preserving the commutative ring structure $\mathbb{R}[A^{(k)}]$ within each head. The total algebraic structure corresponds to the direct sum: $\mathcal{A}_{total} = \bigoplus_{k=1}^{H} \mathbb{R}[A^{(k)}]$.

In this framework, the input $x_t \in \mathbb{R}^D$ is projected/shared across all algebraic subspaces. Following the general SSM formulation, the state evolution for the $k$-th head is instantiated as:

$$h_t^{(k)} = A_t^{(k)} h_{t-1}^{(k)} + B_t^{(k)} x_t, \quad \text{with } A_t^{(k)}, B_t^{(k)} \in \mathbb{R}[A^{(k)}].$$

Crucially, operators from different heads have **disjoint support**. This enforces a strict block-diagonal structure where commutativity is rigorously maintained *within* each head, providing the mathematical guarantee required for the duality derivation in the next section.

**Remark on Positional Information.** A critical advantage of this matrix-recurrence formulation is its strong inductive bias for sequential order. The state evolution within each head is governed by non-commutative matrix multiplication chains (e.g., $\prod A_t$) that inherently encode the temporal structure of the data. This leads to a distinct architectural difference from concurrent approaches. While Mamba-3 (Lahoti et al., 2026) relies on complex-valued parameterization to functionally implement RoPE (Su et al., 2024), our theoretical analysis indicates that such auxiliary encodings are *structural redundancies* in our framework. The endogenous order induced by the matrix polynomial structure is theoretically sufficient for modeling temporal dependencies (formal proof provided in Appendix D).

## B.6. Computational Efficiency from Polynomial Structure

The polynomial parameterization, while introduced primarily for the theoretical purpose of guaranteeing commutativity for the MIMO duality, confers a crucial secondary advantage: it serves as the foundation for a highly efficient computational strategy. This algebraic structure unlocks optimizations on computational load (FLOPs) and communication overhead (I/O).

This approach places our work firmly within a successful lineage of **hardware-aware algorithm design**, where computational structures are co-designed to overcome bottlenecks in modern hardware hierarchies. Seminal works from this lineage include FlashAttention (Dao et al., 2022b), which reorders the attention computation to be optimal for the GPU memory hierarchy, and Monarch matrices (Dao et al., 2022a), which replace dense operators with hardware-efficient structured alternatives.

Analogously, our polynomial parameterization can enable two key optimizations. For FLOPs, it lifts expensive matrix algebra into the domain of fast symbolic polynomial arithmetic via FFT. For I/O, it provides a compact representation that drastically reduces communication costs in distributed settings.

### B.6.1. EFFICIENCY VIA ISOMORPHIC POLYNOMIAL ARITHMETIC

A critical challenge in training time-varying models is the computational cost of prefix products of state matrices, i.e., $A_{t:0}^{(k)} = \prod_{i=0}^{T} A_i^{(k)}$. Standard recursive matrix multiplication incurs a prohibitive complexity of $\mathcal{O}(TD^3)$. We bypass this bottleneck by exploiting the commutative algebraic structure established in Sec. B.5, which guarantees an isomorphism between the matrix subalgebra and the scalar polynomial ring.

This allows us to replace expensive matrix operations with efficient polynomial arithmetic via a two-step procedure:

1. **Mapping to Polynomial Domain:** Instead of multiplying dense matrices directly, we map each matrix $A_t \in \mathbb{R}[A]$ to its corresponding scalar polynomial $p_t(x)$. Due to the ring isomorphism, the matrix product corresponds exactly to the polynomial product:
$$\psi(A_t \cdot A_{t-1}) = p_t(x) \cdot p_{t-1}(x) \pmod{\mu(x)} \tag{5}$$
This transforms the problem from non-commutative matrix algebra (in the general case) to commutative scalar algebra.

2. **Fast Convolution via FFT:** The product of polynomials is computationally equivalent to the convolution of their coefficients. We leverage the Fast Fourier Transform (FFT) to compute these convolutions in log-linear time $\mathcal{O}(D \log D)$, as opposed to the cubic cost of matrix multiplication.

After computing the aggregate polynomial in the frequency domain, we evaluate it on the base matrix to recover the final state. This algebraic reduction decreases the theoretical complexity of the recurrence step from $\mathcal{O}(TD^3)$ to $\mathcal{O}(TD \log D)$, making the training of our expressive LTV model parallelizable and computationally feasible.

As highlighted by recent work on I/O-aware algorithms like FlashAttention (Dao et al., 2022b; Dao, 2023) , the primary bottleneck in modern accelerators is often not raw compute but memory access and inter-device communication. This is crucial for multi-GPU training, where bandwidth across interfaces like NVLink can limit overall throughput (Gholami et al., 2024).

Our polynomial parameterization offers a distinct advantage in this I/O-bound regime. A naive distribution of our model would require broadcasting the full, dense selective matrices $(A_t^{(k)}, B_t^{(k)}, C_t^{(k)})$ from a primary GPU to all others at every time step. This would incur a substantial communication payload of $\mathcal{O}(H \cdot D^2)$ per step for $H$ heads, creating a significant communication bottleneck.

Instead, our approach enables an I/O-aware strategy. The large, time-invariant base matrices $\{A^{(k)}\}$ are broadcast only once at the start of training. Subsequently, at each time step, we only need to transmit the lightweight polynomial coefficients $\{\alpha_{t,j}^{(k)}, \beta_{t,j}^{(k)}, \gamma_{t,j}^{(k)}\}$ used to generate the selective matrices, as noted in Algorithm 1. This reduces the communication payload for each matrix from $\mathcal{O}(D^2)$ to just $\mathcal{O}(p)$ per head, where the polynomial degree $p \ll D^2$.

Our method compresses the model update representation itself and demonstrates the broad applicability of this I/O-first principle.

By minimizing data transfer, we drastically reduce the inter-GPU communication pressure, which allows the GPU compute cores to remain highly utilized instead of stalling while waiting for data. This directly translates to improved wall-clock speed and more efficient scaling during large-scale model training.

## C. Expanded Block Matrix Formulation of the MIMO Duality

This section provides a detailed, expanded view of the block matrix structures central to our proposed MIMO state space duality. The aim is to offer a more intuitive visualization of the global transformation matrix and its factorization into the dual attention form. We focus on a single head $k$ for clarity; in the full model, these computations occur in parallel for all heads.

### C.1. The Global Transformation Matrix $\mathbf{M}^{(k)}$

The complete input-to-output transformation for the $k$-th head, $\mathbf{Y}^{(\mathbf{k})} = \mathbf{M}^{(k)} \rhd \mathbf{X}$, is governed by the block-lower-triangular matrix $\mathbf{M}^{(k)}$ depicted in Eq. (6).

$$
\mathbf{M}^{(k)} = \begin{pmatrix}
C_0^{(k)} B_0^{(k)} & & & & \\
C_1^{(k)} A_1^{(k)} B_0^{(k)} & C_1^{(k)} B_1^{(k)} & & & \\
C_2^{(k)} A_2^{(k)} A_1^{(k)} B_0^{(k)} & C_2^{(k)} A_2^{(k)} B_1^{(k)} & C_2^{(k)} B_2^{(k)} & & \\
\vdots & \vdots & \vdots & \ddots & \\
C_{L-1}^{(k)} A_{L-1:1}^{(k)} B_0^{(k)} & C_{L-1}^{(k)} A_{L-1:2}^{(k)} B_1^{(k)} & \cdots & \cdots & C_{L-1}^{(k)} B_{L-1}^{(k)}
\end{pmatrix} . \quad (6)
$$

Here, $\mathbf{M}^{(k)}$ is an $L \times L$ block matrix with block indices $(i, j)$ over timesteps, and each entry $\mathbf{M}_{ij}^{(k)}$ is a $D \times D$ matrix acting on the feature dimension. Equivalently, $\mathbf{M}^{(k)} \in \mathbb{R}^{(LD) \times (LD)}$.

This structure visualizes the causal nature of the SSM recurrence. The diagonal blocks, e.g., $C_i^{(k)} B_i^{(k)}$, represent the direct influence of the input vector $x_i$ on the output vector $y_i$. The off-diagonal blocks, such as $C_i^{(k)} A_{i:j+1}^{(k)} B_j^{(k)}$, capture the influence of a past input $x_j$ on the current output $y_i$, also serving as an effect that is modulated by the sequence of state transition matrices $A_t^{(k)}$ between them.

## C.2. Structure of the Block Mask Matrix

The factorization derived in Eq. (2) decomposes the global operation into two distinct components. The first is the block mask matrix $\mathbf{L}^{(k)}$, which captures the state transition dynamics and acts as a structured, matrix-valued causal mask.

Expanding its structure, $\mathbf{L}^{(k)}$ takes the form of a block lower-triangular matrix:

$$\mathbf{L}^{(k)} = \begin{pmatrix} I & 0 & \cdots & \cdots & 0 \\ A_1^{(k)} & I & 0 & \cdots & 0 \\ A_2^{(k)}A_1^{(k)} & A_2^{(k)} & I & \cdots & \vdots \\ \vdots & \vdots & \vdots & \ddots & 0 \\ A_{L-1:1}^{(k)} & A_{L-1:2}^{(k)} & \cdots & A_{L-1}^{(k)} & I \end{pmatrix}. \tag{7}$$

As shown in Eq. (7), the identity blocks on the diagonal represent the instantaneous pass-through of information. The strictly lower off-diagonal blocks are products of transition matrices $A_{i:j+1}^{(k)}$. Unlike standard attention masks which are typically binary (0 or $-\infty$) or scalar-valued decay, our mask is matrix-valued. Each block applies a specific linear transformation that rotates and mixes the feature dimensions as the state evolves over time, and it strictly respects causality.

## C.3. The Block Query-Key Interaction

The second component of the duality is the block query-key matrix, which models the content-based interactions between input steps. This term is formed by the outer product of the stacked query matrices $\mathbf{Q}^{(k)}$ and the concatenated key matrices $\mathbf{K}^{(k)}$:

$$\mathbf{Q}^{(k)}(\mathbf{K}^{(k)})^\top = \begin{bmatrix} C_0^{(k)} \\ C_1^{(k)} \\ \vdots \\ C_{L-1}^{(k)} \end{bmatrix} \begin{bmatrix} B_0^{(k)} & B_1^{(k)} & \cdots & B_{L-1}^{(k)} \end{bmatrix}. \tag{8}$$

This operation results in a dense block matrix where the $(i, j)$-th block is the matrix product $C_{i-1}^{(k)}B_{j-1}^{(k)}$. Analogous to the standard dot-product attention which computes scalar similarities, this formulation computes a $D \times D$ interaction matrix for every pair of time steps. This dense interaction map represents the global receptive field of the model before being modulated by the causal decay of the mask $\mathbf{L}^{(k)}$.

Since elements of the same polynomial ring commute, we can rigorously reorder the terms to separate the state dynam- ics from the input-output projections:

$$\mathbf{M}_{ij}^{(k)} = \underbrace{(A_{i-1}^{(k)} A_{i-2}^{(k)} \cdots A_j^{(k)})}_{\text{State Dynamics}} \underbrace{(C_{i-1}^{(k)} B_{j-1}^{(k)})}_{\text{Input-Output Projection}}. \tag{9}$$

Finally, we can obtain:

$$\mathbf{M}^{(k)} = \mathbf{L}^{(k)} \circledast (\mathbf{Q}^{(k)}(\mathbf{K}^{(k)})^\top) \tag{10}$$

which is precisely the matrix-valued SSD shown in Figure 1.

## C.4. Generalization of Prior Art

In this section, we provide the formal justification referenced in Section 5 regarding the reduction of our framework to prior SISO formulations.

**Proposition C.1** (Reduction to Scalar Duality). *If the base matrix $A^{(k)}$ is restricted to a scalar multiple of the identity matrix, i.e., $A^{(k)} = \lambda I$, then the Matrix-Valued Dual Attention (Eq. 10) degenerates into $D$ independent SISO SSD mechanisms.*

*Proof.* If $A^{(k)} = \lambda I$, the polynomial ring $\mathbb{R}[A^{(k)}]$ reduces to the set of scalar matrices $\{cI \mid c \in \mathbb{R}\}$. Consequently, all selective matrices $A_t, B_t, C_t$ become diagonal with uniform entries. The block matrix product $C_i A_{i:j} B_j$ diagonalizes, and the cross-dimensional terms vanish. The system decouples into $D$ parallel channels, recovering the exact form of the structured masked attention in Mamba-2. □

## C.5. The Semiseparable Structure of the MIMO SSM

A key source of efficiency in modern SSMs is that their global operators can be represented as semiseparable matrices (Pernet & Storjohann, 2018; Dao & Gu, 2024). Our MIMO formulation preserves this property, and thus inherits the same computational advantages.

**Proposition C.2.** *The block transformation matrix* $\mathbf{M}^{(k)}$ *for each head is* **block** $D$**-semiseparable**.

*Proof sketch.* For any strictly lower off-diagonal block submatrix $\mathbf{M}^{(k)}_{j+1:j', \, i'+1:i}$ with $j' > j \geq i > i'$, the recurrence can be expanded to give an explicit factorization $\mathbf{M}^{(k)}_{j+1:j', \, i'+1:i} = \mathbf{UV}$ whose inner dimension is $D$. Hence $\operatorname{rank}(\mathbf{M}^{(k)}_{j+1:j', \, i'+1:i}) \leq D$, which is exactly block $D$-semiseparability. The full proof is deferred to Appendix F.1.

## C.6. Detailed Computational Algorithm

Algorithm 1 demonstrates the Forward Pass of MIMOMamba.

---

**Algorithm 1** Computational Duality of the MIMOMamba Forward Pass

---

**Require:** Input sequence $\mathbf{X} \in \mathbb{R}^{L \times D}$, number of heads $H$, polynomial degree $p$
**Ensure:** Output sequence $\mathbf{Y}_{\text{out}} \in \mathbb{R}^{L \times D}$
 1: **Given:** Base matrices $\{A^{(k)}\}_{k=1}^{H} \in \mathbb{R}^{D \times D}$ {Learned and fixed after training}
 2: **Given:** Polynomial coefficients $\{\alpha_{t,j}^{(k)}, \beta_{t,j}^{(k)}, \gamma_{t,j}^{(k)}\}$ {Lightweight coefs broadcastable across GPUs}
 3: **for** each head $k = 1$ to $H$ **do**
 4:     **for** $t = 0$ to $L - 1$ **do**
 5:         $A_t^{(k)} \leftarrow \sum_{j=0}^{p} \alpha_{t,j}^{(k)} (A^{(k)})^j$ {Matrix polynomial parameterization}
 6:         $B_t^{(k)} \leftarrow \sum_{j=0}^{p} \beta_{t,j}^{(k)} (A^{(k)})^j$
 7:         $C_t^{(k)} \leftarrow \sum_{j=0}^{p} \gamma_{t,j}^{(k)} (A^{(k)})^j$
     *{State Space Duality (SSD): Select path based on context (e.g., Training vs. Inference)}*
 8:     **if Parallel Mode** (Dual Attention View) **then** {Efficient for parallel training}
 9:         $\mathbf{Q}^{(k)} \leftarrow [C_0^{(k)\top}, \dots, C_{L-1}^{(k)\top}]^\top$
10:         $\mathbf{K}^{(k)} \leftarrow [B_0^{(k)}, \dots, B_{L-1}^{(k)}]^\top$
11:         Construct mask $\mathbf{L}^{(k)}$ where $L_{ij}^{(k)} = A_{i-1:j}^{(k)}$ if $i > j$, else $I$ or $0$.
12:         $\mathbf{Y}^{(\mathbf{k})} \leftarrow \left(\mathbf{L}^{(k)} \circledast (\mathbf{Q}^{(k)}(\mathbf{K}^{(k)})^\top)\right) \triangleright \mathbf{X}$ {Masked Attention}
13:     **else Recurrent Mode** (SSM View) {Efficient for auto-regressive inference}
14:         **for** $t = 0$ to $L - 1$ **do**
15:             $h_t^{(k)} \leftarrow A_t^{(k)} h_{t-1}^{(k)} + B_t^{(k)} x_t$ {State update}
16:             $y_t^{(k)} \leftarrow C_t^{(k)} h_t^{(k)}$ {Output projection}
17:         $\mathbf{Y}^{(k)} \leftarrow [y_0^{(k)\top}, \dots, y_{L-1}^{(k)\top}]^\top$
     *{Merge multi-head outputs}*
18: $\mathbf{Y}_{\text{concat}} \leftarrow \operatorname{Concat}[\mathbf{Y}^{(1)}, \dots, \mathbf{Y}^{(H)}]$
19: $\mathbf{Y}_{\text{out}} \leftarrow \operatorname{Merge}(\mathbf{Y}_{\text{concat}})$
20: **return** $\mathbf{Y}_{\text{out}}$

---

# D. On Permutation Equivariance, Positional Signals, and Endogenous Order in MIMOMamba

This appendix provides a comprehensive analysis of the positional properties within our framework. We begin by formally proving the permutation equivariance of the row-wise-softmax operation, which underpins standard attention mechanisms.

We then contrast this with the structural properties of MIMO systems.

**A Fundamental Dichotomy.** The mathematical formulations below highlight a key distinction in inductive bias: while attention layers are permutation equivariant by construction, state-space recurrences *inherently enforce sequential order*. This suggests that while Transformers (Radford et al., 2018) require exogenous Position Embeddings to break symmetry, MIMOMamba possesses an *endogenous order* sufficient for modeling temporal dependencies.

### D.1. The Permutation Equivariance of Row-Wise-Softmax

To rigorize the discussion, we restate the lemma regarding the algebraic behavior of the softmax operation under permutation.

**Lemma D.1** (Row-wise-softmax commutes with permutations). *Let $Z \in \mathbb{R}^{L \times L}$ be an arbitrary matrix and $P \in \{0,1\}^{L \times L}$ be a permutation matrix corresponding to a permutation $\pi \in S_L$. The row-wise softmax operation satisfies the following equivariance property:*

$$softmax_{row}(PZP^\top) = P\, softmax_{row}(Z)\, P^\top$$

*Proof.* Let $A = \text{softmax}_{\text{row}}(Z)$, defined strictly as $A_{ij} = \frac{e^{Z_{ij}}}{\sum_{k=1}^{L} e^{Z_{ik}}}$. Consider the permuted matrix $Z' = PZP^\top$. By the definition of permutation matrices, the entries satisfy $Z'_{ij} = Z_{\pi(i),\pi(j)}$. Applying row-wise-softmax to $Z'$ yields $A'$:

$$A'_{ij} = \frac{e^{Z'_{ij}}}{\sum_{k=1}^{L} e^{Z'_{ik}}} = \frac{e^{Z_{\pi(i),\pi(j)}}}{\sum_{k=1}^{L} e^{Z_{\pi(i),\pi(k)}}} \tag{11}$$

Since $\pi$ is a bijection on $\{1, \ldots, L\}$, the set of indices $\{\pi(k)\}_{k=1}^{L}$ is identical to $\{k\}_{k=1}^{L}$. Therefore, the denominator $\sum_{k=1}^{L} e^{Z_{\pi(i),\pi(k)}}$ is equal to $\sum_{k'=1}^{L} e^{Z_{\pi(i),k'}}$. This gives:

$$A'_{ij} = A_{\pi(i),\pi(j)}$$

Thus, $A' = PAP^\top$. $\qquad\square$

This lemma immediately implies that content-only self-attention is permutation equivariant, which motivates the need for position embeddings in Transformer-style models.

### D.2. Self-attention without positional signals is permutation equivariant

Consider a single-head self-attention layer *without* positional signals or masking:

$$\mathcal{A}(\mathbf{X}) = \text{softmax}\left(\frac{\mathbf{Q}\mathbf{K}^\top}{\sqrt{d_k}}\right)\mathbf{V}, \quad \mathbf{Q} = \mathbf{X}\mathbf{W_Q}, \ \mathbf{K} = \mathbf{X}\mathbf{W_K}, \ \mathbf{V} = \mathbf{X}\mathbf{W_V}.$$

Under $\mathbf{X} \mapsto \mathbf{P}\mathbf{X}$ we have $\mathbf{Q} \mapsto \mathbf{P}\mathbf{Q}$, $\mathbf{K} \mapsto \mathbf{P}\mathbf{K}$, $\mathbf{V} \mapsto \mathbf{P}\mathbf{V}$, hence $\mathbf{Q}\mathbf{K}^\top \mapsto \mathbf{P}(\mathbf{Q}\mathbf{K}^\top)\mathbf{P}^\top$.

By Lemma D.1, $\mathcal{A}(\mathbf{P}\mathbf{X}) = \mathbf{P}\,\mathcal{A}(\mathbf{X})$, i.e., content-only attention processes a *bag of tokens* and cannot distinguish $X$ from any time permutation—hence Transformers inject positional information.

**Causal masking alone is *not* a positional representation.** A fixed causal mask breaks global permutation equivariance (it encodes $i < j$) but does not make content interactions depend on absolute/relative indices unless coupled with a positional signal; it only forbids looking ahead. Therefore standard pipelines *still require* PEs (absolute/relative/rotary) to induce index dependence.

### D.3. SSMs encode order structurally and are not permutation equivariant

We use the selective SSM $h_t = A_t h_{t-1} + B_t x_t$, $y_t = C_t h_t$. Unrolling yields a block-lower-triangular map $\mathbf{Y} = M \triangleright \mathbf{X}$ with blocks

$$M_{ij} = \begin{cases} C_{i-1} A_{i-1:j} B_{j-1}, & i > j, \\ C_{i-1} B_{i-1}, & i = j, \\ 0, & i < j. \end{cases}$$

This structure exposes chronology via ordered products $A_{i:j+1}$.

**Proposition D.2** (SSM operators are not permutation equivariant). *Assume nontrivial temporal coupling, i.e., for some $j$ we have $M_{j+1,j} = C_{j+1}A_{j+1}B_j \neq 0$ (and by causality $M_{j,j+1} = 0$). Then there exists a permutation $S$ (the adjacent swap of indices $j$ and $j+1$) and an input $X$ such that $\mathbf{M} \triangleright (\mathbf{SX}) \neq S(\mathbf{M} \triangleright \mathbf{X})$.*

*Proof.* Let $S$ swap time $j$ and $j+1$. Take $X$ with all rows zero except $x_j = u$, where $u$ is chosen so that $M_{j+1,j}u \neq 0$ (possible since $M_{j+1,j} \neq 0$). Then $(\mathbf{M} \triangleright \mathbf{X})_{j+1} = M_{j+1,j}u \neq 0$ and $(\mathbf{M} \triangleright \mathbf{X})_j = M_{j,j}u$. Applying $S$ to the output swaps these two rows, so $(\mathbf{S}(\mathbf{M} \triangleright \mathbf{X}))_j = M_{j+1,j}u \neq 0$.

Now consider the swapped input $\mathbf{SX}$. This signal places the value $u$ at time $j+1$, implying that $(\mathbf{SX})_k = 0$ for all $k \leq j$. Due to the causality of the system (the output at time $j$ depends only on inputs up to time $j$), we have:

$$(\mathbf{M} \triangleright (\mathbf{SX}))_j = \sum_{k=0}^{j} M_{j,k}(\mathbf{SX})_k = 0.$$

Therefore, $(\mathbf{M} \triangleright (\mathbf{SX}))_j \neq (\mathbf{S}(\mathbf{M} \triangleright \mathbf{X}))_j$, proving non-equivariance. $\qquad\square$

### D.4. Matrix-valued dual and endogenous positional structure in MIMOMamba

Under the per-head commutative polynomial parameterization ($A_t, B_t, C_t \in \mathrm{span}\{I, A, \ldots, A^p\}$ for a shared base $A$), the global operator factorizes as

$$\mathbf{Y} = \left(\mathbf{L} \circledast (\mathbf{QK}^\top)\right) \triangleright \mathbf{X},$$

where $\mathbf{Q}$ stacks $\{C_t\}$, $\mathbf{K}$ stacks $\{B_t\}$, and the block-lower-triangular mask $\mathbf{L}$ has $[\mathbf{L}]_{ij} = A_{i-1:j}$ for $i > j$ and $I$ on the diagonal. Here "∘" denotes the *block-wise* Hadamard (Schur) product. Thus the positional mask $\mathbf{L}$ is *endogenous*: it is induced by state transitions (prefix products) rather than injected externally; the scalar SSD is recovered when $D=1$.

### D.5. Should MIMOMamba add positional signals? A theory-first stance

**Default: PE-free.** Since order is structural in SSMs (Section D.3) and $\mathbf{L}$ is endogenous (Section D.4), a PE-free variant is a principled default.

**When a positional bias can help (task-driven).** (i) absolute indexing queries (*e.g.*, "first token/section header"); (ii) breaking shift-invariance when document heads/tails carry special semantics; (iii) ultra-long contexts where a periodic scaffold stabilizes optimization.

### D.6. Compatibility with explicit positional injections

While our proposed framework inherently captures positional information through time-varying coefficients (as detailed in the main text), we here demonstrate that explicit positional injections are *compatible* with our commutative algebra structure. Let the per-head matrices be polynomials $A_t, B_t, C_t \in \mathrm{span}\{I, A, \ldots, A^p\}$. Crucially, the following schemes modify only the coefficient generation logic, preserving the commutativity required for efficient dual factorization.

**(A) Relative-phase injection on I/O projections (RoPE-like).** We can impose relative positional biases solely through the input/output matrices $B_t$ and $C_t$, leaving the selective dynamics of $A_t$ intact. Recall $A_t = \sum_j \alpha_{t,j} A^j$. We keep $\alpha_{t,j}$ input-dependent (selective). For $B_t$ and $C_t$, we modulate their coefficients with sinusoidal functions of absolute time $t$:

$$\beta_{t,j} = b_j^{(c)} \cos(\omega_j t) + b_j^{(s)} \sin(\omega_j t), \quad \gamma_{t,j} = g_j^{(c)} \cos(\omega_j t) - g_j^{(s)} \sin(\omega_j t).$$

This modulation imparts a relative phase shift $\cos(\omega_j(i-j))$ to the interaction term $C_iB_j$. Importantly, the state transition $A_{i:j+1}$ remains governed by the selective coefficients $\{\alpha_{\tau,\cdot}\}_{\tau=j+1}^{i}$, ensuring the system retains its input-dependent selection capabilities while acquiring an explicit relative positional bias in the projection subspace.

**(B) Absolute-index features in coefficient generators.** One may feed normalized time features $\phi(t/T)$ (e.g., Fourier features) as auxiliary inputs into the hyper-networks generating $\{\alpha_{t,j}, \beta_{t,j}, \gamma_{t,j}\}$. Since the resulting matrices remain polynomials in $A$, commutativity is preserved.

**(C) ALiBi-style decay via dynamics regularization.** Instead of external biases, one can induce distance penalties directly via the dynamics. Imposing a uniform contraction on the identity coefficient of $A_t$ (e.g., $\alpha_{t,0} \leftarrow \alpha_{t,0} - \lambda$, equivalent to $A_t \leftarrow \rho A_t$ with $\rho < 1$) enforces an exponential decay in the cumulative operator $A_{i:j+1}$.

**Note on Commutativity.** Any explicit injection must avoid arbitrary channel rotations $R_t$ that are not polynomials in the basis $A$. Such operations would eject $B_t, C_t$ from the shared commutative subalgebra, breaking the fast dual factorization algorithm.

## E. Algebraic Foundations of the Polynomial Parameterization

This appendix establishes the algebraic foundations of the proposed matrix polynomial parameterization. We focus on two critical aspects:

**Efficiency via Cayley-Hamilton:** We demonstrate how high-degree polynomials can be evaluated with constant complexity bounded by the dimension $D$.

**Commutativity for Duality:** We provide the formal proof that this parameterization induces a commutative subalgebra, enabling the core MIMO factorization.

### E.1. Efficient Polynomial Computation via Algebraic Isomorphism

**Formal Algebraic Structure: The Isomorphism Theorem.** Our model leverages the equivalence between matrix algebra and scalar polynomial arithmetic to accelerate computation. Formally, let $A \in \mathbb{R}^{D \times D}$ be the base state matrix. We consider the *evaluation homomorphism* $\phi_A : \mathbb{R}[x] \to \mathbb{R}^{D \times D}$ defined by $\phi_A(p(x)) = p(A)$. The image of this map is the subalgebra $\mathbb{R}[A]$, and its kernel is the ideal generated by the minimal polynomial $\mu_A(x)$ (Horn & Johnson, 2012; Golub & Van Loan, 2013).

By the First Isomorphism Theorem for Rings (Dummit & Foote, 2004), we have the canonical isomorphism:

$$\mathbb{R}[A] \cong \mathbb{R}[x]/\langle \mu_A(x) \rangle. \tag{12}$$

This isomorphism guarantees that operations performed in the scalar polynomial ring $\mathbb{R}[x]$ are structurally identical to operations on the matrices, modulo the minimal polynomial. This provides the theoretical justification for lifting matrix multiplication to the polynomial domain.

**Computational Reduction via Cayley-Hamilton.** While the isomorphism relies on the minimal polynomial $\mu_A(x)$, in practice, we utilize the characteristic polynomial $\chi_A(x)$ via the Cayley-Hamilton theorem (Horn & Johnson, 2012), which guarantees that $\chi_A(A) = 0$. Since $\mu_A(x)$ divides $\chi_A(x)$, the reduction via $\chi_A(x)$ is a valid upper bound for the complexity.

Let the characteristic polynomial be $p(\lambda) = \det(\lambda I - A) = \lambda^D + c_{D-1}\lambda^{D-1} + \cdots + c_0$. The theorem states:

$$A^D + c_{D-1}A^{D-1} + \cdots + c_1 A + c_0 I = 0. \tag{13}$$

Rearranging terms yields the fundamental **reduction rule**:

$$A^D = -\sum_{i=0}^{D-1} c_i A^i. \tag{14}$$

This rule implies that the algebra $\mathbb{R}[A]$ is a vector space of dimension at most $D$, spanned by the basis $\{I, A, \ldots, A^{D-1}\}$. Consequently, any high-degree polynomial matrix $A^p$ (where $p \geq D$) effectively "wraps around" the modulus, allowing it to be expressed as a linear combination of the first $D$ powers. This ensures that the computational complexity of evaluating our state polynomials is strictly bounded by the embedding dimension $D$, regardless of the sequence length or polynomial degree.

**A Concrete Example.** To visualize this reduction mechanism, let us consider a specific $3 \times 3$ base matrix $A$ (i.e., $D = 3$). Suppose its characteristic equation implies eigenvalues $\lambda \in \{2, -1 + j, -1 - j\}$, resulting in the polynomial:

$$\lambda^3 - 2\lambda - 4 = 0.$$

According to the Cayley-Hamilton theorem, replacing the scalar $\lambda$ with the matrix $A$ yields $A^3 - 2A - 4I = 0$. This gives us the specific reduction rule:

$$A^3 = 2A + 4I. \tag{15}$$

This rule implies that any power $A^p$ with $p \geq 3$ can be "folded" back into a quadratic expression. For instance, to compute $A^7$, rather than performing seven sequential matrix multiplications, we apply the rule iteratively to reduce the degree:

$$
\begin{aligned}
A^7 &= (A^3)^2 \cdot A \\
&= (2A + 4I)^2 \cdot A \qquad \text{(Substitute } A^3 = 2A + 4I) \\
&= (4A^2 + 16A + 16I) \cdot A \\
&= 4A^3 + 16A^2 + 16A \\
&= 4(2A + 4I) + 16A^2 + 16A \quad \text{(Substitute } A^3 \text{ again)} \\
&= 8A + 16I + 16A^2 + 16A \\
&= 16A^2 + 24A + 16I.
\end{aligned}
$$

As shown, the high-order power $A^7$ is successfully reduced to a linear combination of basis matrices $\{I, A, A^2\}$.

### E.2. Formal Proof of the Commutative Property

A core claim of our work is that any two matrices that are polynomials of the same base matrix $A$ commute. This property holds universally and does not depend on whether $A$ is diagonalizable.

**Theorem:** Let $P(A) = \sum_{i=0}^{m} p_i A^i$ and $Q(A) = \sum_{j=0}^{n} q_j A^j$ be two matrix polynomials of a square matrix $A$. Then $P(A)Q(A) = Q(A)P(A)$.

**Proof:** We compute the product $P(A)Q(A)$ directly from the definition:

$$
\begin{aligned}
P(A)Q(A) &= \left( \sum_{i=0}^{m} p_i A^i \right) \left( \sum_{j=0}^{n} q_j A^j \right) \\
&= \sum_{i=0}^{m} \sum_{j=0}^{n} (p_i A^i)(q_j A^j) \quad \text{(by distributivity of matrix multiplication)} \\
&= \sum_{i=0}^{m} \sum_{j=0}^{n} p_i q_j (A^i A^j) \quad \text{(since scalars commute with matrices)} \\
&= \sum_{i=0}^{m} \sum_{j=0}^{n} p_i q_j A^{i+j} \quad \text{(by the law of exponents for matrices).}
\end{aligned}
$$

Since scalar multiplication is commutative ($p_i q_j = q_j p_i$), and matrix addition is commutative, we can reorder the summation and terms:

$$
\begin{aligned}
P(A)Q(A) &= \sum_{j=0}^{n} \sum_{i=0}^{m} q_j p_i A^{j+i} \\
&= \left( \sum_{j=0}^{n} q_j A^j \right) \left( \sum_{i=0}^{m} p_i A^i \right) \\
&= Q(A)P(A).
\end{aligned}
$$

This completes the proof. Since all selective matrices within a single head ($A_t$, $B_t$, $C_t$) are defined as polynomials of the same base matrix, this theorem guarantees that they are **mutually commutative**.

### E.3. Consequence: Factorization of the Global Matrix into the Dual Form

The power of the commutative property lies in how it enables the factorization of the global transformation matrix $\mathbf{M}^{(k)}$ into the dual attention form.

Consider a generic off-diagonal block of $\mathbf{M}^{(k)}$ for $i > j$:

$$\mathbf{M}_{ij}^{(k)} = C_{i-1}^{(k)} A_{i-1:j}^{(k)} B_{j-1}^{(k)}.$$

Here, $C_i^{(k)}$, $A_{i:j+1}^{(k)}$ (which is itself a product of polynomials of $A^{(k)}$), and $B_j^{(k)}$ are all polynomials of the same base matrix. By the commutative property proven above, we can reorder this product to separate the state-dynamics term from the projection terms:

$$\mathbf{M}_{ij}^{(k)} = C_{i-1}^{(k)} A_{i-1:j}^{(k)} B_{j-1}^{(k)} = \underbrace{A_{i-1:j}^{(k)}}_{\text{Dynamics Term}} \cdot \underbrace{C_{i-1}^{(k)} B_{j-1}^{(k)}}_{\text{Projection Term}}. \tag{16}$$

This reordering is the crucial step. The "Dynamics Term" $A_{i:j+1}^{(k)}$ depends only on the relative distance between timesteps $i$ and $j$, and forms the structured causal mask $\mathbf{L}^{(k)}$. The "Projection Term" $C_i^{(k)} B_j^{(k)}$ depends only on the absolute positions of the query at time $i$ and the key at time $j$, and forms the block query-key product $(\mathbf{Q}^{(k)}(\mathbf{K}^{(k)})^\top)$.

This factorization can be applied to every block of $\mathbf{M}^{(k)}$, therefore, the entire matrix can be decomposed into a block-wise Hadamard product:

$$\mathbf{M}^{(k)} = \mathbf{L}^{(k)} \circledast (\mathbf{Q}^{(k)}(\mathbf{K}^{(k)})^\top).$$

This demonstrates precisely how the commutative property of matrix polynomials is the direct algebraic mechanism that enables the MIMO state space duality.

### E.4. An Illustrative Example of the MIMO Duality

The purpose of this appendix is not to provide a formal proof, but rather to offer a concrete, step-by-step numerical example. Our goal is to illustrate how the abstract algebraic principles discussed in the main text and proven in Appendix E operate in practice.

**Setup: A Simple 2x2 Base Matrix**   Let's consider a simple, non-symmetric 2x2 base matrix $A$ for a single head: $A = \begin{pmatrix} 1 & 2 \\ 0 & 3 \end{pmatrix}$.

We will now generate two selective matrices, $P$ and $Q$, as simple polynomials of $A$.

**Step 1: Generating Polynomial Matrices**   Let's define two polynomial functions, $p(x) = x + 2$ and $q(x) = x^2 - x$. We can compute the corresponding matrices $P = p(A)$ and $Q = q(A)$.

First, we compute the square of $A$:

$$A^2 = A \cdot A = \begin{pmatrix} 1 & 2 \\ 0 & 3 \end{pmatrix} \begin{pmatrix} 1 & 2 \\ 0 & 3 \end{pmatrix} = \begin{pmatrix} 1 & 8 \\ 0 & 9 \end{pmatrix}.$$

Now, we can compute $P$ and $Q$:

$$P = p(A) = A + 2I = \begin{pmatrix} 1 & 2 \\ 0 & 3 \end{pmatrix} + \begin{pmatrix} 2 & 0 \\ 0 & 2 \end{pmatrix} = \begin{pmatrix} 3 & 2 \\ 0 & 5 \end{pmatrix},$$

$$Q = q(A) = A^2 - A = \begin{pmatrix} 1 & 8 \\ 0 & 9 \end{pmatrix} - \begin{pmatrix} 1 & 2 \\ 0 & 3 \end{pmatrix} = \begin{pmatrix} 0 & 6 \\ 0 & 6 \end{pmatrix}.$$

**Step 2: Demonstrating Commutativity by Direct Calculation**   We now demonstrate that $P$ and $Q$ commute by directly computing both $PQ$ and $QP$.

$$PQ = \begin{pmatrix} 3 & 2 \\ 0 & 5 \end{pmatrix} \begin{pmatrix} 0 & 6 \\ 0 & 6 \end{pmatrix} = \begin{pmatrix} 0 & 30 \\ 0 & 30 \end{pmatrix},$$

$$QP = \begin{pmatrix} 0 & 6 \\ 0 & 6 \end{pmatrix} \begin{pmatrix} 3 & 2 \\ 0 & 5 \end{pmatrix} = \begin{pmatrix} 0 & 30 \\ 0 & 30 \end{pmatrix}.$$

As shown by direct computation, $PQ = QP$. This concretely illustrates the general commutative property proven in Appendix E.

**Step 3: Illustrating the Duality Factorization**   Let's use the commutative property to build a global transformation matrix $\mathbf{M}$ for a sequence of length 2. $\mathbf{M} = \begin{pmatrix} C_0 B_0 & 0 \\ C_1 A_1 B_0 & C_1 B_1 \end{pmatrix}$ , The commutative property allows us to reorder terms, for instance $C_1 A_1 B_0 = A_1 C_1 B_0$. This enables the factorization: $\mathbf{M} = \begin{pmatrix} I & 0 \\ A_1 & I \end{pmatrix} \circledast \left( \begin{bmatrix} C_0 \\ C_1 \end{bmatrix} \begin{bmatrix} B_0 & B_1 \end{bmatrix} \right) =$

$\underbrace{\begin{pmatrix} I & 0 \\ A_1 & I \end{pmatrix}}_{\mathbf{L}} \circledast \underbrace{\begin{pmatrix} C_0 B_0 & C_0 B_1 \\ C_1 B_0 & C_1 B_1 \end{pmatrix}}_{\mathbf{QK}^\top}$ . This example shows how the commutative property, which stems from the polynomial

parameterization, is the direct mechanism that enables the factorization into the dual attention form.

# F. Theoretical Foundations of MIMOMamba

In this appendix, we provide the rigorous mathematical proofs underpinning our model design, specifically focusing on the semiseparable structure of the attention mechanism and the commutativity properties of the state space.

## F.1. Proof of Proposition C.2

**Restatement of Proposition.** *The block transformation matrix $M^{(k)}$ for each head is block $D$-semiseparable.*

Here we prove that the transformation matrix $M^{(k)}$ exhibits a block $D$-semiseparable structure.

*Proof.* By definition, a matrix is $D$-semiseparable if every off-diagonal submatrix extracted entirely from below the main diagonal has rank at most $D$. We prove *block* $D$-semiseparability by exhibiting an explicit rank-$D$ factorization for an arbitrary strictly lower off-diagonal *block* submatrix.

Consider $\mathbf{M}^{(k)}_{j+1:j',\, i'+1:i}$ with $j' > j \geq i > i'$. Since for $r > c$ we have the block formula

$$\mathbf{M}^{(k)}_{r+1,c+1} \;=\; C^{(k)}_r A^{(k)}_{r:c+1} B^{(k)}_c,$$

the strictly lower block submatrix expands as

$$\mathbf{M}^{(k)}_{j+1:j',\, i'+1:i} = \begin{bmatrix} C^{(k)}_j A^{(k)}_{j:i'+1} B^{(k)}_{i'} & \cdots & C^{(k)}_j A^{(k)}_{j:i} B^{(k)}_{i-1} \\ \vdots & & \vdots \\ C^{(k)}_{j'-1} A^{(k)}_{j'-1:i'+1} B^{(k)}_{i'} & \cdots & C^{(k)}_{j'-1} A^{(k)}_{j'-1:i} B^{(k)}_{i-1} \end{bmatrix}.$$

Fix the cut index $i$. For any block-row $r \in \{j, \ldots, j'-1\}$ and block-column $c \in \{i', \ldots, i-2\}$, we can split the transition product as

$$A^{(k)}_{r:c+1} \;=\; A^{(k)}_{r:i} A^{(k)}_{i-1:c+1},$$

and for the last column $c = i-1$ we simply have $A^{(k)}_{r:c+1} = A^{(k)}_{r:i}$. Therefore, $\mathbf{M}^{(k)}_{j+1:j',\, i'+1:i}$ admits the rank-$D$ factorization

$$\mathbf{M}^{(k)}_{j+1:j',\, i'+1:i} = \underbrace{\begin{bmatrix} C^{(k)}_j A^{(k)}_{j:i} \\ \vdots \\ C^{(k)}_{j'-1} A^{(k)}_{j'-1:i} \end{bmatrix}}_{\mathbf{U} \in \mathbb{R}^{(j'-j)D \times D}} \underbrace{\begin{bmatrix} V^{(k)}_{i'} & V^{(k)}_{i'+1} & \cdots & V^{(k)}_{i-1} \end{bmatrix}}_{\mathbf{V} \in \mathbb{R}^{D \times (i-i')D}}, \tag{17}$$

where the $D \times D$ blocks in $\mathbf{V}$ are defined, for $c \in \{i', \ldots, i-1\}$, by

$$V^{(k)}_c := \begin{cases} A^{(k)}_{i-1:c+1} B^{(k)}_c, & c \leq i-2, \\ B^{(k)}_{i-1}, & c = i-1. \end{cases}$$

The key point is that the inner dimension of the factorization in (17) is exactly $D$, independent of the block sizes. Therefore, by the rank inequality for matrix products (Horn & Johnson, 2012),

$$\text{rank}\left(\mathbf{M}^{(k)}_{j+1:j',\, i'+1:i}\right) = \text{rank}(\mathbf{U}\mathbf{V}) \leq \min\{\text{rank}(\mathbf{U}), \text{rank}(\mathbf{V})\} \leq D.$$

Since this bound holds for every strictly lower off-diagonal block submatrix, $\mathbf{M}^{(k)}$ is block $D$-semiseparable. This extends the semiseparable structure exploited by Mamba-2 (Dao & Gu, 2024) to our block matrix formulation, which preserves the computational efficiency of modern SSM kernels. $\square$

## F.2. Matrix Commutativity and its Role in Model Design

This appendix provides a self-contained theoretical justification for a core design choice in the MIMOMamba model: the parameterization of the time-varying state matrices $A_t, B_t, C_t$. We aim to make the material accessible to readers with a foundational knowledge of linear algebra, we first develop intuition and then present formal proofs.

### F.2.1. MOTIVATION: WHY COMMUTATIVITY IS THE KEY TO DUALITY

Our goal is to generalize the SSD from the SISO case to the MIMO setting. Let's start by unrolling the MIMO recurrence relation:

$$h_t = A_t h_{t-1} + B_t x_t, \quad y_t = C_t h_t.$$

The output $y_i$ is a function of all preceding inputs $x_j$ (where $j \leq i$). For example, the influence of input $x_j$ on a later output $y_i$ (with $i > j$) is propagated through the hidden states:

$$y_i = C_i A_i A_{i-1} \cdots A_{j+1} B_j x_j + \text{other terms}.$$

To establish a duality with a structured attention mechanism of the form $\text{Attention} \propto (\mathcal{L} \odot (QK^\top))V$, we need to factor the overall transformation matrix. Specifically, we must be able to rearrange the terms in the expression $C_i A_i \cdots A_{j+1} B_j$ to separate the "mask-like" components (the $A_t$ matrices) from the "query-like" ($C_i$) and "key-like" ($B_j$) components. This requires reordering the matrix products, for instance:

$$C_i A_k B_j \overset{?}{=} (A_k)(C_i B_j).$$

This reordering is only permissible if the matrices commute, i.e., $C_i A_k = A_k C_i$. The challenge, therefore, is to design a model architecture where all selective state matrices $\{A_t, B_t, C_t\}$ for a given head commute with one another.

A simple and powerful way to guarantee this is to generate all of them from a common source: we define them as **matrix polynomials** of a single, shared, time-invariant base matrix $A_{\text{base}}$. Any two polynomials of the same matrix, say $p_1(A_{\text{base}})$ and $p_2(A_{\text{base}})$, are guaranteed to commute.

This elegant solution, however, raises a critical question: by constraining our matrices to be polynomials of a single base matrix, are we severely limiting the model's expressive power? This question is the central motivation for the theoretical exploration that follows.

### F.2.2. THE SEARCH FOR EXPRESSIVE POWER: CENTRALIZERS AND POLYNOMIALS

To determine if our polynomial parameterization is restrictive, we must first define the "universe" of all possible matrices that satisfy our fundamental requirement of commutativity.

1. **The Centralizer $C(A)$: The Gold Standard of Expressiveness.** For a given base matrix $A$, the set of *all* matrices that commute with it is called its **centralizer**, denoted $C(A)$.

   $$C(A) = \{B \in \mathbb{C}^{n \times n} \mid AB = BA\}.$$

   The centralizer $C(A)$ represents the theoretical upper bound on expressiveness; it is the largest possible family of transformations that are compatible with the dynamics of $A$.

2. **The Polynomial Algebra** $\text{Poly}(A)$**: Our Model's Actual Capability.** Our model generates matrices using polynomials. This set is known as the **polynomial algebra** of $A$, denoted $\text{Poly}(A)$.

$$\text{Poly}(A) = \{p(A) \mid p(x) \in \mathbb{C}[x]\}.$$

Since any polynomial of $A$ commutes with $A$, we always have the inclusion $\text{Poly}(A) \subseteq C(A)$.

The crucial question now becomes: **When does our model's capability match the theoretical ideal? That is, under what conditions on the base matrix $A$ does $\text{Poly}(A) = C(A)$?**

The answer lies in the structural properties of the matrix $A$, specifically whether it is **derogatory** or **non-derogatory**.

A matrix is **non-derogatory** if its minimal polynomial is identical to its characteristic polynomial. While this definition is precise, a more intuitive understanding comes from its Jordan canonical form: a matrix is non-derogatory if and only if each of its eigenvalues corresponds to exactly *one* Jordan block.

- **Intuition for a Non-derogatory Matrix:** Think of its structure as "simple" or "non-redundant". For each fundamental mode of operation (eigenvalue), there is only a single, unified chain of command (one Jordan block). Any compatible company-wide policy (a commuting matrix) must be a single directive that propagates down this chain, which can be described by a single polynomial.

- **Intuition for a Derogatory Matrix:** Think of its structure as "complex" or "redundant". At least one mode of operation (eigenvalue) has multiple, independent sub-teams (multiple Jordan blocks) that do not interact. A simple example is the identity matrix $I$, where every direction is an eigenvector for the eigenvalue 1. A commuting matrix can act differently and independently on these different sub-teams (e.g., scaling one direction by 2 and another by 3). Such a complex policy cannot be described by a single polynomial of $I$, as any polynomial $p(I)$ must act identically (as $p(1)I$) on all sub-teams.

This intuition leads us to the main theorem.

### F.2.3. THE CENTRALIZER THEOREM AND ITS IMPLICATIONS FOR MIMOMAMBA

The relationship between a matrix's structure and its commuting algebras is captured by a cornerstone result in linear algebra, which we refer to as the Centralizer Theorem.

**Theorem F.1** (Centralizer Theorem). *A matrix $A \in \mathbb{C}^{n \times n}$ is non-derogatory if and only if its centralizer is equal to its polynomial algebra, i.e., $C(A) = \text{Poly}(A)$.*

This theorem provides a definitive answer to our question and has profound implications for our model design:

- **Choosing a non-derogatory $A_{\text{base}}$ is optimal.** By selecting a non-derogatory matrix for our base matrix, we guarantee that $C(A_{\text{base}}) = \text{Poly}(A_{\text{base}})$. This means our polynomial parameterization is **complete and exhaustive**. It is not a restrictive shortcut; it is a fully expressive method that generates every possible linear transformation that commutes with the base dynamics. Our model thus achieves maximum theoretical expressiveness under the commutativity constraint.

- **Choosing a derogatory $A_{\text{base}}$ would be a critical flaw.** If we were to choose a derogatory matrix (like the identity matrix $I$), the inclusion would be strict: $\text{Poly}(I) \subset C(I)$. Our model would only be able to generate scalar multiples of the identity, while the true space of commuting matrices is the entire space of $n \times n$ matrices. This would severely cripple the model's capacity.

Therefore, the choice of a non-derogatory base matrix is a principled decision that grounds the MIMOMamba architecture in a solid theoretical framework, which guarantees both computational tractability and maximal expressive power.

### F.2.4. FORMAL PROOF OF THE CENTRALIZER THEOREM

For mathematical completeness, we now present the formal proof of the Centralizer Theorem. The central tool for this proof is the **Jordan canonical form**, as it provides a standard representation for any square matrix, including those that are not

diagonalizable. This generalization is particularly important in our context, as foundational matrices in state-space models like the HiPPO matrix are often non-diagonalizable (Gu et al., 2020; Gu, 2023).

Conceptually, the Jordan form is an extension of diagonalization. While a diagonalizable matrix decomposes into a simple diagonal matrix of its eigenvalues, a non-diagonalizable matrix decomposes into **Jordan blocks**. Each block features a single eigenvalue on its main diagonal and, crucially, can have ones on the superdiagonal, which reflect the matrix's more complex, non-decomposable behavior.

As we will demonstrate, the structure of these Jordan blocks—specifically, how many blocks exist for each eigenvalue—is precisely what determines whether a commuting matrix $B$ can be expressed as a polynomial of $A$. The following theorems explore this intimate relationship, building from simpler cases to the general result.

**Lemma F.2.** *Let $A$ and $B$ be square matrices of orders $m$ and $n$, respectively. The homogeneous matrix equation $AX = XB$ has only the trivial solution $X = 0$ if and only if $A$ and $B$ have no common eigenvalues.*

*Proof.* **Necessity ($\Rightarrow$).** This direction is equivalent to proving its contrapositive: if matrices $A$ and $B$ share a common eigenvalue, then the equation $AX = XB$ has a non-trivial solution.

Suppose $A$ and $B$ share a common eigenvalue $\lambda$. Then there exists a non-zero right eigenvector $p$ for $A$ such that $Ap = \lambda p$, and a non-zero left eigenvector $q$ for $B$ such that $q^\top B = \lambda q^\top$.

Define the matrix $X$ as the outer product of $p$ and $q$, i.e., $X = pq^\top$. Since both $p$ and $q$ are non-zero vectors, $X$ is a non-zero matrix.

We can now verify that this $X$ is a solution. On the one hand, we compute $AX$:

$$AX = A(pq^\top) = (Ap)q^\top = (\lambda p)q^\top = \lambda(pq^\top) = \lambda X$$

On the other hand, we compute $XB$:

$$XB = (pq^\top)B = p(q^\top B) = p(\lambda q^\top) = \lambda(pq^\top) = \lambda X$$

Thus, we have $AX = XB$, and since $X$ is non-zero, a non-trivial solution exists. This completes the proof of necessity.

**Sufficiency ($\Leftarrow$).** If an $m \times n$ matrix $X$ satisfies $AX = XB$, then for every non-negative integer $k$, we have $A^k X = XB^k$. Consequently, for any polynomial $f(x)$, it holds that $f(A)X = Xf(B)$.

Let the characteristic polynomial of $A$ be $\varphi(\lambda) = (\lambda - \lambda_1) \cdots (\lambda - \lambda_m)$. By the Cayley–Hamilton theorem (Horn & Johnson, 2012), $\varphi(A) = 0$. Since $A$ and $B$ have no common eigenvalues, the determinant of $\varphi(B)$ is non-zero: $\det \varphi(B) = \det(B - \lambda_1 I) \cdots \det(B - \lambda_m I) \neq 0$. Thus, $\varphi(B)$ is invertible. From the relation $\varphi(A)X = X\varphi(B)$, we have $0 \cdot X = X\varphi(B)$, which implies $X = 0$ by right-multiplying by $\varphi(B)^{-1}$.

$\square$

**Theorem F.3.** *Let $J_m(\lambda)$ denote the $m \times m$ Jordan block with eigenvalue $\lambda$. Let $J = \mathrm{diag}(J_{m_1}(\lambda_1), \ldots, J_{m_s}(\lambda_s))$, where the eigenvalues $\lambda_1, \ldots, \lambda_s$ are **pairwise distinct**. A matrix $B$ commutes with $J$ if and only if $B$ is a block diagonal matrix*

$$B = \mathrm{diag}(B_{11}, B_{22}, \ldots, B_{ss}),$$

*where each diagonal block $B_{ii}$ is of size $m_i \times m_i$ and commutes with the corresponding Jordan block $J_{m_i}(\lambda_i)$.*

*Proof.* **Sufficiency** is immediate. For **necessity**, we partition $B$ conformally with $J$ as $B = [B_{ij}]_{1 \leq i,j \leq s}$. The equation $JB = BJ$ implies $J_{m_i}(\lambda_i)B_{ij} = B_{ij}J_{m_j}(\lambda_j)$ for all $i, j$. When $i \neq j$, the matrices $J_{m_i}(\lambda_i)$ and $J_{m_j}(\lambda_j)$ have no common eigenvalues. By Lemma F.2, we must have $B_{ij} = 0$. Thus, $B$ must be block diagonal. $\square$

*Remark* F.4. The hypothesis that the eigenvalues $\lambda_1, \ldots, \lambda_s$ are pairwise distinct is essential. To see this, consider the following counterexample where the matrix is derogatory (i.e., has repeated eigenvalues across different Jordan blocks). Let:

$$A = \begin{bmatrix} 1 & 1 & 0 \\ 0 & 1 & 0 \\ 0 & 0 & 1 \end{bmatrix}, \qquad B = \begin{bmatrix} 0 & 0 & 2 \\ 0 & 0 & 0 \\ 0 & 3 & 0 \end{bmatrix}.$$

The matrix $A$ is in Jordan form, composed of two blocks for the same eigenvalue $\lambda = 1$: $J_2(1)$ and $J_1(1)$. A direct calculation shows that $AB = BA$, yet $B$ is clearly not block-diagonal in the partition corresponding to $A$'s Jordan blocks. The off-diagonal blocks are non-zero, which shows that Theorem F.3 does not hold for derogatory matrices.

**Theorem F.5.** *Let $J$ be an $m \times m$ Jordan block. A matrix $B$ commutes with $J$ if and only if $B$ can be expressed as a polynomial in $J$ of degree at most $m - 1$.*

*Proof.* **Sufficiency** is clear. For **necessity**, let $J = \lambda I + N$, where $N$ is the nilpotent matrix with ones on the first superdiagonal and zeros elsewhere. The condition $BJ = JB$ is equivalent to $BN = NB$. It is a standard result that a matrix commutes with $N$ if and only if it is an upper triangular Toeplitz matrix.

$$B = \begin{bmatrix} b_0 & b_1 & \dots & b_{m-1} \\ & b_0 & \ddots & \vdots \\ & & \ddots & b_1 \\ & & & b_0 \end{bmatrix}.$$

Such a matrix can be uniquely written as a polynomial in $N$: $B = \sum_{k=0}^{m-1} b_k N^k$. Since $N = J - \lambda I$, $B$ can also be expressed as a polynomial in $J$. $\qquad\square$

With these lemmas, we can now prove the main theorem.

*Proof of the Centralizer Theorem.* ($\Rightarrow$) Assume $A$ is non-derogatory. This implies that in its Jordan canonical form, $J = P^{-1}AP$, each distinct eigenvalue corresponds to exactly one Jordan block. So, $J = \operatorname{diag}(J_{m_1}(\lambda_1), \dots, J_{m_s}(\lambda_s))$, where the $\lambda_i$ are all distinct.

If $AB = BA$, then $J(P^{-1}BP) = (P^{-1}BP)J$. By Theorem F.3, $P^{-1}BP$ must be block diagonal, $P^{-1}BP = \operatorname{diag}(B_1, \dots, B_s)$, where each $B_i$ commutes with $J_{m_i}(\lambda_i)$.

By Theorem F.5, each $B_i$ is a polynomial in its corresponding Jordan block, $B_i = f_i(J_{m_i}(\lambda_i))$.

Using the Chinese Remainder Theorem for polynomials (Dummit & Foote, 2004), we can construct a single polynomial $g(x)$ such that $g(x) \equiv f_i(x) \pmod{(x - \lambda_i)^{m_i}}$ for all $i = 1, \dots, s$. This single polynomial $g(x)$ acts on each Jordan block $J_{m_i}(\lambda_i)$ in the same way as its respective polynomial $f_i(x)$. Therefore, $g(J) = \operatorname{diag}(f_1(J_1), \dots, f_s(J_s)) = P^{-1}BP$. This implies $B = Pg(J)P^{-1} = g(PJP^{-1}) = g(A)$, proving that $B$ is a polynomial in $A$.

($\Leftarrow$) (By contrapositive) Assume $A$ is derogatory. This means at least one eigenvalue, say $\lambda_k$, corresponds to two or more Jordan blocks. For instance, $A$ is similar to a Jordan form $J = \operatorname{diag}(J_a(\lambda_k), J_b(\lambda_k), \dots)$. In this case, we can construct a commuting matrix that is not a polynomial in $A$. Consider a block diagonal matrix $B'$ corresponding to $J$ that is the identity on the $J_a(\lambda_k)$ block and the zero matrix on the $J_b(\lambda_k)$ block (and zero elsewhere). This $B'$ clearly commutes with $J$. However, any polynomial $p(A)$ corresponds to $p(J)$, which must apply the same polynomial transformation $p(J_m(\lambda_k))$ to all blocks corresponding to the same eigenvalue $\lambda_k$. It cannot act as the identity on one block and as zero on another for the same eigenvalue. Thus, the matrix $B = PB'P^{-1}$ is in $C(A)$ but not in $\operatorname{Poly}(A)$. $\qquad\square$

### F.2.5. A FIELD-INDEPENDENT PROOF VIA RATIONAL CANONICAL FORM

**Reader's note.** The proof above is intuitive and pedagogically transparent, but it relies on the Jordan canonical form, which exists only over algebraically closed fields such as $\mathbb{C}$. To fully substantiate the real-domain claim made in the Remark of §F.2.7, we provide here a self-contained proof of the Centralizer Theorem that is valid over an *arbitrary* field $\mathbb{F}$, in particular over $\mathbb{R}$. The argument uses the *rational canonical form* (equivalently, the invariant-factor decomposition of finitely generated modules over the polynomial ring $\mathbb{F}[x]$) (Dummit & Foote, 2004), thereby avoiding any appeal to algebraic closure. Readers without a background in module theory may safely skip the details of this subsection: the conclusion is identical to that of the Centralizer Theorem proved above, and the present treatment is included only to remove any residual concern about transferring the result from $\mathbb{C}$ to $\mathbb{R}$.

**Theorem F.6** (Centralizer Theorem over an Arbitrary Field). *Let $\mathbb{F}$ be any field, let $A \in \mathbb{F}^{n \times n}$, and define*

$$C_{\mathbb{F}}(A) := \{ B \in \mathbb{F}^{n \times n} : AB = BA \}, \qquad \mathbb{F}[A] := \{ g(A) : g \in \mathbb{F}[x] \}.$$

*Let $m_A(x)$ and $p_A(x)$ denote the minimal and characteristic polynomials of $A$, respectively. Then*

$$C_{\mathbb{F}}(A) = \mathbb{F}[A] \quad \Longleftrightarrow \quad m_A(x) = p_A(x).$$

*In particular, taking $\mathbb{F} = \mathbb{R}$ yields the real-valued case relevant to our implementation.*

*Proof.* Let $V = \mathbb{F}^n$ and let $T \colon V \to V$ denote the linear map induced by $A$, that is, $T(v) = Av$. For a polynomial $q(x) = \sum_{k=0}^{d} q_k x^k \in \mathbb{F}[x]$, write $q(T) := \sum_{k=0}^{d} q_k T^k$. With this convention, $\mathbb{F}[A]$ coincides with $\{\, q(T) : q \in \mathbb{F}[x] \,\}$ written in matrix form. Since every polynomial in $T$ commutes with $T$,

$$\mathbb{F}[T] \subseteq C_{\mathbb{F}}(T).$$

The nontrivial assertion is the reverse inclusion under the hypothesis $m_T = p_T$, together with its converse.

**Module-theoretic setup.** We view $V$ as a module over the polynomial ring $\mathbb{F}[x]$ by declaring that multiplication by $x$ means applying $T$:

$$x \cdot v := Tv, \qquad v \in V.$$

By the structure theorem for finitely generated modules over the principal ideal domain $\mathbb{F}[x]$ (Dummit & Foote, 2004), there exist monic nonconstant polynomials $a_1(x), \ldots, a_r(x) \in \mathbb{F}[x]$ with

$$a_1 \mid a_2 \mid \cdots \mid a_r$$

such that

$$V \cong \bigoplus_{i=1}^{r} \mathbb{F}[x]/(a_i(x))$$

as $\mathbb{F}[x]$-modules. The polynomials $a_i$ are the *invariant factors* of $T$, and they satisfy

$$p_T(x) = \prod_{i=1}^{r} a_i(x), \qquad m_T(x) = a_r(x).$$

Consequently,

$$m_T(x) = p_T(x) \quad \Longleftrightarrow \quad r = 1.$$

Equivalently, $m_T = p_T$ holds if and only if $V$ is a *cyclic* $\mathbb{F}[x]$-module, meaning that there exists a vector $v \in V$ for which

$$V = \mathrm{span}_{\mathbb{F}}\{\, v,\, Tv,\, T^2 v,\, \ldots,\, T^{n-1} v \,\}.$$

Since $\dim_{\mathbb{F}} V = n$, the family $\{\, v,\, Tv,\, \ldots,\, T^{n-1} v \,\}$ is then a basis of $V$.

**Forward direction:** $m_T = p_T \implies C_{\mathbb{F}}(T) = \mathbb{F}[T]$.

Assume $m_T = p_T$. By the discussion above, there exists $v \in V$ such that

$$\mathcal{B} = \{\, v,\, Tv,\, T^2 v,\, \ldots,\, T^{n-1} v \,\}$$

is a basis of $V$. Let $S \in C_{\mathbb{F}}(T)$, i.e., $ST = TS$. Since $\mathcal{B}$ is a basis, there exist unique scalars $c_0, \ldots, c_{n-1} \in \mathbb{F}$ with

$$Sv = c_0\, v + c_1\, Tv + \cdots + c_{n-1}\, T^{n-1} v.$$

Define

$$g(x) := c_0 + c_1 x + \cdots + c_{n-1} x^{n-1} \in \mathbb{F}[x],$$

so that $Sv = g(T)v$. Because $S$ commutes with $T$, for every $j \in \{0, 1, \ldots, n-1\}$ we have

$$S(T^j v) \;=\; T^j(Sv) \;=\; T^j g(T)v \;=\; g(T)\, T^j v.$$

Thus $S$ and $g(T)$ agree on every element of the basis $\mathcal{B}$. Hence $S = g(T) \in \mathbb{F}[T]$. Since $S \in C_{\mathbb{F}}(T)$ was arbitrary, we conclude $C_{\mathbb{F}}(T) \subseteq \mathbb{F}[T]$, and combining with the trivial inclusion yields $C_{\mathbb{F}}(T) = \mathbb{F}[T]$.

**Converse:** $C_{\mathbb{F}}(T) = \mathbb{F}[T] \implies m_T = p_T$.

We prove the equivalent contrapositive statement $m_T \neq p_T \implies C_{\mathbb{F}}(T) \neq \mathbb{F}[T]$ by exhibiting an explicit $S \in C_{\mathbb{F}}(T) \setminus \mathbb{F}[T]$. Assume $m_T \neq p_T$, equivalently $r \geq 2$ in the invariant-factor decomposition above. Write

$$V \cong M_1 \oplus \cdots \oplus M_r, \qquad M_i := \mathbb{F}[x]/(a_i(x)),$$

and let $e_i \in M_i$ denote the class of 1. Set $m(x) := a_r(x) = m_T(x)$. Since $a_1 \mid m$, the quotient $m(x)/a_1(x)$ is itself a polynomial in $\mathbb{F}[x]$ (not merely a rational function); we use this polynomial below.

Define an $\mathbb{F}[x]$-linear map $\psi \colon M_1 \to M_r$ by

$$\psi(e_1) := \frac{m(x)}{a_1(x)} \cdot e_r.$$

To verify well-definedness, recall that in $M_1$ we have $a_1(x) \cdot e_1 = 0$; in $M_r$ we then compute

$$a_1(x) \cdot \psi(e_1) = a_1(x) \cdot \frac{m(x)}{a_1(x)} \cdot e_r = m(x) \cdot e_r = 0,$$

since $M_r = \mathbb{F}[x]/(m(x))$. Hence $\psi$ extends uniquely to a well-defined $\mathbb{F}[x]$-linear map on all of $M_1 = \mathbb{F}[x]/(a_1(x))$. Moreover, $\psi \neq 0$: the class of $m(x)/a_1(x)$ in $\mathbb{F}[x]/(m(x))$ is nonzero because $\deg(m/a_1) < \deg m$.

Now define an $\mathbb{F}[x]$-linear endomorphism $S \colon V \to V$ by

$$S\big|_{M_1} = \psi, \qquad S\big|_{M_i} = 0 \text{ for } i = 2, \ldots, r.$$

Being $\mathbb{F}[x]$-linear, $S$ commutes with the action of $x$ on $V$, which is precisely $T$; thus $S \in C_{\mathbb{F}}(T)$.

We claim that $S \notin \mathbb{F}[T]$. Suppose, for contradiction, that $S = q(T)$ for some $q \in \mathbb{F}[x]$. By construction, $S$ vanishes on $M_r$; hence $q(T)$ vanishes on $M_r$. Since $M_r \cong \mathbb{F}[x]/(m(x))$, a polynomial $q$ acts as zero on $M_r$ if and only if $m(x) \mid q(x)$. Because $a_i \mid m$ for every $i$, the divisibility $m \mid q$ implies that $q(T)$ vanishes on every summand $M_i$, hence on all of $V$. Therefore $S = q(T) = 0$, contradicting the fact that $S$ is nonzero on $M_1$. Thus $S \in C_{\mathbb{F}}(T) \setminus \mathbb{F}[T]$, so $C_{\mathbb{F}}(T) \neq \mathbb{F}[T]$.

Combining the two directions yields

$$C_{\mathbb{F}}(T) = \mathbb{F}[T] \iff m_T(x) = p_T(x).$$

Translating from $T$ back to its matrix representation $A$ gives the stated theorem; taking $\mathbb{F} = \mathbb{R}$ yields the real-valued case used in our implementation. $\qquad\square$

### F.2.6. EQUIVALENT CHARACTERIZATIONS OF NON-DEROGATORY MATRICES

The property of being non-derogatory is so fundamental in linear algebra that its importance is underscored by its many equivalent definitions. The equivalence of the following statements is a classic result in matrix theory (Horn & Johnson, 2012).

For a linear operator $A$ on an $n$-dimensional complex vector space, the following statements are equivalent:

1. The characteristic polynomial of $A$ is equal to its minimal polynomial.

2. In the Jordan canonical form of $A$, there is exactly one Jordan block for each eigenvalue.

3. The geometric multiplicity of every eigenvalue of $A$ is 1 (i.e., $\dim(\ker(A - \lambda_i I)) = 1$).

4. $A$ is similar to a companion matrix. Specifically, if the characteristic polynomial of $A$ is the monic polynomial $p(x) = c_0 + c_1 x + \cdots + c_{n-1} x^{n-1} + x^n$, then $A$ is similar to the **Frobenius companion matrix** of $p(x)$, which is defined as:

$$C(p) = \begin{bmatrix} 0 & 0 & \cdots & 0 & -c_0 \\ 1 & 0 & \cdots & 0 & -c_1 \\ 0 & 1 & \cdots & 0 & -c_2 \\ \vdots & \vdots & \ddots & \vdots & \vdots \\ 0 & 0 & \cdots & 1 & -c_{n-1} \end{bmatrix}.$$

This matrix has the coefficients of the polynomial (with negative signs) in its last column and ones on the subdiagonal, which provides a canonical representation for non-derogatory matrices.

5. There exists a cyclic vector $\alpha$ such that $\{\alpha, A\alpha, A^2\alpha, \ldots, A^{n-1}\alpha\}$ is a basis for the space.

6. **The centralizer of $A$ equals its polynomial algebra, i.e., $C(A) = \mathrm{Poly}(A)$.**

7. The space admits only a finite number of $A$-invariant subspaces.

Item 6 on this list is precisely the core result that underpins our model's design, connecting a clean algebraic condition to a guarantee of expressive completeness for the MIMOMamba architecture.

### F.2.7. GENERICITY OF THE NON-DEROGATORY CONDITION

A potential concern regarding the theoretical framework presented in Theorem F.1 is whether the requirement for the base matrix $A$ to be non-derogatory imposes a restrictive constraint on the model's parameter space. In this section, we demonstrate that this condition is, in fact, extremely weak. Specifically, we show that non-derogatory matrices are *generic* in the space of square matrices, meaning they form a dense, open subset with a complement of measure zero (Horn & Johnson, 2012).

**Algebraic Perspective and Intuition.** Recall that a matrix $A \in \mathbb{C}^{D \times D}$ is non-derogatory if and only if its minimal polynomial coincides with its characteristic polynomial ($m_A(\lambda) = p_A(\lambda)$). To provide a more geometric intuition:

- **Geometric Multiplicity:** $A$ is non-derogatory if and only if the geometric multiplicity of every eigenvalue is exactly 1.

- **Jordan Structure:** Equivalently, in its Jordan Normal Form, there is exactly one Jordan block associated with each distinct eigenvalue.

While the definition relies on polynomial invariants, there exists a simple and powerful **sufficient condition**: *if a matrix has $D$ distinct eigenvalues, it is guaranteed to be non-derogatory.* This is because distinct eigenvalues imply that the characteristic polynomial has no repeated roots, forcing the minimal polynomial to include all factors, thus $m_A(\lambda) = p_A(\lambda)$. Note that the converse is not true (a non-derogatory matrix can have repeated eigenvalues, e.g., a single large Jordan block), but the *distinct-eigenvalue* case is the most prevalent in generic settings.

**Remark: Applicability to Real-Valued Models.** While the theoretical analysis above utilizes the field of complex numbers $\mathbb{C}$ and the standard Jordan Normal Form for algebraic convenience (as $\mathbb{C}$ is algebraically closed), the conclusions regarding the centralizer structure remain fully valid over the real field $\mathbb{R}$. For a real-valued matrix $A \in \mathbb{R}^{D \times D}$ (as used in our implementation), the same algebraic characterization holds by considering the *Real Jordan Form* or the *Rational Canonical Form* (Horn & Johnson, 2012). Consequently, the polynomial representation of the centralizer is preserved in the real domain, justifying the application of our theory to real-valued neural network architectures.

**Probabilistic Implication for Initialization.** The condition of having repeated eigenvalues is algebraically characterized by the vanishing of the *discriminant* of the characteristic polynomial. Let $p(\lambda) = \det(\lambda I - A)$ denote the characteristic polynomial of $A$. We define $\mathcal{D}(A) = \mathrm{disc}(\det(\lambda I - A))$, where $\mathrm{disc}(\cdot)$ is the polynomial discriminant operator which vanishes if and only if the polynomial has multiple roots.

Since $\mathcal{D}(A)$ is a non-trivial multivariate polynomial function of the entries of $A$, its zero set constitutes a proper algebraic subvariety of lower dimension.

This leads to a direct probabilistic consequence for the initialization of our model, consistent with standard results in Random Matrix Theory (Mehta, 2004).

**Proposition F.7** (Almost Sure Non-Derogatory Initialization). *Let the entries of the base matrix $A \in \mathbb{R}^{D \times D}$ be drawn independently from any continuous probability distribution (e.g., Gaussian $\mathcal{N}(0, \sigma^2)$, Uniform $\mathcal{U}[-a, a]$, or Xavier/Kaiming initialization). Then,*

$$\mathbb{P}(A \text{ is non-derogatory}) = 1. \tag{18}$$

*Proof.* Let $\mathcal{S}_{\text{repeated}}$ be the set of matrices with repeated eigenvalues. This set is contained within the zero set of the discriminant polynomial $\mathcal{Z}(\mathcal{D}) = \{M \in \mathbb{R}^{D \times D} \mid \mathcal{D}(M) = 0\}$. Since the entries are drawn from a continuous distribution, the probability mass assigned to any set of Lebesgue measure zero (such as an algebraic subvariety) is zero. Consequently, almost all random matrices possess distinct eigenvalues. Since distinct eigenvalues imply the non-derogatory property, we have:

$$\mathbb{P}(A \text{ is non-derogatory}) \geq \mathbb{P}(A \text{ has distinct eigenvalues}) = 1.$$

Thus, standard random initialization automatically yields a non-derogatory base matrix with probability 1. $\qquad \square$

*Remark* F.8 (Why the complex Jordan-form proof still yields a real polynomial). *Reader's note.* The following remark, intended for mathematically inclined readers, provides an alternative justification for the real-domain claim of §F.2.7: it shows that even if one applies the complex Jordan-form proof of the Centralizer Theorem to a real matrix $A \in \mathbb{R}^{D \times D}$, the resulting polynomial representation can always be chosen with *real* coefficients. This serves as an independent cross-check of the field-independent proof given in Theorem F.6.

Let $A \in \mathbb{R}^{D \times D}$ be non-derogatory. We first observe that the minimal polynomial of $A$ is the same whether $A$ is viewed as a real or as a complex matrix. Indeed, suppose $q(x) \in \mathbb{C}[x]$ satisfies $q(A) = 0$; writing $q(x) = u(x) + i\,v(x)$ with $u, v \in \mathbb{R}[x]$, we have

$$q(A) \;=\; u(A) + i\,v(A) \;=\; 0,$$

which, since $A$ is real, forces $u(A) = 0$ and $v(A) = 0$ separately. Hence the real minimal polynomial $m_A \in \mathbb{R}[x]$ divides both $u$ and $v$, and therefore divides $q$ in $\mathbb{C}[x]$; allowing complex coefficients cannot produce a smaller minimal polynomial.

Since the characteristic polynomial $p_A(x) = \det(xI - A)$ has the same coefficients whether $A$ is viewed over $\mathbb{R}$ or over $\mathbb{C}$, combining this with the equality $m_A^{\mathbb{R}}(x) = m_A^{\mathbb{C}}(x)$ established in the previous paragraph shows that $A$ is non-derogatory over $\mathbb{C}$ if and only if it is non-derogatory over $\mathbb{R}$. Hence the real non-derogatory hypothesis on $A$ permits us to apply the complex Centralizer Theorem.

Now let $B \in \mathbb{R}^{D \times D}$ satisfy $AB = BA$. Applying the complex Centralizer Theorem (Theorem stated in §F.2.3 and proved above via the Jordan form) yields a polynomial $g \in \mathbb{C}[x]$ with $B = g(A)$. Since $A$ and $B$ are real, taking entrywise complex conjugation gives

$$B \;=\; \overline{B} \;=\; \overline{g(A)} \;=\; \overline{g}(A),$$

where $\overline{g}$ denotes coefficientwise conjugation of $g$, and the last equality uses the fact that $\overline{A^k} = A^k$ for all $k \geq 0$ because $A$ is real. Define the real polynomial

$$h(x) \;:=\; \tfrac{1}{2}\big(g(x) + \overline{g}(x)\big) \;\in\; \mathbb{R}[x].$$

Then

$$h(A) \;=\; \tfrac{1}{2}\big(g(A) + \overline{g}(A)\big) \;=\; \tfrac{1}{2}(B + B) \;=\; B.$$

Hence every real matrix commuting with $A$ admits a real-polynomial representation, namely $B \in \mathbb{R}[A]$. This confirms that the complex Jordan-form proof is compatible with the real-domain conclusion, in agreement with Theorem F.6.

*Remark* F.9 (Algebraic-variety characterization of the derogatory set). *Reader's note.* The following remark, intended for mathematically inclined readers, refines the discriminant-based argument used in the proof of Proposition F.7 by exhibiting the derogatory set as an *explicit* algebraic subvariety of $\mathbb{R}^{D^2}$, without passing through complex eigenvalues. The two characterizations (discriminant vanishing versus rank deficiency of the polynomial-in-$A$ matrix introduced below) are complementary; both yield Lebesgue measure zero, but the rank-based characterization is sharper.

For $A \in \mathbb{R}^{D \times D}$, define the $D^2 \times D$ matrix

$$\Phi(A) \;:=\; \begin{bmatrix} \text{vec}(I) & \text{vec}(A) & \text{vec}(A^2) & \cdots & \text{vec}(A^{D-1}) \end{bmatrix},$$

where $\text{vec}(\cdot)$ stacks the entries of a $D \times D$ matrix into a column vector of length $D^2$. By the Cayley–Hamilton theorem (Horn & Johnson, 2012), $\deg m_A \leq D$; moreover, $\deg m_A = D$ if and only if $I, A, A^2, \ldots, A^{D-1}$ are linearly independent in $\mathbb{R}^{D \times D}$. Since $\deg p_A = D$ always, we obtain the chain of equivalences

$$A \text{ is non-derogatory} \;\Longleftrightarrow\; m_A = p_A \;\Longleftrightarrow\; \text{rank}\,\Phi(A) = D.$$

Equivalently, $A$ is derogatory if and only if every $D \times D$ minor of $\Phi(A)$ vanishes. Each such minor is a polynomial in the entries of $A$, so the derogatory set is the common zero locus of finitely many real polynomials, i.e., a real algebraic subvariety of $\mathbb{R}^{D^2}$.

This subvariety is *proper*. For example, take $N$ to be a single nilpotent Jordan block of size $D$; then $I, N, N^2, \ldots, N^{D-1}$ are linearly independent, so $\operatorname{rank} \Phi(N) = D$ and $N$ is non-derogatory. Hence not all the defining minors of $\Phi$ vanish identically.

By the previous paragraph, at least one $D \times D$ minor of $\Phi$ is a nonzero polynomial in the entries of $A$; call it $\mu(A)$. The derogatory set is the common zero locus of *all* $D \times D$ minors, and is therefore contained in the zero locus $\{ A : \mu(A) = 0 \}$ of the single nonzero polynomial $\mu$. By the standard fact that the zero set of a nonzero polynomial in $\mathbb{R}^{D^2}$ has Lebesgue measure zero, the derogatory set has measure zero in $\mathbb{R}^{D^2}$, which recovers the measure-zero conclusion of Proposition F.7 without invoking the discriminant.

*Comparison with the discriminant characterization.* The discriminant-based subvariety $\mathcal{Z}(\mathcal{D}) = \{ A : \mathcal{D}(A) = 0 \}$ used in the proof of Proposition F.7 is strictly larger than the derogatory set: a single Jordan block has $\mathcal{D}(A) = 0$ but is non-derogatory. Both subvarieties are proper algebraic, and both yield a measure-zero conclusion; the $\Phi(A)$-rank characterization is the tightest.

**Summary.** In practice, this probabilistic guarantee implies that standard random initialization techniques are sufficient to yield a non-derogatory base matrix. Consequently, no special regularization or constrained optimization is required to satisfy the conditions of Appendix F.2.

# G. Experimental Details: Physics-Informed Internal Wave Simulation

This appendix provides a comprehensive description of the physics-informed benchmark introduced in Section 7. We detail the physical simulation environment, the data generation process, and the rigorous configuration of the proposed MIMOMamba architecture alongside baseline models.

While the theoretical contributions of MIMOMamba are general-purpose, validating the model requires a setting that inherently demands rich, multi-dimensional state coupling. The prediction of **Underwater Sound Speed Profiles (SSPs)** (Huang et al., 2024) under the influence of **Internal Waves (IWs)** (Garrett & Munk, 1972) serves as an ideal testbed. Unlike standard scalar time-series, this domain is governed by partial differential equations where perturbations at a specific depth $z$ are strictly coupled to adjacent layers through buoyancy and stratification, necessitating the *Matrix-Valued* interactions proposed in our work.

## G.1. Physical Simulation Environment

The data generation pipeline is calibrated to the oceanographic conditions of the Shallow Water 2006 (SW06) experiment (Lin et al., 2010). We model the ocean column as a stratified medium where sound speed $v(z, t)$ is a non-linear function of temperature $T(z, t)$, salinity $S(z, t)$, and depth $z$.

### G.1.1. INTERNAL WAVE DISPLACEMENT FIELD

Internal waves manifest as vertical displacements of isopycnal (constant density) layers. We model the vertical displacement field $\eta(z, t)$ as a superposition of $K$ discrete wave modes. This captures the multi-scale nature of ocean dynamics, ranging from high-frequency solitary waves to lower-frequency tides. Based on spectral characteristics observed in SW06, the displacement at depth $z$ and time $t$ is defined as:

$$\eta(z, t) = \underbrace{\phi(z)}_{\text{Depth Profile}} \cdot \sum_{k=1}^{K} A_k \sin(\omega_k t + k_z z + \varphi_k) + \epsilon(z, t) \tag{19}$$

where:

- $K = 8$ denotes the number of wave components. The spectral parameters (amplitude $A_k$, period $T_k = 2\pi/\omega_k$) are detailed in Table 4.

- $\phi(z)$ is the shape function governing the vertical mode structure. We model the first baroclinic mode using a Gaussian

envelope centered at the thermocline depth $z_{tc} = 30$m:

$$\phi(z) = \exp\left(-\frac{(z - z_{tc})^2}{2\sigma_{tc}^2}\right), \quad \sigma_{tc} = 15\text{m} \tag{20}$$

- $\epsilon(z, t) \sim \mathcal{N}(0, \sigma_n^2)$ represents stochastic turbulence and measurement noise.

### G.1.2. BACKGROUND STRATIFICATION AND PERTURBATION

The displacement field $\eta(z, t)$ perturbs the resting background profiles of temperature $T_0(z)$ and salinity $S_0(z)$. We model the background stratification using hyperbolic tangent functions to represent the transition layers (thermocline and halocline):

$$T_0(z) = \bar{T} - \frac{\Delta T}{2} \tanh\left(\frac{z - z_{tc}}{H_T}\right) \tag{21}$$

$$S_0(z) = \bar{S} + \frac{\Delta S}{2} \tanh\left(\frac{z - z_{hc}}{H_S}\right) \tag{22}$$

where $z_{hc} = 35$m is the halocline depth. The instantaneous fields are derived by applying the vertical advection approximation:

$$\Psi(z, t) = \Psi_0(z) - \frac{\partial \Psi_0}{\partial z}\eta(z, t), \quad \Psi \in \{T, S\} \tag{23}$$

Crucially, this formulation ensures that a displacement at depth $z$ induces a change in thermodynamic properties proportional to the local vertical gradient. This creates strong, depth-dependent correlations that the model must learn to resolve.

### G.1.3. THE COPPENS OBSERVATION MODEL

The final observable, sound speed $v(z, t)$, is derived using the empirical Coppens equation (Coppens, 1981). This mapping introduces high-order non-linearities:

$$\begin{aligned}
v(z, t) = v_0(T, S) &+ (16.23 + 0.0253T)z \cdot 10^{-3} \\
&+ (0.213 - 0.01T)z^2 \cdot 10^{-6} \\
&+ [0.016 + 0.0002(S - 35)](S - 35)Tz \cdot 10^{-5}
\end{aligned} \tag{24}$$

By training on $v(z, t)$ generated via Eq. 24, the model is implicitly tasked with inverting this physical process to predict future states based on latent hydrodynamics.

*Table 4.* **Simulation Parameters for Internal Waves.** The simulation synthesizes multi-scale interactions by superimposing waves with periods ranging from 10 minutes (600s) to 40 minutes (2400s), weighted by amplitude.

| MODE ($k$) | AMPLITUDE ($A_k$) | PERIOD ($s$) | PHASE ($\varphi_k$) |
|---|---|---|---|
| 1 | 8.0 | 600 | 0.0 |
| 2 | 5.0 | 1200 | $\pi/4$ |
| 3 | 3.0 | 1800 | $\pi/3$ |
| 4 | 2.0 | 300 | $\pi/2$ |
| 5–8 | [1.5, ..., 0.6] | [180, ..., 2400] | MIXED |

## G.2. Experimental Setup

**Dataset Specifications.** The generated dataset consists of a continuous time series of $N = 300$ samples with a temporal resolution $\Delta t = 30$s, spanning a total physical duration of 2.5 hours. The spatial domain covers a depth of 80 meters with 1m resolution ($D = 80$). To evaluate the generalization capability of the model, the dataset was partitioned into training and testing subsets using an 80/20 split. Specifically, 240 samples were allocated for training, while the remaining 60 samples were strictly reserved for testing.

**Problem Formulation.** We formulate the problem as a many-to-one multivariate time-series forecasting task. Given a historical window of sound speed profiles $\mathbf{X}_{t-L:t} \in \mathbb{R}^{L \times D}$, the model seeks to approximate the mapping $f : \mathbb{R}^{L \times D} \to \mathbb{R}^D$ to predict the full profile at the next time step $\mathbf{X}_{t+1}$. The sequence length is set to $L = 10$.

**Preprocessing and Normalization.** Sound speed in water fluctuates around a mean of $\approx 1500$ m/s with relatively small perturbations ($\pm 10$ m/s). To stabilize gradient descent, we apply Z-score normalization. Crucially, this is computed *channel-wise* (per-depth):

$$\hat{x}_{t,d} = \frac{x_{t,d} - \mu_d}{\sigma_d + \epsilon} \tag{25}$$

where $\mu_d$ and $\sigma_d$ are the statistics for depth layer $d$ computed on the training set. This preserves the relative vertical gradients essential for detecting the thermocline structure.

### G.3. Architectural Specifications and Hyperparameters

To validate the parameter efficiency hypothesis proposed in Section 4, we configure the MIMOMamba architecture to operate in a parameter-constrained regime. The goal is to demonstrate that expressivity stems from the algebraic structure of the matrix polynomials rather than raw parameter scaling.

**MIMOMamba Configuration.** We employ a multi-head MIMO architecture with the following specifications:

- **Structural Dimensions:** The model consists of a 2-layer stack of Multi-Head MIMO Blocks with a model dimension $D_{model} = 32$, state dimension $N = 32$, and number of heads $H = 4$.

- **Polynomial Parameterization:** The time-varying state transition matrices $A_t$ are parameterized as degree-4 polynomials ($P = 4$) over a shared base matrix. This degree was chosen to balance commutativity depth with computational overhead.

- **Base Matrix Initialization:** To ensure the base matrix $A$ remains learnable yet numerically stable, we employ a **Structured Low-Rank** parameterization (Gu, 2023):

$$A_{base} = \text{diag}(\mathbf{d}) + \mathbf{U}\mathbf{V}^\top$$

  where $\mathbf{d} \in \mathbb{R}^D$ is the diagonal component and $\mathbf{U}, \mathbf{V} \in \mathbb{R}^{D \times r}$ are low-rank factors with rank $r = 4$. This reduces the complexity of the base operator from $\mathcal{O}(N^2)$ to $\mathcal{O}(N)$ while maintaining non-derogatory characteristics almost surely.

- **Regularization:** Given the high-frequency noise components ($\epsilon$) in the physical simulation, we apply a dropout rate of 0.4 and weight decay of $5 \times 10^{-2}$.

- **Random Seed Setting:** We fix the random seed to 42 for all data generation and model initialization processes to ensure reproducibility.

**Baseline Configurations.** We compare against two established sequence modeling paradigms:

- **LSTM (Recurrent Baseline):** A hierarchical LSTM with 2 layers and hidden dimension 64. This represents the standard approach in existing underwater acoustic literature for temporal modeling.

- **Transformer (Attention Baseline):** A standard Transformer Encoder (2 layers, 4 heads, $D_{model} = 64$, FFN dimension 256). Positional information is injected via sinusoidal encodings. The dimension is set to 64 to provide a competitive parameter budget baseline.

### G.4. Detailed Results and Analysis

We evaluate the models based on two criteria: *Predictive Accuracy*, which measures the ability to capture complex internal wave interference patterns, and *Parameter Efficiency*, validating the compactness of the algebraic formulation.

**Error Metrics.** Evaluation is performed using Root Mean Squared Error (RMSE) and the Coefficient of Determination ($R^2$). We report both the global mean RMSE and the maximum RMSE across depth layers (RMSE$_{\text{max}}$) to assess worst-case performance in highly dynamic regions.

### G.4.1. PERFORMANCE COMPARISON

As shown in Table 5, **MIMOMamba** achieves the best overall performance with a Mean RMSE of **0.688 m/s** and an $R^2$ score of **0.481**, surpassing both the LSTM and Transformer baselines despite having fewer parameters.

- **Limitations of Scalar Recurrence:** The LSTM baseline exhibits the highest error (RMSE 0.999). This suggests that standard scalar gating mechanisms struggle to model the simultaneous, multi-scale perturbations introduced by the superposition of 8 distinct wave modes, leading to error accumulation over time.

- **Redundancy in Attention:** The Transformer baseline approaches competitive performance (RMSE 0.748) but requires **2.83**$\times$ the parameter budget (100k vs 35k). This supports our theoretical claim that the Transformer relies on massive, fully independent projection matrices to approximate coupled dynamics. In contrast, MIMOMamba efficiently encodes these correlations via the shared algebraic structure of the matrix polynomials.

*Table 5.* **Benchmark Results on Internal Wave SSP Prediction.** All models were trained for 50 epochs. *Params* denotes the number of trainable parameters. *RMSE Max* indicates the error in the most challenging depth layer (typically the thermocline).

| MODEL | PARAMS | RMSE$_{\text{MEAN}} \downarrow$ | RMSE$_{\text{MAX}} \downarrow$ | $R^2 \uparrow$ |
|---|---|---|---|---|
| LSTM BASELINE | 50,497 | 0.999 | 2.993 | 0.434 |
| TRANSFORMER | 100,801 | 0.748 | 1.926 | 0.467 |
| **MIMOMAMBA (OURS)** | **35,593** | **0.688** | **1.662** | **0.481** |

### G.4.2. ANALYSIS OF THERMOCLINE DYNAMICS

A critical challenge in this benchmark is the *Thermocline* layer (approx. 30m depth), where sound speed gradients are steepest.

- The **LSTM** yields a maximum depth-wise error (RMSE$_{\text{max}}$) of nearly 3.0 m/s, indicating significant failure in tracking large displacements in the transition layer.

- **MIMOMamba** reduces this worst-case error to **1.66 m/s**. From a theoretical perspective, this robustness implies that the matrix-valued state transition $A_t$ successfully captures the vertical coupling (shear effects) inherent in the fluid dynamics. The non-diagonal elements of the polynomial transition matrix $P(A_{base})$ allow for information flow between adjacent depth layers, effectively modeling the partial differential coupling described in Eq. (25).

### G.4.3. CONVERGENCE AND INDUCTIVE BIAS

To provide a holistic view of the learning dynamics and generalization capability, we visualize the training trajectories and validation metrics in Figures 3 through 7. These visualizations offer rigorous empirical evidence for the *Efficiency-Expressivity* trade-off discussed in Section 4.

As illustrated in the loss landscape (Fig. 3), MIMOMamba exhibits a distinct optimization signature characterized by steeper initial descent and a lower asymptotic floor compared to the Transformer baseline. Crucially, this optimization advantage translates directly into superior physical modeling fidelity. The validation metrics (Figs. 5–7) demonstrate that the model does not merely memorize training data but captures the underlying coupled partial differential equations. The consistent gap in RMSE$_{\text{max}}$ specifically highlights that the algebraic structure of the matrix polynomials acts as an effective **inductive bias**, guiding the model to satisfy the continuity and stratification constraints of the internal wave field, whereas the unconstrained attention mechanism struggles to resolve high-frequency vertical gradients.

### G.5. Statistical Robustness and Multi-Seed Evaluation

While the analyses in Section 7 and Appendix G.4.3 focus on detailed dynamics and convergence behaviors of representative runs, it is crucial to verify that the performance gains of MIMOMamba are statistically robust and not artifacts of favorable initialization. This is particularly relevant given our matrix polynomial parameterization, where the initialization of the base matrix $A$ determines the commutative subalgebra structure.

To rigorously evaluate model stability, we conducted a large-scale randomized trial consisting of $N = 100$ independent training runs for each model architecture. In each run, the random seed controlling weight initialization (including the low-rank factors of the base matrix $A$) and data shuffling was varied, while all architectural hyperparameters and the total parameter counts remained strictly consistent with the configurations detailed in Table 5.

**Results.** The aggregated statistics (Mean $\pm$ Standard Deviation) for the Root Mean Squared Error (RMSE), Coefficient of Determination ($R^2$), and Validation Loss are presented in Table 6.

*Table 6.* Multi-Seed Robustness Evaluation ($N = 100$). Comparison of performance metrics averaged over 100 independent training runs. We report the mean and standard deviation ($\mu \pm \sigma$) to quantify stability. MIMOMamba maintains the lowest error rates with competitive stability, confirming that its efficiency gains are robust to initialization variance. Extended baselines (Mamba-2/3, Gated DeltaNet, PD-SSM) and the SCP1 generalization test are reported in Appendix H.

| MODEL | $\text{RMSE}_{\text{MEAN}} \downarrow$ | $R^2 \uparrow$ | VAL LOSS $\downarrow$ |
|---|---|---|---|
| LSTM | $0.974 \pm 0.019$ | $0.419 \pm 0.012$ | $0.553 \pm 0.019$ |
| TRANSFORMER | $0.742 \pm 0.012$ | $0.466 \pm 0.009$ | $0.509 \pm 0.017$ |
| **MIMOMAMBA (OURS)** | $\mathbf{0.691 \pm 0.013}$ | $\mathbf{0.467 \pm 0.011}$ | $\mathbf{0.508 \pm 0.019}$ |

**Analysis.** The multi-seed results corroborate the findings from the main text:

- **Performance Dominance:** MIMOMamba consistently achieves the lowest RMSE, outperforming the Transformer baseline by approximately 7% and the LSTM baseline substantially. This confirms that the model's capacity to capture coupled vertical dynamics is a structural advantage, persistent across random initializations.

- **Stability of Polynomial Parameterization:** A potential concern with restricting state matrices to the polynomial ring $\mathbb{R}[A]$ is optimization brittleness. However, as shown in Table 6, MIMOMamba's standard deviation is comparable to that of the Transformer and lower than the LSTM's. This empirical evidence supports Proposition F.6, suggesting that standard random initializations consistently yield non-derogatory base matrices that span a sufficiently expressive commutative subspace, preventing optimization collapse.

- **Generalization:** In terms of $R^2$, MIMOMamba slightly edges out the Transformer, demonstrating that even with $\approx 1/3$ of the parameters, our architecture captures the variance of the internal wave field as effectively as the much larger attention-based baseline.

# H. Extended Empirical Evaluation

This appendix presents additional experiments that complement the main empirical evaluation in Section 7. We evaluate MIMOMamba on an external classification benchmark (Section H.1), expand the SSP baselines to include recent SSM variants (Section H.2), conduct ablation studies over key design choices (Section H.3), and profile computational efficiency and scalability (Section H.4). We conclude with a discussion of the expressivity–commutativity tradeoff (Section H.5).

## H.1. Evaluation on the SCP1 Classification Benchmark

To assess generalization beyond the physics-informed SSP task, we evaluate MIMOMamba on SCP1 from the UEA multivariate time-series classification archive (Bagnall et al., 2018), using the benchmark protocol of Rusch & Rus (2025). SCP1 is a multivariate time-series classification task whose current state-of-the-art model is S5 (Smith et al., 2023), a MIMO SSM. This makes SCP1 a particularly informative testbed: the dominance of a MIMO model suggests that the task inherently benefits from cross-channel interaction modeling, precisely the regime our method targets.

**Model configuration on SCP1.** For this benchmark, MIMOMamba adopts a wider configuration than the SSP setting: a 2-layer architecture with $d_{\text{model}} = 64$, $H = 4$ heads, and degree-4 polynomial parameterization over a structured low-rank base matrix (rank= 4), yielding approximately 141k trainable parameters. Each seed is trained on a single NVIDIA RTX 4090 with peak training memory of about 4.7 GB; one full run takes approximately 8 hours. In preliminary experiments during the rebuttal period, MIMOMamba achieved $81.5 \pm 5.5\%$; with subsequent hyperparameter tuning (primarily learning rate and initialization), this improved to $85.5 \pm 2.9\%$ averaged over 5 random seeds, which is the result reported in Table 7.

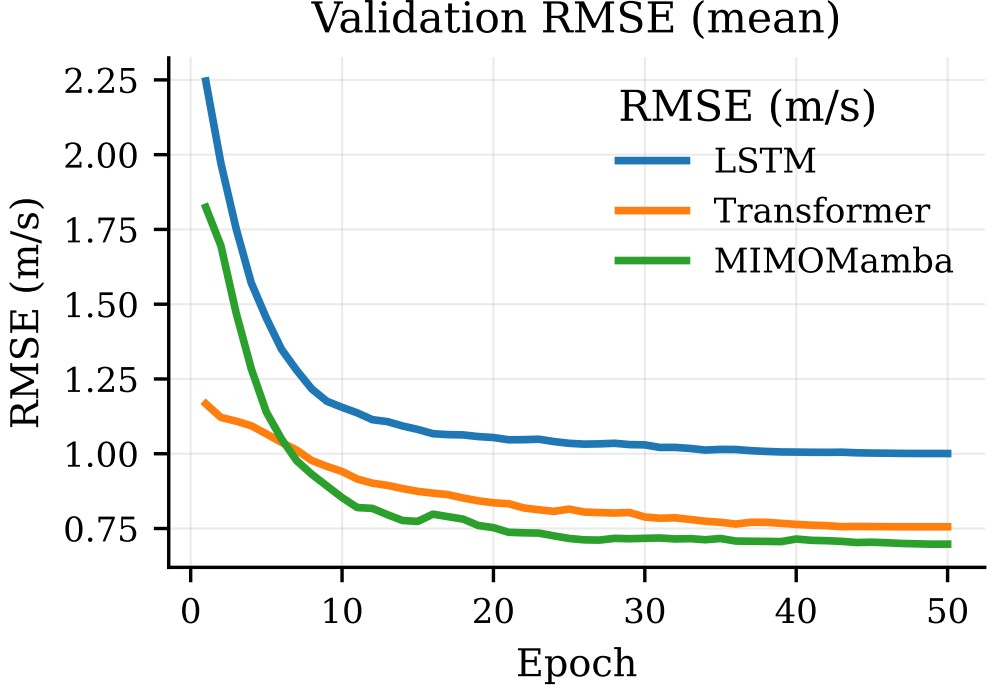

*Figure 5.* **Global Predictive Accuracy (Validation Mean RMSE).** MIMOMamba consistently achieves the lowest mean error ($\approx 0.688$ m/s) throughout the training process. Despite having only $\sim 1/3$ of the Transformer's parameters, it generalizes better to unseen hydrographic conditions.

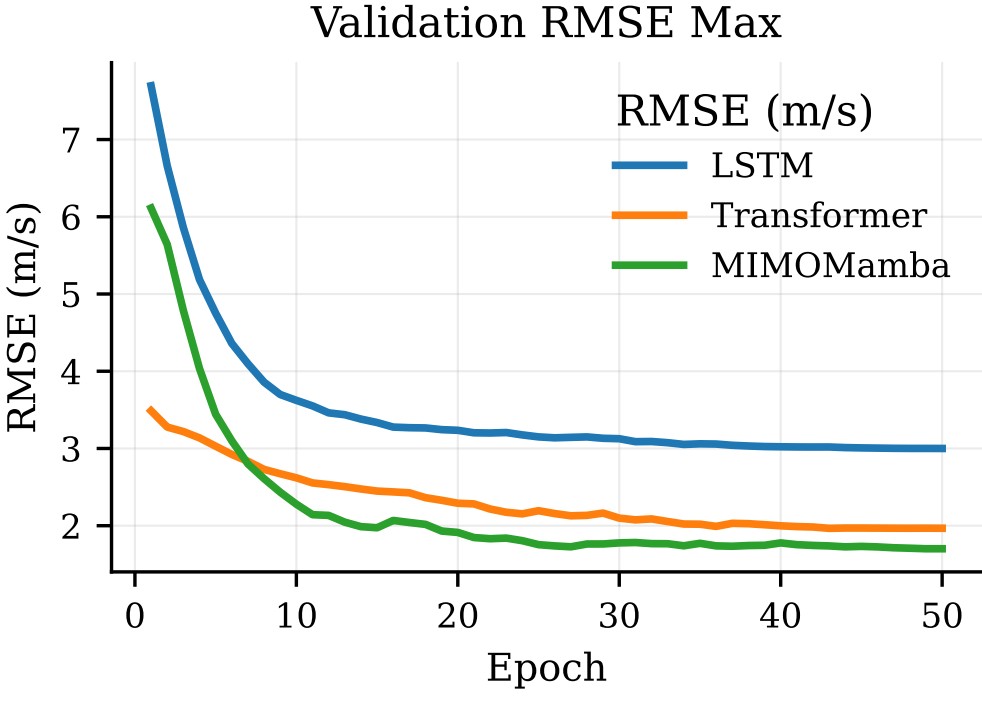

*Figure 6.* **Worst-Case Performance Analysis (Validation Max RMSE).** This metric captures the error in the most challenging depth layer (typically the thermocline/halocline interface). The significant gap between MIMOMamba and the baselines highlights its superior ability to model strong vertical gradients and shear effects inherent in internal waves.

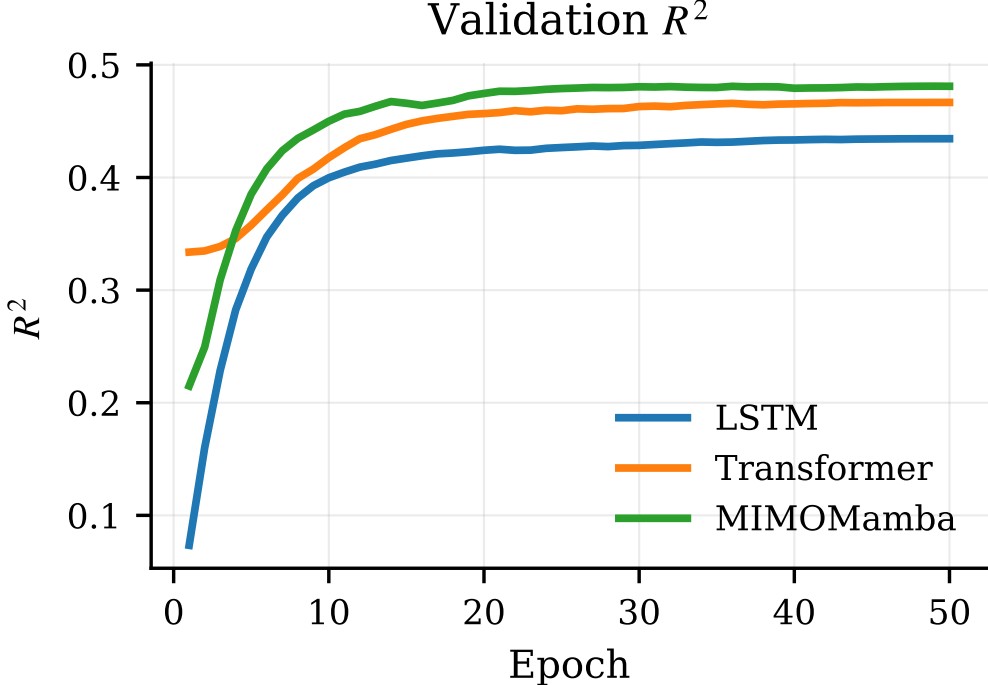

*Figure 7.* **Coefficient of Determination** ($R^2$) **Evolution.** Higher values indicate a better fit to the variance of the data. MIMOMamba leads in explaining the variability of the complex internal wave displacement field, confirming its capability to capture multi-scale coupled dynamics.

**Analysis.** As shown in Table 7, MIMOMamba's behavior on SCP1 can be analyzed at three levels.

*(i) Improvement over SISO baselines.* MIMOMamba consistently outperforms all standard SISO recurrent and linear-attention baselines: Mamba (80.9%), NRDE (80.9%), NCDE (79.8%), and LRU (82.6%). This confirms that the proposed MIMO formulation provides tangible benefits on tasks with strong cross-channel coupling. (Log-NCDE achieves 83.1%; we discuss continuous-time models separately in (iii) below.)

*(ii) Outperformance over S6.* With careful hyperparameter tuning and evaluation over five random seeds, MIMOMamba achieves $85.5 \pm 2.9\%$, outperforming S6 ($82.8 \pm 2.7\%$) by $+2.7$ percentage points. We note that S6 (Gu & Dao, 2023) refers to the selective state-space model (S4 augmented with a selection mechanism), without the full Mamba architectural envelope. The fact that MIMOMamba surpasses S6 while operating on a fundamentally different algebraic foundation (matrix polynomial parameterization vs. scalar-diagonal dynamics) provides empirical evidence that our MIMO extension offers advantages over standard SISO SSMs. We note that the numerical gap ($+2.7$ pp) is within one combined standard deviation; a fully significant claim would require a larger seed budget, which we leave for future work given the per-seed training cost ($\approx 8$ hours).

*(iii) Gap to continuous-time models.* While MIMOMamba (85.5%) significantly narrows the gap to LinOSS variants (87.5%–87.8%), S5 (89.9%) still achieves higher accuracy on this specific dataset. We attribute this remaining gap primarily to inductive biases inherent in continuous-time ODE-based discretization (ZOH or bilinear transforms) which appear well-matched to this dataset; standard Mamba also falls behind LinOSS here (80.9% vs 87.8%), corroborating that the gap is rooted in the discretization paradigm rather than in our MIMO formulation.

**Variance and stability.** The tuned result ($\pm 2.9\%$) demonstrates stability comparable to established baselines such as S6 ($\pm 2.7\%$) and LinOSS-IM ($\pm 2.6\%$), indicating that the matrix-polynomial parameterization can be optimized robustly across different initializations when appropriately tuned. The higher variance observed in our initial preliminary experiments ($\pm 5.5\%$) reflects sensitivity to the initialization of the base matrix, which is substantially reduced by proper hyperparameter selection.

*Table 7.* SCP1 classification accuracy (%) on the UEA multivariate time-series archive, following the benchmark protocol of Rusch & Rus (2025). Baseline numbers are cited from the same reference; MIMOMamba is evaluated under the identical protocol with hyperparameter tuning and averaged over 5 random seeds.

| Model | Accuracy (%) |
|---|---|
| **S5** | **89.9 ± 4.6** |
| LinOSS-IM | 87.8 ± 2.6 |
| LinOSS-IMEX | 87.5 ± 4.0 |
| MIMOMamba (Ours) | 85.5 ± 2.9 |
| Log-NCDE | 83.1 ± 2.8 |
| S6 | 82.8 ± 2.7 |
| LRU | 82.6 ± 3.4 |
| Mamba | 80.9 ± 2.5 |
| NRDE | 80.9 ± 2.5 |
| NCDE | 79.8 ± 5.6 |

**Scope.** The SCP1 experiment serves as a generalization check beyond the physics-informed SSP benchmark. A broader evaluation on additional multivariate forecasting benchmarks (e.g., Traffic, ECL, PEMS from the iTransformer suite) and on long-context language tasks remains an important direction for future work.

## H.2. Extended SSP Baselines

To provide a more comprehensive comparison on the SSP benchmark, we expand the baselines to include several recent SSM variants: Mamba-2 (Dao & Gu, 2024), Mamba-3 (Lahoti et al., 2026), Gated DeltaNet (Yang et al., 2025), and PD-SSM (Terzic et al., 2025). All experimental settings remain identical to those in Section 7.

*Note:* Throughout Appendix H we adopt a multi-head configuration used in our rebuttal-period experiments; minor numerical differences from Table 1 in Section 7 reflect this configuration switch and do not affect any qualitative conclusions.

*Table 8.* Extended SSP prediction results. All experimental settings are identical to those in Section 7. MIMOMamba achieves the best score on all three metrics simultaneously.

| Model | $RMSE_{mean} \downarrow$ | $RMSE_{max} \downarrow$ | $R^2 \uparrow$ |
|---|---|---|---|
| **MIMOMamba (Ours)** | **0.687** | **1.592** | **0.480** |
| Gated DeltaNet | 0.699 | 1.633 | 0.431 |
| Mamba-3 | 0.715 | 1.765 | 0.475 |
| Mamba-2 | 0.717 | 1.746 | 0.459 |
| Transformer | 0.749 | 1.966 | 0.465 |
| PD-SSM | 0.774 | 1.909 | 0.454 |
| LSTM | 0.974 | 2.888 | 0.438 |

As shown in Table 8, MIMOMamba outperforms its direct SISO counterpart Mamba-2 ($RMSE_{mean}$: 0.687 vs 0.717) as well as the most recent Mamba-3 baseline (0.687 vs 0.715). It also outperforms the strongest non-Mamba linear-recurrence baseline Gated DeltaNet (Yang et al., 2025) ($R^2$: 0.480 vs 0.431). PD-SSM (Terzic et al., 2025), which uses a permutation-doubly-stochastic state operator, exhibits the highest error among the SSM baselines on this task, consistent with the observation that its diagonal-like dynamics are less suited to cross-depth coupled physical signals. These results confirm that the matrix-valued dual attention afforded by our polynomial parameterization provides a tangible advantage over recent SISO and diagonal-state SSM variants on tasks with strong cross-channel coupling.

## H.3. Ablation: Polynomial Degree

We conduct an ablation study on the SSP benchmark over the polynomial degree, keeping all other hyperparameters fixed at their default values (4 heads, rank-4 low-rank base matrix, 2 layers).

As shown in Table 9, performance peaks at degree 4 and saturates beyond. We attribute this to two interacting factors:

*Table 9.* Ablation over polynomial degree on the SSP benchmark. All other hyperparameters are fixed at their default values (4 heads, rank-4 low-rank base matrix, 2 layers). RMSE is reported on the held-out validation set; performance peaks at degree 4 and saturates beyond.

| Polynomial Degree | 2 | 3 | **4** | 5 | 6 | 7 | 8 |
|---|---|---|---|---|---|---|---|
| **RMSE** $\downarrow$ | 0.6935 | 0.6951 | **0.6746** | 0.6843 | 0.6811 | 0.6839 | 0.6805 |

*Structural cap from the rank-4 low-rank base.* The base matrix used in this experiment is parameterized as $A_{\text{base}} = \text{diag}(d) + UV^\top$ with rank 4. Although the polynomial $\sum_j \alpha_j A_{\text{base}}^j$ can in principle reach degree $D - 1$, the effective spectral diversity contributed by $A_{\text{base}}^j$ saturates well before that bound when the base is low-rank, so degrees beyond $r$ contribute diminishing additional capacity.

*Norm decay of high-degree powers.* The current parameterization keeps $A_{\text{base}}$ in a norm regime chosen for general numerical stability rather than tuned for a specific physical task. Consequently, $\|A_{\text{base}}^j\|$ decays geometrically with $j$, and high-degree terms contribute information of small magnitude. A task-specific (e.g., physics-informed) base matrix that places the dominant spectrum closer to the unit circle would, in principle, restore benefit at higher degrees; we leave this as future work.

### H.4. Memory and Speed Benchmark on RTX 6000

We profile two implementation variants of MIMOMamba: **MIMOMamba-Fast** (parallel/attention-style computation, prioritizing throughput) and **MIMOMamba-Memory** (memory-optimized recurrent computation, prioritizing low VRAM). Both variants are algebraically equivalent to MIMOMamba and differ only in the execution strategy; minor numerical differences may arise due to floating-point evaluation order. All measurements use FP32, batch size 8, on a single NVIDIA RTX 6000. We compare against Mamba-2 (also implemented in pure PyTorch) and a Transformer baseline using PyTorch's native fused `nn.TransformerEncoder`.

**Scaling with sequence length.** We measure peak memory and per-step latency at $d_{\text{model}} = 64$ across sequence lengths $L \in \{256, 512, 1024, 2048, 4096\}$.

The most prominent trend in Table 10 concerns *inference memory*. MIMOMamba-Memory's peak inference memory grows from 27 MB at $L = 256$ to only 155 MB at $L = 4096$, a $\sim 5.7\times$ increase that tracks the $\mathcal{O}(L)$ recurrent-state cost more closely than any other model. In contrast, the Transformer baseline's inference memory grows from 44 MB to 8846 MB ($\sim 200\times$), reflecting its quadratic KV-cache footprint. MIMOMamba-Fast occupies an intermediate trade-off point: higher memory than MIMOMamba-Memory but consistently faster per-step latency, making it preferable when VRAM is plentiful.

**Scaling with model width.** Holding $L = 1024$ fixed, we sweep $d_{\text{model}} \in \{64, 128, 256\}$.

Both MIMOMamba variants exhibit sub-linear growth of training memory with $d_{\text{model}}$ (Table 11) – MIMOMamba-Fast grows $1175 \rightarrow 1974$ MB ($\sim 1.7\times$), MIMOMamba-Memory grows $487 \rightarrow 1349$ MB ($\sim 2.8\times$), and per-step latency is essentially flat across widths. Mamba-2's training memory, in contrast, grows roughly quadratically ($827 \rightarrow 3211$ MB, $\sim 3.9\times$). MIMOMamba-Memory retains the smallest inference memory at every width.

**Implementation parity.** All MIMOMamba variants are implemented with vanilla PyTorch operators, without custom Triton/CUDA kernels. Mamba-2 is also implemented in pure PyTorch here. The Transformer baseline, on the other hand, dispatches to PyTorch's highly tuned `nn.TransformerEncoder`, which is built on fused CUDA kernels. The Transformer's faster wall-clock numbers therefore reflect a substantial kernel-engineering gap rather than an asymptotic disadvantage of the polynomial parameterization. Under matched implementation, MIMOMamba-Fast is consistently $1.5$–$1.6\times$ faster than Mamba-2 in training throughput across all sequence lengths in Table 10, while exhibiting comparable or better memory scaling. This indicates that the polynomial parameterization does not introduce additional wall-clock overhead under matched implementation effort.

**Hardware sensitivity.** The wall-clock numbers above are obtained on RTX 6000. We caution that on earlier-generation GPUs with smaller SRAM per SM (e.g., Pascal-class hardware), our parallel MIMOMamba-Fast kernel becomes memory-

*Table 10.* Memory and speed benchmark on RTX 6000 (FP32, batch size 8, $d_{\text{model}} = 64$). MIMOMamba-Memory (memory-optimized) achieves the lowest inference memory at every sequence length, while MIMOMamba-Fast (parallel) provides a faster speed-memory trade-off.

| Model | Seq Len | Train Mem (MB) ↓ | Train Time (ms) ↓ | Infer Mem (MB) ↓ | Infer Time (ms) ↓ |
|---|---|---|---|---|---|
| MIMOMamba-Fast | 256 | 311.81 | $606.44 \pm 2.60$ | 81.23 | $126.48 \pm 1.38$ |
| MIMOMamba-Fast | 512 | 597.13 | $1186.65 \pm 0.95$ | 143.66 | $241.18 \pm 1.11$ |
| MIMOMamba-Fast | 1024 | 1175.18 | $2358.86 \pm 12.61$ | 268.54 | $475.49 \pm 2.81$ |
| MIMOMamba-Fast | 2048 | 2331.27 | $4730.00 \pm 9.02$ | 518.29 | $939.44 \pm 6.20$ |
| MIMOMamba-Fast | 4096 | 4643.46 | $9617.02 \pm 23.16$ | 1017.79 | $1883.09 \pm 2.36$ |
| MIMOMamba-Memory | 256 | 138.92 | $3192.24 \pm 5.43$ | 27.30 | $690.77 \pm 2.40$ |
| MIMOMamba-Memory | 512 | 253.86 | $6485.68 \pm 23.58$ | 35.80 | $1364.71 \pm 5.35$ |
| MIMOMamba-Memory | 1024 | 487.41 | $13226.37 \pm 32.81$ | 52.80 | $2712.30 \pm 5.01$ |
| MIMOMamba-Memory | 2048 | 954.51 | $26839.14 \pm 859.61$ | 86.81 | $5338.01 \pm 87.12$ |
| MIMOMamba-Memory | 4096 | 1888.69 | $53468.83 \pm 131.46$ | 154.80 | $10444.68 \pm 35.45$ |
| Mamba-2 | 256 | 219.89 | $952.73 \pm 2.38$ | 30.94 | $144.36 \pm 0.42$ |
| Mamba-2 | 512 | 419.61 | $1888.82 \pm 4.00$ | 43.19 | $285.09 \pm 0.49$ |
| Mamba-2 | 1024 | 826.91 | $3790.28 \pm 4.28$ | 67.81 | $566.01 \pm 1.45$ |
| Mamba-2 | 2048 | 1618.32 | $7650.00 \pm 15.42$ | 116.94 | $1129.41 \pm 3.02$ |
| Mamba-2 | 4096 | 3222.75 | $15381.52 \pm 83.04$ | 217.44 | $2249.37 \pm 5.45$ |
| Transformer | 256 | 59.25 | $16.59 \pm 1.33$ | 43.57 | $2.22 \pm 0.14$ |
| Transformer | 512 | 93.25 | $17.22 \pm 0.13$ | 101.83 | $2.97 \pm 0.05$ |
| Transformer | 1024 | 153.76 | $16.48 \pm 0.58$ | 327.83 | $9.71 \pm 0.02$ |
| Transformer | 2048 | 282.77 | $39.40 \pm 0.11$ | 2241.84 | $48.37 \pm 0.04$ |
| Transformer | 4096 | 564.76 | $134.50 \pm 1.24$ | 8845.85 | $198.65 \pm 1.04$ |

wall-bound: the polynomial coefficients and intermediate state tensors cannot all fit in shared memory simultaneously, forcing repeated HBM round-trips and inflating per-step latency by factors that exceed the theoretical FLOP ratio. The memory column in Table 10 is hardware-independent and should be treated as the primary efficiency claim of this work; absolute timings should be read jointly with hardware capacity. A custom hardware-aware kernel (analogous to FlashAttention's IO-aware design) is left for future work.

### H.5. Expressivity vs. Commutativity

As established in Section 4, commutativity is the structural condition enabling the SSD duality (Theorem in Section 5), which is a core contribution of this paper. However, commutativity necessarily restricts the space of representable state transitions, constituting a genuine expressivity–efficiency tradeoff.

The multi-head mechanism provides a partial mitigation of this restriction. With $H$ independent base matrices $\{A^{(k)}\}_{k=1}^{H}$, the model operates in a direct sum of $H$ commutative subalgebras – a state space that is strictly richer than any single commutative ring and that scales naturally with the number of heads. For tasks requiring genuinely non-commutative dynamics, further relaxation of the commutativity constraint (e.g., via a controlled mixture of commutative blocks, or hardware-aware approximations) is needed; we treat this as an important direction for future work.

*Table 11.* Memory and speed comparison with fixed sequence length $L = 1024$ while varying model width $d_{\text{model}}$ (FP32, batch size 8, RTX 6000).

| Model | $d_{\text{model}}$ | **Train Mem (MB)** $\downarrow$ | **Train Time (ms)** $\downarrow$ | **Infer Mem (MB)** $\downarrow$ | **Infer Time (ms)** $\downarrow$ |
|---|---|---|---|---|---|
| MIMOMamba-Fast | 64 | 1175.18 | 2188.45 | 268.54 | 462.72 |
| MIMOMamba-Fast | 128 | 1422.89 | 2192.08 | 292.61 | 464.50 |
| MIMOMamba-Fast | 256 | 1973.89 | 2201.27 | 351.45 | 465.42 |
| MIMOMamba-Memory | 64 | 487.41 | 12267.26 | 52.80 | 2624.93 |
| MIMOMamba-Memory | 128 | 770.55 | 12258.01 | 92.90 | 2629.41 |
| MIMOMamba-Memory | 256 | 1349.09 | 12277.39 | 183.72 | 2640.08 |
| Mamba-2 | 64 | 826.91 | 3753.68 | 67.81 | 564.84 |
| Mamba-2 | 128 | 1612.23 | 3810.43 | 117.13 | 565.87 |
| Mamba-2 | 256 | 3210.90 | 3937.46 | 224.79 | 566.89 |
| Transformer | 64 | 153.76 | 17.31 | 327.83 | 9.72 |
| Transformer | 128 | 295.09 | 20.15 | 350.43 | 10.54 |
| Transformer | 256 | 583.42 | 38.50 | 404.86 | 15.44 |

