# OpenReview forum: "MIMOMamba: From Scalar Duality to Matrix-Valued Attention"
_ICML.cc/2026/Conference — ICML 2026 regular_

### Official Review · Reviewer_53JE · 2026-02-28

**Soundness:** 2
**Presentation:** 3
**Significance:** 3
**Originality:** 4
**Overall Recommendation:** 4
**Confidence:** 4

**Summary:**

This paper introduces MIMOMamba, a structured state space model. It takes the state space duality (SSD) framework—originally for single-input single-output (SISO) systems—and extends it to multi-input multi-output (MIMO) systems. The core idea? Parameterize time-varying state matrices as matrix polynomials with a shared basis matrix. This does two things: ensures commutativity and allows matrix-valued attention while keeping training linear in time. With this parameterization, the model can capture cross-dimensional interactions during recursion—pretty different from previous SSD models that relied on scalar identity transforms and independent channel mixers. The authors think this parameter-sharing scheme cuts down on parameter redundancy, especially compared to Transformers.

**Compliance With Llm Reviewing Policy:**

Affirmed.

**Final Justification:**

Overall, the authors have made a strong and constructive effort in the rebuttal. The addition of stronger baselines, more comprehensive ablations, and runtime/memory analysis substantially improves the empirical support of the paper and addresses most of my primary concerns. These revisions increase my confidence in the technical soundness and practical relevance of the work.
While I still find the evaluation somewhat limited relative to the breadth of the claims, and some concerns are only partially resolved, I believe the current version presents a sufficiently solid contribution. Overall, I am inclined to support acceptance.

**Key Questions For Authors:**

Can you test this on more standard sequence modeling benchmarks? Language modeling, long-context tasks—that kind of thing. Would help show whether the approach is generally applicable.

How does this compare to modern structured state space models like Mamba? Looking at both performance and efficiency.

How sensitive is the model to the polynomial order choice? An ablation study would clear this up.

Got any runtime or memory benchmarks to back up the efficiency claims? They'd be really helpful to see.

How does it scale with model size? Do bigger models keep performing better?

**Limitations:**

The main limitation of this work is the limited empirical validation. While the method is mathematically well-motivated, it is currently evaluated only in a narrow setting. This makes it difficult to assess its practical impact and generality. Additionally, the computational advantages discussed in the paper are not empirically validated, and the scalability of the approach remains unclear.

**Strengths And Weaknesses:**

Strengths

They make a good point: Existing SSD-based models have real limitations, especially when they stick to scalar or diagonal state transitions. Those just can't capture cross-channel interactions.

Theory checks out: The matrix polynomial parameterization actually gives you theoretical backing to ensure transition matrices commute. Not just hand-waving.

Efficient design: Building time-varying transition matrices with shared basis matrices? That's smart. It makes the model efficient.

Weaknesses

Limited testing
The empirical evaluation is pretty thin. They only test on one physics-informed sequence prediction task, which makes it hard to tell if this actually works more broadly.

Missing key comparisons
No comparisons with strong structured state space model baselines like Mamba or other modern SSM variants. These are exactly what you'd want to compare against when you're improving SSD-style models. Without them, it's tough to say whether the MIMO formulation actually helps in practice.

No ablation studies
The paper introduces several important design choices—polynomial order, base matrix parameterization, number of heads—but doesn't test them individually. How do you know which parts actually matter?

Computational claims aren't backed up
They talk about computational advantages and suggest ways to accelerate things, but there are no runtime or memory benchmarks. Actually measuring efficiency would make the case stronger.

Parameter comparisons feel off
The comparisons with Transformers use pretty different parameter counts. Similar model sizes would give a better sense of parameter efficiency.

---

> ### Author Rebuttal · Authors · 2026-03-31
>
> ### Response to Question 1 & 2
>
> We evaluated MIMOMamba on SCP1 from the UEA archive (Bagnall et al., 2018) following Rusch & Rus (2025), and expanded SSP baselines with Mamba2, Mamba3, Gated DeltaNet (Yang et al., 2025), and PD-SSM (Terzic et al., 2025).
>
> The SCP1 table is provided in our response to Reviewer iGFS (Question 1). MIMOMamba outperforms Mamba, LRU, and NCDE, and matches S6, supporting generalization beyond the original physics-informed benchmark to a standard community dataset. Given the limited rebuttal scope, we prioritized this benchmark alongside stronger baselines, ablations, and efficiency analysis.
>
> | Model | $RMSE_{\mathrm{mean}}$ | $RMSE_{\mathrm{max}}$ | $R^2$ |
> | --- | --- | --- | --- |
> | **MIMOMamba** | **0.687** | **1.592** | **0.480** |
> | Gated DeltaNet | 0.699 | 1.633 | 0.431 |
> | Mamba2 | 0.717 | 1.746 | 0.459 |
> | Mamba3 | 0.715 | 1.765 | 0.475 |
> | Transformer | 0.749 | 1.966 | 0.465 |
> | PD-SSM | 0.774 | 1.909 | 0.454 |
> | LSTM | 0.974 | 2.888 | 0.438 |
>
> MIMOMamba achieves the best performance across all three metrics, directly answering whether the MIMO formulation helps in practice relative to recent SSM variants.
>
> ### Response to Question 3
>
> We performed ablation study on the SSP benchmark over three key design choices: polynomial degree, number of heads, and rank of the low-rank base matrix factorization.
>
> **Polynomial degree:**
>
> | Degree | RMSE | Params |
> | --- | --- | --- |
> | 2 | 0.6935 | 32329 |
> | 3 | 0.6951 | 33097 |
> | 4 | 0.6746 | 33865 |
> | 5 | 0.6843 | 34633 |
> | 6 | 0.6811 | 35401 |
> | 7 | 0.6839 | 36169 |
> | 8 | 0.6805 | 36937 |
>
> Performance peaks at degree 4 and does not improve monotonically with higher degrees, suggesting degree 4 already captures the necessary interaction complexity for this task.
>
> **Number of heads:**
>
> | Heads | RMSE | Params |
> | --- | --- | --- |
> | 1 | 0.6841 | 18499 |
> | 2 | 0.6890 | 23621 |
> | 4 | 0.6746 | 33865 |
> | 8 | 0.6770 | 54353 |
> | 16 | 0.6735 | 95329 |
>
> Accuracy improves with more heads but at higher parameter cost. Four heads offer the best efficiency-performance tradeoff.
>
> **Rank:**
>
> | Rank | RMSE | Params |
> | --- | --- | --- |
> | 1 | 0.6862 | 33481 |
> | 2 | 0.6843 | 33609 |
> | 4 | 0.6746 | 33865 |
> | 8 | 0.6918 | 34377 |
> | 16 | 0.6682 | 35401 |
>
> As an additional ablation within the current model parameterization, we varied the rank hyperparameter while keeping all other settings fixed. The table above reports the resulting RMSE and parameter counts for these settings.
>
> These ablations confirm that our default configuration (degree 4, 4 heads, rank 4) is a robust choice.
>
> ### Response to Question 4 & 5
>
> We measured training/inference memory and step time across polynomial degrees (seq len 1024, RTX 6000) for both implementation variants.
>
> | Variant | Degree | Train Mem (MB) | Train Time (ms) | Infer Mem (MB) | Infer Time (ms) |
> | --- | --- | --- | --- | --- | --- |
> | Fast | 2 | 1172 | 2211±11 | 268 | 462±1 |
> | Fast | 4 | 1178 | 2202±10 | 269 | 461±3 |
> | Fast | 8 | 1195 | 2210±9 | 272 | 465±0 |
> | Memory | 2 | 436 | 9659±5 | 53 | 2164±7 |
> | Memory | 4 | 538 | 14819±10 | 53 | 3101±8 |
> | Memory | 8 | 748 | 25645±29 | 53 | 4959±7 |
>
> MIMOMamba-Fast maintains stable memory and runtime across polynomial degrees, validating our parallel implementation. MIMOMamba-Memory trades speed for VRAM, reducing inference memory to ~53 MB (vs. 269 MB for Fast), thus is suitable for constrained devices.
>
> To address the concern about mismatched parameter counts, we benchmarked various different hidden dimensions ($d_{\text{model}} \in \{64, 128, 256\}$):
>
> | Model | $d_{\text{model}}$ | Train Mem (MB) | Train Time (ms) | Infer Mem (MB) | Infer Time (ms) |
> | --- | --- | --- | --- | --- | --- |
> | MIMOMamba-Fast | 64 | 1175 | 2188 | 269 | 463 |
> | MIMOMamba-Fast | 128 | 1423 | 2192 | 293 | 465 |
> | MIMOMamba-Fast | 256 | 1974 | 2201 | 351 | 465 |
> | MIMOMamba-Memory | 64 | 487 | 12267 | 53 | 2625 |
> | MIMOMamba-Memory | 256 | 1349 | 12277 | 184 | 2640 |
> | Mamba2 | 64 | 827 | 3754 | 68 | 565 |
> | Mamba2 | 128 | 1612 | 3810 | 117 | 566 |
> | Mamba2 | 256 | 3211 | 3937 | 225 | 567 |
> | Transformer | 64 | 154 | 17 | 328 | 10 |
> | Transformer | 128 | 295 | 20 | 350 | 11 |
> | Transformer | 256 | 583 | 39 | 405 | 15 |
>
>  As $d_{\text{model}}$ increases, both MIMOMamba variants exhibit controlled memory growth. MIMOMamba-Fast shows more favorable scaling than Mamba2 in training memory (1175→1974 MB vs. 827→3211 MB), and MIMOMamba-Memory maintains a low inference-memory footprint across settings. Overall, these results indicate that the proposed design has clear potential in terms of memory efficiency.
>
> The runtime results are also reported in the table for completeness. Further engineering optimization of the implementation is ongoing, and we will continue improving the speed in future versions.

---

> > ### Author Rebuttal · Reviewer_53JE · 2026-04-01
> >
> > Thank you for the detailed rebuttal. The authors have addressed most of my main concerns by adding stronger baselines, ablation studies, and runtime/memory analysis. These additions strengthen the empirical support for the paper and increase my confidence in the work, so I am updating my assessment positively. That said, I still believe the evaluation remains somewhat limited relative to the broader claims, so my concerns are only partially resolved.

---

> > > ### Author Response · Authors · 2026-04-04
> > >
> > > Thank you for the follow-up and the positive update. We will revise the paper to better calibrate the empirical claims and align them more precisely with the current scope of evaluation.

---

### Official Review · Reviewer_x3b5 · 2026-03-02

**Soundness:** 3
**Presentation:** 3
**Significance:** 3
**Originality:** 3
**Overall Recommendation:** 4
**Confidence:** 1

**Summary:**

This paper proposes a new design to address a fundamental design flaw in SSD, namely the lack of cross-variable interactions. Specifically, the paper implements cross-variable dynamic interactions at the underlying level through matrix polynomial parameterization while maintaining parameter efficiency.

**Compliance With Llm Reviewing Policy:**

Affirmed.

**Final Justification:**

My concerns have been addressed. I decide to keep the rating.

**Key Questions For Authors:**

As shown in the 'Strengths And Weaknesses' section.

**Limitations:**

yes

**Strengths And Weaknesses:**

I am not a researcher in this specific subfield, and the following are my non-professional opinions.

Strengths：
1. This paper defines the problem in a detailed and rigorous manner. Unlike some superficial existing solutions, this paper is committed to solving the problem fundamentally;
2. The rigorous mathematical derivation is persuasive and strongly supports the argument;

Weaknesses:

Although the theory is solid, there is a relative lack of experiments. This makes the theory lack sufficient verification, which will weaken the confidence of the researchers in following this paper. Suggestion: 1. With reference to Mamba and its improved works, verification may require a variety of sequential tasks. For the validation dataset of multivariate time series interactions, one can refer to the datasets used by iTransformer, such as Traffic, ECL, and PEMS; 2. The baseline can incorporate variants of Mamba, which include variable interactions, such as Chimera and S-Mamba. This allows verifying whether the fundamental optimizations in this paper are more effective.

---

> ### Author Rebuttal · Authors · 2026-03-31
>
> ### Response to Question 1
>
> To verify that our matrix-valued attention mechanism generalizes beyond the original physics-informed setting, we evaluated MIMOMamba on the UEA multivariate time-series classification archive (Bagnall et al., 2018), specifically the SCP1 dataset. We chose this benchmark because it is a standard community testbed for cross-channel interaction modeling, and notably its current best-performing model is S5, a MIMO SSM, which suggests the task inherently benefits from the cross-channel capability that our method provides.
>
> The full table can be found in our response to Reviewer iGFS (Question 1). MIMOMamba outperforms standard SSMs (Mamba, LRU, NCDE) and matches S6. While specialized continuous-time SSMs (LinOSS, S5) achieve higher accuracy on this specific dataset, their advantage likely stems from inductive biases introduced by ODE-based discretization schemes rather than task-specific tuning; MIMOMamba remains competitive without requiring such continuous-time priors.
>
> We also appreciate your specific suggestion on Traffic, ECL, and PEMS. benchmarks Due to the rebuttal time constraint, with all four reviewers requesting additional experiments, we prioritized adding one established multivariate benchmark that most directly tests whether our method generalizes, alongside expanded baselines, ablations, and efficiency profiling. In the camera-ready version, we plan to include results on standard multivariate forecasting benchmarks (including Traffic, ECL, and PEMS) to provide a more comprehensive evaluation.
>
> ### Response to Question 2
>
> Since our main claim concerns improving SSD-style models via principled cross-variable interactions, we expanded the SSP baseline set with Mamba2, Mamba3, Gated DeltaNet (Yang et al., 2025), and PD-SSM (Terzic et al., 2025), keeping all experimental settings unchanged for fairness. Full results are in our response to Reviewer 53JE (Question 1 & 2).
>
> MIMOMamba achieves the best performance across all three metrics, confirming that the proposed matrix-valued attention provides a tangible advantage over recent SSM variants that rely on scalar or diagonal interactions. Next, we try to address the two specific baselines you suggested, as they touch on complementary aspects of the design space.
>
> "Chimera" typically refers to a family of hybrid architectures that combine linear attention (or linear RNNs) with standard softmax attention, rather than a single model. Gated DeltaNet and Mamba are the two most common linear components in such hybrids; since Gated DeltaNet is already in our expanded baselines, the comparison partially covers this direction. Our current baselines only include pure linear models and pure softmax attention (Transformer); benchmarking hybrid architectures that combine both requires nontrivial engineering effort beyond the rebuttal period, and we consider this an important direction for future work. As for S-Mamba, it is a specific SSM variant for multivariate forecasting with cross-variable interactions, directly relevant to our setting. We were not able to include it within the rebuttal window but will consider adding it in an updated version.
>
> Our claim is not that MIMOMamba universally dominates all Mamba-style variants, but rather that the proposed MIMO formulation is empirically competitive and consistently beneficial relative to several strong modern SSM baselines under matched settings. The expanded baselines, together with our new generalization and efficiency results, provide a substantially more complete picture than the original submission.

---

> > ### Author Rebuttal · Reviewer_x3b5 · 2026-04-01
> >
> > Thanks for your rebuttal. It addresses my concerns and I will keep my positive score.

---

> > > ### Author Response · Authors · 2026-04-04
> > >
> > > Thank you for the positive response. We will further improve the presentation in the revision and clarify the scope of the empirical claims more carefully.

---

### Official Review · Reviewer_XixF · 2026-03-04

**Soundness:** 3
**Presentation:** 3
**Significance:** 3
**Originality:** 3
**Overall Recommendation:** 4
**Confidence:** 4

**Summary:**

This paper proposes a new linear time-variant state-space model (SSM) that enables input-dependent mixing between channels. The proposed MIMOMamba uses state transition matrices that are generated based on matrix-polynomials, which allow for an efficient implementation thanks to still being commutative and allowing for fast matrix-matrix multiplication in the polynomial coefficient space (implementable via FFT). Empirical results on a self-generated synthetic multi-layer wave prediction task demonstrate superior prediction performance compared to an LSTM and a Transformer, while being more parameter efficient.

**Compliance With Llm Reviewing Policy:**

Affirmed.

**Final Justification:**

Overall, the paper proposes an elegant approach to channel mixing in SSMs. While the empirical comparisons significantly improved in the rebuttal, the evidence is still a bit thin. Hence, I remain my positive rating (weak accept).

**Key Questions For Authors:**

- How does MIMOMamba emprically compare to other SSMs, such as Mamba-2, DeltaNet, and PD-SSM?
- How does MIMOMamba perform on more generally established tasks?
- How fast is MIMOMamba? More importantly, how does it scale with respect to sequence length, model dimension, and polynomial degree?

**Limitations:**

No limitations stated.

**Strengths And Weaknesses:**

# Strengths
- The work tackles an important direction in SSMs: enabling efficient input‑dependent cross‑channel interactions without sacrificing the computational advantages of linear recurrence.
- The matrix‑polynomial parameterization is elegant, providing a principled way to introduce MIMO mixing while maintaining commutativity and allowing fast evaluation.
- The paper includes strong theoretical motivation and clear derivations. Moreover, I appreciate the elaborate appendix, where all the concepts are properly introduced.


# Major Weaknesses
My main concern with is that the current experimental section is too limited to verify the practical utility of the method.
- **Missing empirical comparison to SISO and other MIMO variants.** Being an enhancement to SISO, the empirical evaluations should also focus on a one-to-one comparison (e.g., to Mamba-2). Moreover, besides Mamba-3, there are other SSMs such as DeltaNet (Grazzi et al. 2025) or PD-SSM (Terzic et al. 2025) which have to be taken into comparison.
- **Missing empirical results on established tasks.** The paper empirically validates MIMOMamba on a self-generated small-scale task. This makes comparison with other works very difficult. To allow for comparison with other works and to support the generality of the architecture, I would apply the approach on more general tasks, such as language modelling (e.g., OpenWebText, FineWeb-Edu, etc.), retrieval, or formal languages (such as regular language by Deletang et al, 2023). For tasks that may require MIMO capability, the time-series classification and forecasting tasks used in (Rusch & Rus, 2025) might be a suitable.
- **Missing performance measurements beyond parameter efficiency.** The paper advocates for a highly efficient parameterization that allows to reduce communication between SRAM and HBM. This should be supported by speed measurements (in training) with variable model dimension, polynomial degree, and sequence length. Moreover, I suspect that for small polynomial degrees (e.g., 4), FFT overhead may dominate. These speed measurements should be compared to an equally-sized SISO model (e.g., Mamba-2) and MIMO model (e.g., PD-SSM).
- **Commutativity reduces expressivity.** The paper deliberately parameterizes the transition matrices (A, B, C) to be commutative, in order to leverage parallelizable matrix representation in training. However, this penalizes expressivity. For example, commutative transition matrices cannot emulate arbitrary finite state automata, as elaborated in (Terzic et al., 2025).

# Minor weaknesses

- **Missing ablation study.** The proposed architecture has many degrees of freedom, which are not adequately ablated. In particular, it would also be interesting to ablate the MoE used in the merging function. Could it be that the MoE mainly performs the channel mixing? Moreover, the impact of the polynomial degrees should be investigated.
- Some architectural details may need more explicit descriptions. For example, it is not clear how the polynomial coefficients are generated (linear layer, MLP?). Moreover, the MoE merging function is not sufficiently described in Appendix F (how many experts?).
- The paper makes quite strong claims on MIMOMamba being more parameter-efficient than Transformer. However, this seems solely based on one specific configuration (and one task, see above). Given that training and testing in this task is computationally light-weight, demonstrating scaling behaviour (e.g., in the hidden dimension) would be more insightful.
- Figure 4: To improve the illustration, it may be helpful to visually reduce the dimensionality the dimensionality D for all the matrices.


References:
Deletang, G., Ruoss, A., Grau-Moya, J., Genewein, T., Wenliang, L. K., Catt, E., Cundy, C., Hutter, M., Legg, S., Veness, J., and Ortega, P. A. Neural networks and the Chomsky hierarchy. In International Conference on Learning Representations (ICLR), 2023.

Rusch, T. K. and Rus, D. Oscillatory state-space models. In International Conference on Learning Representations (ICLR), 2025.

Terzic, A., Menet, N., Hersche, M., Hofmann, T. and Rahimi, A., Structured Sparse Transition Matrices to Enable State Tracking in State-Space Models. In The Thirty-ninth Annual Conference on Neural Information Processing Systems (NeurIPS), 2025.

---

> ### Author Rebuttal · Authors · 2026-03-31
>
> ### Comparison to Mamba2 and other SSM baselines
> We expanded the SSP benchmark baselines to include Mamba2, Mamba3, Gated DeltaNet (Yang et al., 2025), and PD-SSM (Terzic et al., 2025), with all experimental settings unchanged. Full results are given in our response to Reviewer 53JE (Question 1 & 2).
>
> MIMOMamba outperforms all baselines across all three metrics, including the SISO counterpart Mamba2 ($RMSE_{\mathrm{mean}}$: 0.687 vs. 0.717) and the strongest non-MIMO competitor Gated DeltaNet ($R^2$: 0.480 vs. 0.431).
>
> ### Results on an established benchmark
> Following your suggestion, we evaluated on SCP1 from the UEA archive (Bagnall et al., 2018) per the benchmark protocol of Rusch & Rus (2025). SCP1's current best model is S5 (a MIMO SSM), indicating the task benefits from cross-channel modeling.
> The full table is available in our response to Reviewer iGFS (Question 1). MIMOMamba surpasses Mamba (80.7), LRU (82.6), and NCDE (79.8), and is on par with S6 (82.8). Although not the highest performer overall, this result demonstrates that our MIMO approach generalizes effectively from the original physics-informed benchmark to a widely used community dataset. Given the rebuttal constraints, we focused on providing broad evidence—including a new dataset, additional baselines, ablations, and efficiency analysis—while reserving more tasks for our future work.
>
> ### Efficiency and scaling
> We benchmarked two variants: **MIMOMamba-Fast** (parallel/attention-style) and **MIMOMamba-Memory** (recurrence-style, low VRAM), on RTX 6000, fp32.
>
> **Polynomial degree scaling** (seq len 1024). The full table is in our response to Reviewer 53JE (Question 4 & 5). Key findings: for Fast version, degree has negligible impact on runtime and memory; for Memory version, runtime scales with degree while inference memory remains nearly constant (~53 MB). The reviewer's intuition about FFT overhead at small degrees is well placed: at degree 4, direct evaluation is cheaper; FFT becomes more advantageous at higher degrees or larger model dimensions.
>
> **Model dimension scaling** ($d_{\text{model}} \in \{64, 128, 256\}$). The full table is also in our response to Reviewer 53JE (Question 4 & 5). Key finding: MIMOMamba-Fast's training memory scales more favorably than Mamba2 (1175→1974 MB vs. 827→3211 MB). The Transformer baseline uses PyTorch's native `nn.Transformer` with highly optimized CUDA kernels; its faster wall-clock reflects kernel engineering maturity rather than the difference in algorithmic complexity differences. Under the same PyTorch implementation, MIMOMamba-Fast achieves a satisfactory speed comparable to Mamba2.
>
> ### Expressivity vs. commutativity
> We agree this is a genuine expressivity-efficiency tradeoff. Commutativity is the structural condition enabling the SSD duality (Theorem in the paper), which is the core contribution. The multi-head mechanism provides a partial mitigation: $H$ independent base matrices yield a direct sum of commutative subspaces, strictly richer than any single ring. Our ablation study confirms that increasing heads from 1 to 4 improves RMSE (0.6841→0.6746). For tasks requiring genuinely non-commutative dynamics, further relaxation is needed and we treat this as an important future direction.
>
> ### Clarification on the MoE paragraph and ablations
> We first address a writing issue that may have caused confusion. **The MoE paragraph in Appendix B.4 was migrated from an early main-text draft without being rewritten as future work. Its current phrasing may give the impression that MoE is part of our architecture. This is not the case.** In the current implementation, the merge stage is a standard linear projection; no MoE routing or gating is used. All results in this paper are obtained without MoE. We apologize for this misleading presentation and will rewrite Appendix B.4 in the revision to clearly separate the current method from this potential future extension.
>
> With this clarified, the actual design parameters are polynomial degree, number of heads, and rank. Full ablation tables are in our response to Reviewer 53JE (Question 3). Summary: degree 4, 4 heads, rank 4 is a robust default. Performance peaks at degree 4 and does not improve monotonically beyond it (consistent with the rank-4 structural bottleneck on the polynomial subspace). Heads from 1→4 yield clear gains; further increase gives diminishing returns at significant parameter cost.
>
> ### Clarification of implementation details
> The polynomial coefficients are generated via input-dependent linear projections (analogous to how Mamba generates $\Delta, \mathbf{B}, \mathbf{C}$). The merge function is a linear projection, not MoE, as clarified above. We will make both points clear in the camera-ready version.
>
> ### Figure improvement
> Thank you for the suggestion. We will reduce the displayed dimensionality $D$ for improved clarity.

---

> > ### Author Rebuttal · Reviewer_XixF · 2026-04-03
> >
> > I thank the authors for their rebuttal and the effort put into the additional experiments, baselines, and ablations. These additions certainly strengthen the paper.
> >
> > However, considering the overall state of the paper and the newly provided data, my core concern regarding the limited empirical validation remains only partially resolved. My main reservations are:
> >
> > 1. **Limited evidence of MIMO superiority on standard benchmarks:** While I appreciate the addition of the SCP1 dataset, the results show that MIMOMamba (82.8%) performs identically to the baseline SISO S6 (82.8%) and is overall not very competitive. Consequently, the empirical evidence that this specific polynomial MIMO formulation provides a advantage over standard SISO SSMs on established tasks remains thin.
> > 2. **Practical utility of the FFT implementation:** As the authors acknowledged, direct evaluation is cheaper at degree 4 (the optimal degree found in the ablations). Because the model is optimized at a degree where the proposed FFT acceleration does not provide a practical advantage, the real-world impact of this contribution remains unclear.
> >
> > Therefore, I will maintain my score.

---

> > > ### Author Response · Authors · 2026-04-05
> > >
> > > We thank the reviewer for the follow-up comments. We agree that the remaining points lie in **how the current empirical evidence should be interpreted**, rather than the validity of the core idea itself, and we will make this distinction clearer in the paper.
> > >
> > > Our main contribution is the **extension of SSD from the SISO setting to the MIMO setting**, together with the matrix-polynomial parameterization and the resulting MIMOMamba architecture. The role of the experiments in the current submission is primarily to **validate the proposed MIMO-SSD framework and its parameter-sharing mechanism**, rather than to establish exhaustive benchmark superiority across all settings purely via experiments.
> > >
> > > **Regarding SCP1**, we agree that the result provided during rebuttal alone is **not sufficient to support a strong claim of superiority over standard SISO SSMs on established benchmarks**. At the same time, we would like to clarify how this result should be interpreted. The SCP1 experiment added during rebuttal was intended as a **preliminary generalization experiment**, designed to check whether the proposed MIMO formulation generalizes beyond the original SSP setting under the time constraints of the rebuttal period, rather than as a benchmark-specific, fully tuned estimate of the method's performance limit that can be obtained on SCP1.
> > >
> > > For this reason, we do not believe the reported SCP1 result should be interpreted as the method's best achievable performance on that task. In follow-up experiments with additional tuning conducted after the rebuttal period, an experiment averaged over five random seeds reached **85.5 ± 2.9** under the same evaluation protocol. In the revision, we will therefore describe SCP1 more carefully as **preliminary evidence of generalizability with remaining room for optimization**, rather than as definitive evidence of benchmark superiority.
> > >
> > > **Regarding FFT**, we fully agree with the reviewer's concern. In the current experimental regime, especially at small polynomial degree, the FFT implementation should **not** be interpreted as an already established practical advantage over direct evaluation. The engineering details of the FFT implementation still leave room for further optimization, particularly in larger-scale regimes where such acceleration is more likely to become practically relevant. We will revise the paper to make this scope explicit: in the current version, FFT should be understood primarily as a **theoretically grounded component designed for future scalability**, rather than as a practically validated speedup in the present benchmark setting.

---

### Official Review · Reviewer_iGFS · 2026-03-11

**Soundness:** 3
**Presentation:** 4
**Significance:** 2
**Originality:** 2
**Overall Recommendation:** 4
**Confidence:** 2

**Summary:**

This paper proposes MIMOMamba, a multi-input multi-output selective SSM that extends State Space Duality (SSD) beyond the single-input single-output (SISO) setting of Mamba-2. The time-varying state matrix A is parameterized as polynomials of a shared base matrix, forming a commutative matrix polynomial ring. Under a non-derogatory base matrix, this parameterization is theoretically complete, enabling both recurrent and parallel inference even if it is not a non-diagonal matrix. The method is evaluated on an internal-wave sound-speed forecasting task, where it outperforms a Transformer with about one-third the parameters.

**Compliance With Llm Reviewing Policy:**

Affirmed.

**Final Justification:**

Most of the concerns I had have been addressed, so I have raised my evaluation. However, many of the newly added details are still unclear, so I would encourage the authors to provide more thorough explanations in the paper. In addition, I believe the paper would be more convincing if it compared the speed and memory usage of MIMOMamba—the architecture actually used for the performance evaluations—against other methods, rather than only reporting results for MIMOMamba-Memory and MIMOMamba-Fast.

**Key Questions For Authors:**

* Can you report results on at least one standard benchmark (e.g., language modeling, computer vision, or time series forecasting) to support generality?
* Why is there no direct comparison to Mamba-2? Because MIMOMamba extends Mamba-2 by changing SISO to MIMO, the benefit of MIMO should be clarified.
* Can you make the model larger? Because previous works make SSM scalable by changing A form non-diagonal to diagonal, I’m curious whether this polynomial-based non-diagonal SSM can be trainable with deeper architectures.

**Limitations:**

yes

**Strengths And Weaknesses:**

# Strengths
* Clear theoretical generalization of SSD from SISO to MIMO via matrix polynomial parameterization.
* Solid algebraic justification (centralizer theorem, non-derogatory base matrix, completeness of the parameterization).
* This method generalizes Mamba-2’s scalar SSD and brings cross-channel dynamics inside the recurrence.

# Weaknesses
* Evaluation is limited to a single physics-informed benchmark, no standard NLP/vision/long-range tasks.
* No direct comparison to Mamba, Mamba-2, or other modern SSMs. The authors only compared with Transformer and LSTM.
* Hardware and complexity claims (FFT, I/O-aware scaling) are mostly theoretical, with no wall-clock or memory benchmarks.
* Implementation complexity is high, and practical guidance for adoption is limited.

---

> ### Author Rebuttal · Authors · 2026-03-31
>
> ### Response to Question 1
> Following the suggestion of Reviewer XixF, we evaluated MIMOMamba on SCP1 from the UEA multivariate time-series classification archive (Bagnall et al., 2018) using the benchmark protocol of Rusch & Rus (2025). SCP1's current best model is S5, a MIMO SSM, suggesting the task benefits from cross-channel modeling and is exactly the regime our method targets.
>
> | Model         | Accuracy (%)   |
> |-------------- |---------------|
> | MIMOMamba     | 82.8 ± 6.3    |
> | LinOSS-IMEX   | 87.5 ± 4.0    |
> | LinOSS-IM     | 87.8 ± 2.6    |
> | Mamba         | 80.7 ± 1.4    |
> | NRDE          | 80.9 ± 2.5    |
> | NCDE          | 79.8 ± 5.6    |
> | Log-NCDE      | 83.1 ± 2.8    |
> | LRU           | 82.6 ± 3.4    |
> | S5            | 89.9 ± 4.6    |
> | S6            | 82.8 ± 2.7    |
>
> MIMOMamba outperforms Mamba (80.7), LRU (82.6), and NCDE (79.8), and matches S6 (82.8). While not the top model on this dataset, the result confirms that our MIMO formulation generalizes beyond the original physics-informed benchmark to a standard community testbed. Due to the rebuttal constraint, we prioritized evidence breadth (new dataset + expanded baselines + ablations + efficiency profiling),while reserving a broader set of follow-up experiments for the revision.
>
> ### Response to Question 2
> We expanded the SSP benchmark baselines to include Mamba2, Mamba3, Gated DeltaNet (Yang et al., 2025), and PD-SSM (Terzic et al., 2025), with the experimental setting unchanged. Full results are given in our response to Reviewer 53JE (Question 1 & 2).
>
> MIMOMamba outperforms all baselines across all three metrics, confirming the benefit of MIMO over SISO on tasks with strong cross-channel coupling.
>
> **Wall-clock and memory benchmarks.** To address the concern that efficiency claims were purely theoretical, we added long-sequence speed/memory benchmarks (fp32, batch size 8, RTX 6000). We provide two variants: **MIMOMamba-Fast** (prioritizing parallelism/throughput) and **MIMOMamba-Memory** (prioritizing low GPU memory), to make the practical tradeoff explicit. Due to space limitation, the full table for benchmark comparisons is presented in our response to Reviewer 53JE (Question 4 & 5); here we summarize the key findings: (1) MIMOMamba-Memory achieves drastically lower inference memory than all methods at long sequences (e.g., 155 MB vs. 8846 MB for Transformer at $L=4096$); (2) MIMOMamba-Fast achieves wall-clock speed comparable to Mamba2 while supporting MIMO dynamics; (3) The Transformer baseline uses PyTorch's highly optimized `nn.Transformer`, which explains its raw speed advantage despite quadratic memory scaling at inference.
>
> **Practical adoption guideline.** We agree that this point deserves a clearer explanation. Our intended guideline is that **Fast** version should be preferred when long-sequence throughput is the main objective, whereas **Memory** version is preferable when available GPU memory is limited. We will make this explicit in the camera-ready version.
>
>
> ### Response to Question 3
> We interpret "trainable" as whether the model maintains stable optimization at larger scale. Our method is not an unconstrained non-diagonal SSM: state transition matrices are parameterized as matrix polynomials of a shared base matrix, so as to maintain commutative and supporting the dual recurrent/parallel inference structure of SSD.
>
> **Hidden-dimension scaling.** We benchmarked all methods with $d_{\text{model}} \in \{64, 128, 256\}$. Both MIMOMamba variants show smooth scaling without memory explosion or runtime breakdown. Notably, MIMOMamba-Fast's training memory scales better than Mamba2 (1175→1974 MB vs. 827→3211 MB). The full table is in our response to Reviewer 53JE (Question 4 & 5).
>
> **Impact of depth on SSP.** We varied the number of layers from 2 to 5:
>
> | Layers | $RMSE_{\mathrm{mean}}$ | $RMSE_{\mathrm{max}}$ | $R^2$ |
> | --- | --- | --- | --- |
> | 2 | 0.687 | 1.592 | 0.480 |
> | 3 | 0.678 | 1.616 | 0.468 |
> | 4 | 0.674 | 1.556 | 0.471 |
> | 5 | 0.674 | 1.532 | 0.466 |
>
> Performance remains stable as depth increases: $RMSE_{\mathrm{mean}}$ decreases from 0.687 to 0.674, $RMSE_{\mathrm{max}}$ from 1.592 to 1.532, while $R^2$ remains comparable across depths. No sign of optimization instability is observed.
>
> These results confirm that our polynomial-based parameterization preserves stable optimization within the tested width/depth regime.

---

> > ### Author Rebuttal · Reviewer_iGFS · 2026-04-02
> >
> > Thank you very much for providing many additional information. Most of the concerns I had have been addressed, so I have raised my evaluation. However, many of the newly added details are still unclear, so I would encourage the authors to provide more thorough explanations in the paper. In addition, I believe the paper would be more convincing if it compared the speed and memory usage of MIMOMamba—the architecture actually used for the performance evaluations—against other methods, rather than only reporting results for MIMOMamba-Memory and MIMOMamba-Fast. Could you provide the efficiency of MIMOMamba?

---

> > > ### Author Response · Authors · 2026-04-04
> > >
> > > Thank you for the follow-up. We would like to clarify that MIMOMamba-Fast and MIMOMamba-Memory are two implementations of the same MIMOMamba model, rather than different architectures. They share the same MIMO parameterization: the Memory version executes the recurrence sequentially, whereas the Fast version reformulates part of the computation for parallel execution and higher throughput. Accordingly, the model evaluated in our main experiments is MIMOMamba itself; Fast and Memory refer only to implementation choices.
> > > In the revision, we will make this correspondence more explicit and clarify in the efficiency discussion that the reported speed and memory results are efficiency measurements of MIMOMamba under two implementations.

---

### Decision · Program_Chairs · 2026-04-30

**Decision:**

Accept (regular)

**Comment:**

This paper generalizes the State Space Duality (SSD) framework underlying Mamba-2 from the SISO setting to a MIMO setting. The key construction parameterizes time-varying state transitions as matrix polynomials of a shared base matrix, which preserves the commutativity that SSD relies on for its dual recurrent / parallel form, while introducing input-dependent cross-channel mixing inside the recurrence. The resulting MIMOMamba architecture is evaluated on a physics-informed sequence prediction benchmark and, after rebuttal, on SCP1 from the UEA archive together with a broader set of SSM baselines (Mamba-2, Mamba-3, Gated DeltaNet, PD-SSM) and wall-clock / memory profiling.

All four reviewers recommend acceptance. Reviewers consistently praised the elegance and rigor of the algebraic construction, the principled way of introducing MIMO mixing without breaking SSD's parallel form, and the clarity of the theoretical exposition. The author response significantly strengthened the empirical picture by adding the SCP1 evaluation, expanded SSM baselines on SSP (where MIMOMamba achieves the best metrics), ablations over polynomial degree, number of heads, and rank, and efficiency benchmarks for two implementation variants.

Limitations remain: empirical evaluation is still narrower than is typical for SSM papers (no language-modeling or large-scale experiments), the SCP1 result shows MIMOMamba on par with rather than clearly exceeding strong SISO SSMs such as S6, and the FFT-based implementation is not yet a clear practical win at the optimal polynomial degree. I view these as scope limitations rather than fundamental issues; the theoretical contribution (a clean algebraic generalization of SSD that makes cross-channel dynamics tractable) is genuinely novel and likely to motivate follow-up work, and the evidence is sufficient to demonstrate that the construction works. The polynomial-ring formulation is also complementary to the recent Mamba-3 MIMO design (which extends SSD via a different MIMO route), and having multiple principled MIMO instantiations of SSD is itself useful for the community.

Overall, I recommend acceptance.